# Federated Minimax Optimization with Client Heterogeneity

**Pranay Sharma**                                  *pranaysh@andrew.cmu.edu*
*Department of Electrical and Computer Engineering*
*Carnegie Mellon University*

**Rohan Panda**                                    *rohanpan@andrew.cmu.edu*
*Department of Electrical and Computer Engineering*
*Carnegie Mellon University*

**Gauri Joshi**                                    *gaurij@andrew.cmu.edu*
*Department of Electrical and Computer Engineering*
*Carnegie Mellon University*

**Reviewed on OpenReview:** *https://openreview.net/forum?id=NnUmg1chLL*

## Abstract

Minimax optimization has seen a surge in interest with the advent of modern applications such as GANs, and it is inherently more challenging than simple minimization. The difficulty is exacerbated by the training data residing at multiple edge devices or *clients*, especially when these clients can have heterogeneous datasets and heterogeneous local computation capabilities. We propose a general federated minimax optimization framework that subsumes such settings and several existing methods like Local SGDA. We show that naive aggregation of model updates made by clients running unequal number of local steps can result in optimizing a mismatched objective function – a phenomenon previously observed in standard federated minimization. To fix this problem, we propose normalizing the client updates by the number of local steps. We analyze the convergence of the proposed algorithm for classes of nonconvex-concave and nonconvex-nonconcave functions and characterize the impact of heterogeneous client data, partial client participation, and heterogeneous local computations. For all the function classes considered, we significantly improve the existing computation and communication complexity results. Experimental results support our theoretical claims.

## 1 Introduction

The massive surge in machine learning (ML) research in the past decade has brought forth new applications that cannot be modeled as simple minimization problems. Many of these problems, including generative adversarial networks (GANs) Goodfellow et al. (2014); Arjovsky et al. (2017); Sanjabi et al. (2018), adversarial neural network training Madry et al. (2018), robust optimization Namkoong & Duchi (2016); Mohajerin Esfahani & Kuhn (2018), and fair machine learning Madras et al. (2018); Mohri et al. (2019), have an underlying min-max structure. However, the underlying problem is often nonconvex, while classical minimax theory deals almost exclusively with convex-concave problems.

Another feature of modern ML applications is the inherently distributed nature of the training data Xing et al. (2016). The data collection is often outsourced to edge devices or *clients*. However, the clients may then be unable (due to resource constraints) or unwilling (due to privacy concerns) to share their data with a *central server*. Federated Learning (FL) Konečnỳ et al. (2016); Kairouz et al. (2019) was proposed to alleviate this problem. In exchange for retaining control of their data, the clients shoulder some of the computational load, and run part of the training process locally, using only their own data. The communication with the server is infrequent, leading to further resource savings. Since its introduction, FL has been an active area of research, with some remarkable successes Li et al. (2020); Wang et al. (2021). Research has shown practical

Table 1: Comparison of (**per client**) stochastic gradient complexity and the number of communication rounds needed to reach an $\epsilon$-stationary solution (Definition 1), for different classes of nonconvex minimax problems. Here, $n$ is the total number of clients. For a fair comparison with existing works, our results in this table are specialized to the case when all clients (i) have equal weights ($p_i = 1/n$), (ii) perform equal number of local updates ($\tau_i = \tau$), and (iii) use the same local update algorithm SGDA. See Table 2 for comparison under more general settings, when (i)-(iii) do not hold.

| Work | Setting and Assumptions | | Full Client Participation (FCP) | |
|---|---|---|---|---|
| | System Heterogeneity[a] | Partial Client Participation | Stochastic Gradient Complexity | Communication Rounds |
| Nonconvex-Strongly-concave (NC-SC)/Nonconvex-Polyak-Łojasiewicz (NC-PL): Theorem 1 | | | | |
| ($n = 1$) Lin et al. (2020a) | - | - | $\mathcal{O}(1/\epsilon^4)$ | - |
| Sharma et al. (2022) | ✗ | ✗ | $\mathcal{O}(1/(n\epsilon^4))$ | $\mathcal{O}(1/\epsilon^3)$ |
| Yang et al. (2022a) | ✗ | ✓ | $\mathcal{O}(1/(n\epsilon^4))$ | $\mathcal{O}(1/\epsilon^2)$ |
| **Ours:** (Corollary 1.2, Remark 3) | ✓ | ✓ | $\mathcal{O}\left(1/(n\epsilon^4)\right)$ | $\mathcal{O}\left(1/\epsilon^2\right)$ |
| Nonconvex-Concave (NC-C): Theorem 2 | | | | |
| ($n = 1$) Lin et al. (2020a) | - | - | $\mathcal{O}(1/\epsilon^8)$ | - |
| Sharma et al. (2022) | ✗ | ✗ | $\mathcal{O}(1/(n\epsilon^8))$ | $\mathcal{O}(1/\epsilon^7)$ |
| **Ours:** (Corollary 2.2) | ✓ | ✓ | $\mathcal{O}\left(1/(n\epsilon^8)\right)$ | $\mathcal{O}\left(1/\epsilon^4\right)$ |
| Nonconvex-One-point-concave (NC-1PC): Theorem 2 | | | | |
| Deng & Mahdavi (2021) | ✗ | ✗ | $\mathcal{O}(1/\epsilon^{12})$ | $\mathcal{O}(n^{1/6}/\epsilon^8)$ |
| Sharma et al. (2022) | ✗ | ✗ | $\mathcal{O}(1/\epsilon^8)$ | $\mathcal{O}(1/\epsilon^7)$ |
| **Ours:** (Remark 5) | ✓ | ✓ | $\mathcal{O}\left(1/(n\epsilon^8)\right)$ | $\mathcal{O}\left(1/\epsilon^4\right)$ |

[a] Individual clients can run an unequal number of local iterations, using different local optimizers (see Section 4).

benefits of, and provided theoretical justifications for commonly used practical techniques, such as, multiple local updates at the clients Stich (2018); Khaled et al. (2020); Koloskova et al. (2020); Wang & Joshi (2021), partial client participation Yang et al. (2021), communication compression Hamer et al. (2020); Chen et al. (2021). Further, impact of heterogeneity in the clients' local data Zhao et al. (2018); Sattler et al. (2019), as well as their system capabilities Wang et al. (2020); Mitra et al. (2021) has been studied. However, all this research has been focused almost solely on simple minimization problems.

With its increasing usage in large-scale applications, FL systems must adapt to a wide range of clients. Data heterogeneity has received significant attention from the community. However, system-level heterogeneity remains relatively unexplored. The effect of client variability or *heterogeneity* can be controlled by forcing all the clients to carry out an equal number of local updates and utilize the same local optimizer Yu et al. (2019); Haddadpour et al. (2019). However, this approach is inefficient if the client dataset sizes are widely different. Also, it would entail faster clients sitting idle for long durations Reisizadeh et al. (2022); Tziotis et al. (2022), waiting for stragglers to finish. Additionally, using the same optimizer might be inefficient or expensive for clients, depending on their system capabilities. Therefore, adapting to system-level heterogeneity forms a desideratum for real-world FL schemes.

**Contributions.** We consider a general federated minimax optimization framework, in the presence of both inter-client data and system heterogeneity. **System heterogeneity** means the participating clients can run an unequal number of local steps, and utilize different local solvers. We consider the problem

$$\min_{\mathbf{x} \in \mathbb{R}^{d_x}} \max_{\mathbf{y} \in \mathcal{Y}} \left\{ F(\mathbf{x}, \mathbf{y}) := \sum_{i=1}^{n} p_i f_i(\mathbf{x}, \mathbf{y}) \right\}, \tag{1}$$

where $f_i$ is the local loss of client $i$, $p_i$ is the weight assigned to client $i$ (e.g., the relative sample size), and $n$ is the total number of clients. We study several classes of nonconvex minimax problems (see Table 1). Further,

- In our proposed algorithm, the participating clients may each perform different number of local steps, with different local optimizers. In this setting, naive aggregation of local model updates (as done in existing methods like Local Stochastic Gradient Descent Ascent) may lead to convergence in terms of a mismatched global objective (Corollaries 1.1, 2.1). We propose a simple normalization strategy to fix this problem.

- We achieve order-optimal or state-of-the-art computation complexity and significantly improve the communication complexity of existing methods (Corollaries 1.2, 2.2).
- Under the special case where all the clients (i) are assigned equal weights $p_i = 1/n$ in (1), (ii) carry out equal number of local updates ($\tau_i = \tau$ for all $i$), and (iii) utilize the same local-update algorithm, our results become directly comparable with existing work (see Table 1) and improve upon them as follows.

  1. For nonconvex-strongly-concave (NC-SC - Corollary 1.2) and nonconvex-PL (NC-PL - Remark 3) problems, our method has the order-optimal gradient complexity $\mathcal{O}(1/(n\epsilon^4))$. Further, we improve the communication from $\mathcal{O}(1/\epsilon^3)$ in Sharma et al. (2022) to $\mathcal{O}(1/\epsilon^2)$.[1]
  2. For nonconvex-concave (NC-C - Corollary 2.2) and nonconvex-one-point-concave (NC-1PC - Remark 5) problems, we achieve state-of-the-art gradient complexity, while significantly improving the communication costs from $\mathcal{O}(1/\epsilon^7)$ in Sharma et al. (2022) to $\mathcal{O}(1/\epsilon^4)$. For NC-1PC functions, we prove the linear speedup in gradient complexity with $n$ that was conjectured in Sharma et al. (2022).
  3. As an intermediate result in our proof, we prove the theoretical convergence of Local SGD for one-point-convex function minimization (see Lemma C.5 in Appendix C.4). The achieved convergence rate is the same as that shown for convex minimization in the existing literature Khaled et al. (2020).

It is worth pointing out that our proof technique is different from existing minimax literature (e.g., Sharma et al. (2022); Yang et al. (2022b)). With all the clients carrying out the same number of local steps, the existing federated analyses rely on virtual sequences of average iterates, to mimic the proof steps in centralized settings Lin et al. (2020a); Yang et al. (2022c). In our case, since different clients run different number of local steps, this strategy is no longer viable (see Remark 9).

## 2 Related Work

### 2.1 Single-client minimax

**Nonconvex-Strongly-concave (NC-SC).** To our knowledge, Lin et al. (2020a) is the first work to analyze a single-loop algorithm for stochastic (and deterministic) NC-SC problems. Although the $\mathcal{O}(\kappa^3/\epsilon^4)$ complexity shown is optimal in $\epsilon$, the algorithm required $\mathcal{O}(\epsilon^{-2})$ batch-size. Qiu et al. (2020) utilized momentum to achieve $\mathcal{O}(\epsilon^{-4})$ convergence with $\mathcal{O}(1)$ batch-size. Recent works Yang et al. (2022c); Sharma et al. (2022) achieve the same rate without momentum. Yang et al. (2022c) also improved the dependence on the condition number $\kappa$. Second-order stationarity for NC-SC has been recently studied in Luo & Chen (2021). Lower bounds for this problem class have appeared in Luo et al. (2020); Li et al. (2021); Zhang et al. (2021).

**Nonconvex-Concave (NC-C).** Again, Lin et al. (2020a) was the first to analyze a single-loop algorithm for stochastic NC-C problems, proving $\mathcal{O}(\epsilon^{-8})$ complexity. In deterministic problems, this has been improved using nested Nouiehed et al. (2019); Thekumparampil et al. (2019) as well as single-loop Xu et al. (2020); Zhang et al. (2020) algorithms. For stochastic problems, Rafique et al. (2021) and the recent work Zhang et al. (2022) improved the complexity to $\mathcal{O}(\epsilon^{-6})$. However, both the algorithms have a nested structure, which at every step, solve a simpler problem iteratively. Achieving $\mathcal{O}(\epsilon^{-6})$ complexity with a single-loop algorithm has so far proved elusive.

### 2.2 Distributed/Federated Minimax

Recent years have also seen an increasing body of work in distributed minimax optimization. Some of this work is focused on decentralized settings, as in Rogozin et al. (2021); Beznosikov et al. (2021b,c); Metelev et al. (2022).

Of immediate relevance to us is the federated setting, where clients carry out multiple local updates between successive communication rounds. The relevant works which focused on convex-concave problems include Reisizadeh et al. (2020); Hou et al. (2021); Liao et al. (2021); Sun & Wei (2022). Special classes of nonconvex

---

[1]The recent work Yang et al. (2022a) proposes FSGDA algorithm and also achieves $\mathcal{O}(1/\epsilon^2)$ communication cost for NC-PL functions. However, our work is more general since we allow different number of local steps and different local solvers at the clients.

minimax problems in the federated setting have been studied in recent works, such as, nonconvex-linear Deng et al. (2020), nonconvex-PL Deng & Mahdavi (2021); Xie et al. (2021), and nonconvex-one-point-concave Deng & Mahdavi (2021). The complexity guarantees for several function classes considered in Deng & Mahdavi (2021) were further improved in Sharma et al. (2022). However, all these works consider specialized federated settings, either assuming full-client participation, or system-wise identical clients, each carrying out equal number of local updates. As we see in this paper, partial client participation is the most source of error in simple FL algorithms. Also, system-level heterogeneity can have crucial implications on the algorithm performance.

**Comparison with Wang et al. (2020); Sharma et al. (2022); Yang et al. (2022a).** Wang et al. (2022a) was, to our knowledge, the first work to consider the problem of system heterogeneity in simple minimization problems, and proposed a normalized averaging scheme to avoid optimizing an inconsistent objective. Compared to Wang et al. (2020), we consider a more challenging problem and achieve higher communication savings (Table 1)[2]. Sharma et al. (2022) studied minimax problems in the federated setting but assumed an equal number of SGDA-like local updates, with full client participation. The recent work Yang et al. (2022a) considers NC-SC problem with full and partial client participation and achieves similar communication savings as ours. In comparison, our work considers a more general minimax FL framework with partial client participation, clients running an unequal number of local updates, and using different local solvers. Further, we analyze multiple classes of nonconvex-concave and nonconvex-nonconcave functions, improving the communication and computation complexity of existing minimax methods.

## 3 Preliminaries

**Notations.** We let $\|\cdot\|$ denote the Euclidean norm $\|\cdot\|_2$. Given a positive integer $m$, the set $\{1, 2, \ldots, m\}$ is denoted by $[m]$. Vectors at client $i$ are denoted with subscript $i$, e.g., $\mathbf{x}_i$, while iteration indices are denoted using superscripts, e.g., $\mathbf{y}^{(t)}$ or $\mathbf{y}^{(t,k)}$. Given a function $g$, we define its gradient vector as $\left[\nabla_x g(\mathbf{x}, \mathbf{y})^\top, \nabla_y g(\mathbf{x}, \mathbf{y})^\top\right]^\top$, and its stochastic gradient as $\nabla g(\mathbf{x}, \mathbf{y}; \xi)$, where $\xi$ denotes the randomness.

**Convergence Metrics.** In the presence of nonconvexity, we can only prove convergence to an *approximate* stationary point, which is defined next.

**Definition 1** ($\epsilon$-Stationarity). A point $\mathbf{x}$ is an $\epsilon$-stationary point of a differentiable function $g$ if $\|\nabla g(\mathbf{x})\| \leq \epsilon$.

**Definition 2.** Stochastic Gradient (SG) complexity is the total number of gradients computed by all the clients during the course of the algorithm.

In special cases, where all the clients are weighted equally ($p_i = 1/n$, for all $i \in [n]$) and carry out equal number of local steps $\tau$, we state the *per-client* gradient complexity for comparison with existing work. See Table 1 and Corollaries 1.2 and 2.2.

**Definition 3** (Communication Rounds). During a single communication round, the server sends its *global* model to a set of clients, which carry out multiple local updates starting from the same model, and return their *local* vectors to the server. The server then aggregates these local vectors to arrive at a new global model. Throughout this paper, we denote the number of communication rounds by $T$.

Next, we discuss some assumptions used in the paper.

**Assumption 1** (Smoothness). Each local function $f_i$ is differentiable and has Lipschitz continuous gradients. That is, there exists a constant $L_f > 0$ such that at each client $i \in [n]$, for all $\mathbf{x}, \mathbf{x}' \in \mathbb{R}^{d_1}$ and $\mathbf{y}, \mathbf{y}' \in \mathcal{Y}$,

$$\|\nabla f_i(\mathbf{x}, \mathbf{y}) - \nabla f_i(\mathbf{x}', \mathbf{y}')\| \leq L_f \|(\mathbf{x}, \mathbf{y}) - (\mathbf{x}', \mathbf{y}')\|.$$

**Assumption 2** (Bounded Diameter). The constraint set $\mathcal{Y}$ is convex and bounded.

---

[2]Under the conditions $p_i = 1/n, \tau_i = \tau$ for all $i$, for smooth minimization problems, Wang et al. (2020) requires $\mathcal{O}(1/\epsilon^3)$ communication rounds. For NC-SC problems (a harder problem class), we show an improved $\mathcal{O}(1/\epsilon^2)$ communication rounds.

**Assumption 3** (*Local* Variance)**.** The stochastic gradient oracle at each client is *unbiased*. Also, there exist constants $\sigma_L, \beta_L \geq 0$ such that at each client $i \in [n]$, for all $\mathbf{x}, \mathbf{y}$,

$$\mathbb{E}_{\xi_i}[\nabla f_i(\mathbf{x}, \mathbf{y}; \xi_i)] = \nabla f_i(\mathbf{x}, \mathbf{y}),$$
$$\mathbb{E}_{\xi_i} \|\nabla f_i(\mathbf{x}, \mathbf{y}; \xi_i) - \nabla f_i(\mathbf{x}, \mathbf{y})\|^2 \leq \beta_L^2 \|\nabla f_i(\mathbf{x}, \mathbf{y})\|^2 + \sigma_L^2.$$

**Assumption 4** (*Global* Heterogeneity)**.** For any set of non-negative weights $\{w_i\}_{i=1}^n$ such that $\sum_{i=1}^n w_i = 1$, there exist constants $\beta_G \geq 1, \sigma_G \geq 0$ such that for all $\mathbf{x}, \mathbf{y}$,

$$\sum_{i=1}^n w_i \|\nabla_x f_i(\mathbf{x}, \mathbf{y})\|^2 \leq \beta_G^2 \left\|\sum_{i=1}^n w_i \nabla_x f_i(\mathbf{x}, \mathbf{y})\right\|^2 + \sigma_G^2,$$
$$\sum_{i=1}^n w_i \|\nabla_y f_i(\mathbf{x}, \mathbf{y})\|^2 \leq \beta_G^2 \left\|\sum_{i=1}^n w_i \nabla_y f_i(\mathbf{x}, \mathbf{y})\right\|^2 + \sigma_G^2.$$

If all $f_i$'s are identical, we have $\beta_G = 1$, and $\sigma_G = 0$.

Most existing work uses simplified versions of Assumptions 3, 4, assuming $\beta_L = 0$ and/or $\beta_G = 0$.

## 4 Algorithm for Heterogeneous Federated Minimax Optimization

In this section, we propose a federated minimax algorithm to handle system heterogeneity across clients.

### 4.1 Limitations of Local SGDA

Following the success of FedAvg McMahan et al. (2017) in FL, Deng & Mahdavi (2021) was the first to explore a simple extension Local stochastic gradient descent-ascent (SGDA) in minimax problems. Between successive communication rounds, clients take multiple simultaneous descent/ascent steps to respectively update the min-variable $\mathbf{x}$ and max-variable $\mathbf{y}$. Subsequent work in Sharma et al. (2022) improved the convergence results and showed that LocalSGDA achieves optimal gradient complexity for several classes of nonconvex minimax problems. However, existing work on LocalSGDA also assumes the participation of all $n$ clients in every communication round. More crucially, as observed with simple minimization problems Wang et al. (2020), if clients carry out an unequal number of local updates, or if their local optimizers are not all the same, LocalSGDA (like FedAvg) might converge to the stationary point of a different objective. This is further discussed in Sections 5.1 and 5.2, and illustrated in Figure 1, where the learning process gets disproportionately skewed towards the clients carrying out more local updates.

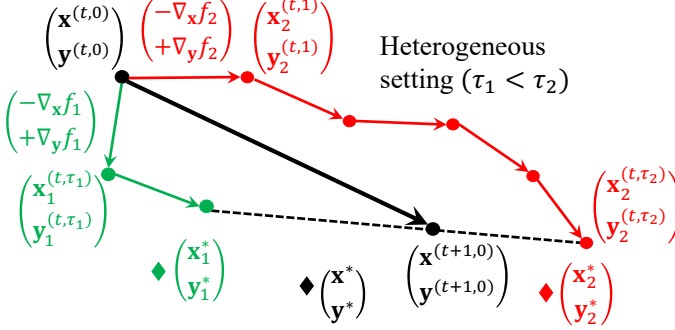

Figure 1: FedAvg with heterogeneous local updates. The green (red) triangle represents the local optimizer of $f_1(f_2)$, while $(\mathbf{x}^*, \mathbf{y}^*)$ is the global optimizer. The number of local updates at the clients is $\tau_1 = 2$, $\tau_2 = 5$.

**Generalized Local SGDA Update Rule.** To understand this mismatched convergence phenomenon with naive aggregation in local SGDA, recall that Local SGDA updates are of the form

$$\mathbf{x}^{(t+1)} = \mathbf{x}^{(t)} + \gamma_x^s \sum_{i=1}^n p_i \Delta_{\mathbf{x},i}^{(t)}, \qquad \mathbf{y}^{(t+1)} = \mathbf{y}^{(t)} + \gamma_y^s \sum_{i=1}^n p_i \Delta_{\mathbf{y},i}^{(t)},$$

where $\gamma_x^s, \gamma_y^s$ are the server learning rates, $\Delta_{\mathbf{x},i}^{(t)} = \frac{1}{\eta_x^c}\big(\mathbf{x}_i^{(t,\tau_i^{(t)})} - \mathbf{x}^{(t)}\big)$, $\Delta_{\mathbf{y},i}^{(t)} = \frac{1}{\eta_y^c}\big(\mathbf{y}_i^{(t,\tau_i^{(t)})} - \mathbf{y}^{(t)}\big)$ are the scaled local updates. $\mathbf{x}_i^{(t,\tau_i^{(t)})}$ is the iterate at client $i$ after taking $\tau_i^{(t)}$ local steps, and $\eta_x^c, \eta_y^c$ are the client learning rates. Let us consider a generalized version of this update rule where $\Delta_{\mathbf{x},i}^{(t)}, \Delta_{\mathbf{y},i}^{(t)}$ are linear combinations of local stochastic gradients computed by client $i$, as $\Delta_{\mathbf{y},i}^{(t)} = \sum_{k=0}^{\tau_i^{(t)}-1} a_i^{(t,k)} \nabla_y f_i(\mathbf{x}_i^{(t,k)}, \mathbf{y}_i^{(t,k)}; \xi_i^{(t,k)})$, where $a_i^{(t,k)} \geq 0$. Commonly used client optimizers, such as, SGD, local momentum, variable local learning rates can be accommodated in this general form (see Appendix A.1 for some examples). For this more general form, we can rewrite the $\mathbf{x}, \mathbf{y}$ updates at the server as follows

$$\begin{aligned}
\mathbf{x}^{(t+1)} &= \mathbf{x}^{(t)} - \gamma_x^s \sum_{i=1}^n p_i \mathbf{G}_{\mathbf{x},i}^{(t)} \frac{\bar{\boldsymbol{a}}_i^{(t)}}{\|\bar{\boldsymbol{a}}_i^{(t)}\|_1} \|\bar{\boldsymbol{a}}_i^{(t)}\|_1 \\
&= \mathbf{x}^{(t)} - \underbrace{\Big(\sum_{j=1}^n p_j \|\bar{\boldsymbol{a}}_j^{(t)}\|_1\Big)}_{\tau_{\text{eff}}^{(t)}} \gamma_x^s \sum_{i=1}^n \underbrace{\frac{p_i \|\bar{\boldsymbol{a}}_i^{(t)}\|_1}{\sum_{j=1}^n p_j \|\bar{\boldsymbol{a}}_j^{(t)}\|_1}}_{w_i} \underbrace{\frac{\mathbf{G}_{\mathbf{x},i}^{(t)} \bar{\boldsymbol{a}}_i^{(t)}}{\|\bar{\boldsymbol{a}}_i^{(t)}\|_1}}_{\mathbf{g}_{\mathbf{x},i}^{(t)}},
\end{aligned} \qquad (2)$$

$$\mathbf{y}^{(t+1)} = \mathbf{y}^{(t)} + \tau_{\text{eff}}^{(t)} \gamma_y^s \sum_{i=1}^n w_i \mathbf{g}_{\mathbf{y},i}^{(t)},$$

where $\mathbf{G}_{\mathbf{x},i}^{(t)} = [\nabla_x f_i(\mathbf{x}_i^{(t,k)}, \mathbf{y}_i^{(t,k)}; \xi_i^{(t,k)})]_{k=0}^{\tau_i^{(t)}} \in \mathbb{R}^{d_x \times \tau_i^{(t)}}$ contains the $\tau_i^{(t)}$ stochastic gradients stacked column-wise, $\bar{\boldsymbol{a}}_i^{(t)} = [a_i^{t,0}, a_i^{t,1}, \ldots, a_i^{t,\tau_i^{(t)}-1}]^\top$, $\mathbf{g}_{\mathbf{x},i}^{(t)}, \mathbf{g}_{\mathbf{y},i}^{(t)}$ are the normalized aggregates of the stochastic gradients and $\tau_{\text{eff}}^{(t)}$ is the *effective* number of local steps. Note that for simplicity, we assume that the constraint set $\mathcal{Y}$ has a large diameter. However, our algorithm can be easily modified to accommodate projection steps. Similar to

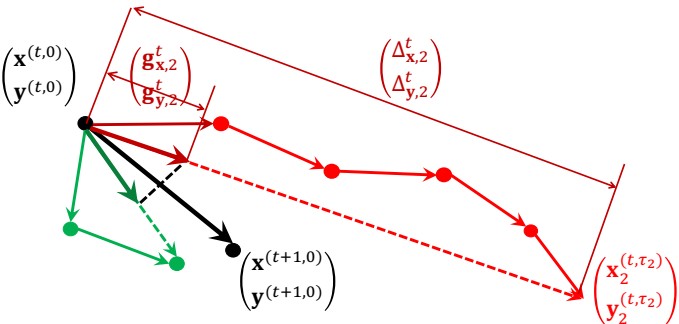

Figure 2: Generalized update rule in (2). Note that $(\mathbf{g}_{\mathbf{x},i}^{(t)}, \mathbf{g}_{\mathbf{y},i}^{(t)}) = \frac{1}{\tau_i}(\Delta_{\mathbf{x},i}^{(t)}, \Delta_{\mathbf{y},i}^{(t)})$. Also, at the server, the weighted sum $\sum_{i=1}^n w_i \mathbf{g}_{\mathbf{x},i}^{(t)}$ gets scaled by $\tau_{\text{eff}}^{(t)}$.

the observation for simple minimization problems in Wang et al. (2020), we see in Theorems 1, 2 that the resulting iterates of this general algorithm end up converging to the stationary point of a different objective $\widetilde{F} = \sum_{i=1}^n w_i f_i$. Further, in Corollary 1.1, we observe that this mismatch is a result of using weights $w_i$ in (2) to weigh the clients' contribution.

## 4.2 Proposed Normalized Federated Minimax Algorithm

From the generalized update rule, we can see that setting the weights $w_i$ equal to $p_i$ will ensure that the surrogate objective $\tilde{F}$ matches with the original global objective $F$. Setting $w_i = p_i$ results in normalization

---

**Algorithm 1** Fed-Norm-SGDA and Fed-Norm-SGDA+

---

1: **Input:** initialization $\mathbf{x}^{(0)}, \mathbf{y}^{(0)}$, Number of communication rounds $T$, learning rates: client $\{\eta_x^c, \eta_y^c\}$, server $\{\gamma_x^s, \gamma_y^s\}$, #local-updates $\{\tau_i^{(t)}\}_{i,t}$, $S$, $s = -1$

2: **for** $t = 0$ to $T - 1$ **do**

3:      Server selects client set $\mathcal{C}^{(t)}$; sends them $(\mathbf{x}^{(t)}, \mathbf{y}^{(t)})$

4:      **if** $t \bmod S = 0$ **then**

5:          $s \leftarrow s + 1$

6:          Server sends $\widehat{\mathbf{x}}^{(s)} = \mathbf{x}^{(t)}$ to clients in $\mathcal{C}^{(t)}$

7:      **end if**

8:      $\mathbf{x}_i^{(t,0)} = \mathbf{x}^{(t)}$, $\mathbf{y}_i^{(t,0)} = \mathbf{y}^{(t)}$ for $i \in \mathcal{C}^{(t)}$

9:      **for** $k = 0, \ldots, \tau_i^{(t)} - 1$ **do**

10:          $\mathbf{x}_i^{(t,k+1)} = \mathbf{x}_i^{(t,k)} - \eta_x^c a_i^{(t,k)} \nabla_x f_i(\mathbf{x}_i^{(t,k)}, \mathbf{y}_i^{(t,k)}; \xi_i^{(t,k)})$

11:          $\mathbf{y}_i^{(t,k+1)} = \mathbf{y}_i^{(t,k)} + \eta_y^c a_i^{(t,k)} \nabla_y f_i(\widehat{\mathbf{x}}^{(s)}, \mathbf{y}_i^{(t,k)}; \xi_i^{(t,k)})$        # $\mathbf{y}$-update for Fed-Norm-SGDA+

12:          $\mathbf{y}_i^{(t,k+1)} = \mathbf{y}_i^{(t,k)} + \eta_y^c a_i^{(t,k)} \nabla_y f_i(\mathbf{x}_i^{(t,k)}, \mathbf{y}_i^{(t,k)}; \xi_i^{(t,k)})$        # $\mathbf{y}$-update for Fed-Norm-SGDA

13:      **end for**

14:      Client $i$ aggregates its gradients to compute $\mathbf{g}_{\mathbf{x},i}^{(t)}, \mathbf{g}_{\mathbf{y},i}^{(t)}$

15:      $\mathbf{g}_{\mathbf{x},i}^{(t)} = \sum_{k=0}^{\tau_i^{(t)}-1} \frac{a_i^{(t,k)}}{\|\bar{\boldsymbol{a}}_i^{(t)}\|_1} \nabla_x f_i(\mathbf{x}_i^{(t,k)}, \mathbf{y}_i^{(t,k)}; \xi_i^{(t,k)})$

16:      $\mathbf{g}_{\mathbf{y},i}^{(t)} = \sum_{k=0}^{\tau_i^{(t)}-1} \frac{a_i^{(t,k)}}{\|\bar{\boldsymbol{a}}_i^{(t)}\|_1} \nabla_y f_i(\widehat{\mathbf{x}}^{(s)}, \mathbf{y}_i^{(t,k)}; \xi_i^{(t,k)})$

17:      $\mathbf{g}_{\mathbf{y},i}^{(t)} = \sum_{k=0}^{\tau_i^{(t)}-1} \frac{a_i^{(t,k)}}{\|\bar{\boldsymbol{a}}_i^{(t)}\|_1} \nabla_y f_i(\mathbf{x}_i^{(t,k)}, \mathbf{y}_i^{(t,k)}; \xi_i^{(t,k)})$

18:      Clients $i \in \mathcal{C}^{(t)}$ communicate $\{\mathbf{g}_{\mathbf{x},i}^{(t)}, \mathbf{g}_{\mathbf{y},i}^{(t)}\}$ to the server

19:      Server computes aggregate vectors $\{\mathbf{g}_{\mathbf{x}}^{(t)}, \mathbf{g}_{\mathbf{y}}^{(t)}\}$ using (3)

20:      Server step: $\left\{\mathbf{x}^{(t+1)} = \mathbf{x}^{(t)} - \tau_{\text{eff}}^{(t)} \gamma_x^s \mathbf{g}_{\mathbf{x}}^{(t)}, \quad \mathbf{y}^{(t+1)} = \mathbf{y}^{(t)} + \tau_{\text{eff}}^{(t)} \gamma_y^s \mathbf{g}_{\mathbf{y}}^{(t)}\right.$

21: **end for**

22: **Return:** $\bar{\mathbf{x}}^{(T)}$ drawn uniformly at random from $\{\mathbf{x}^{(t)}\}_{t=1}^T$

---

of the local progress at each client before their aggregation at the server. As a result, we can preserve convergence to a stationary point of the original objective function $F$, even with heterogeneous $\{\tau_i^{(t)}\}$, as we see in Theorem 1 and Theorem 2.

The algorithm follows the steps given in Algorithm 1. In each communication round $t$, the server selects a client set $\mathcal{C}^{(t)}$ and communicates its model parameters $(\mathbf{x}^{(t)}, \mathbf{y}^{(t)})$ to these clients. The selected clients then run multiple local stochastic gradient steps. The number of local steps $\{\tau_i^{(t)}\}$ can vary across clients and across rounds. At the end of $\tau_i^{(t)}$ local steps, client $i$ aggregates its local stochastic gradients into $\{\mathbf{g}_{\mathbf{x},i}^{(t)}, \mathbf{g}_{\mathbf{y},i}^{(t)}\}$, which are then sent to the server. Note that the gradients at client $i$, $\{\nabla f_i(\cdot, \cdot; \xi_i^{(t,k)})\}_{k=0}^{\tau_i^{(t)}}$, are normalized by $\|\bar{\boldsymbol{a}}_i^{(t)}\|_1$, where $\bar{\boldsymbol{a}}_i^{(t)} = [a_i^{t,0}, a_i^{t,1}, \ldots, a_i^{t,\tau_i^{(t)}-1}]^\top$ is the vector of weights assigned to individual stochastic gradients in the local updates.[3] The server aggregates these local vectors to compute global direction estimates $\mathbf{g}_{\mathbf{x}}^{(t)}, \mathbf{g}_{\mathbf{y}}^{(t)}$, which are then used to update the server model parameters $(\mathbf{x}^{(t)}, \mathbf{y}^{(t)})$.

---

[3]For LocalSGDA Deng & Mahdavi (2021); Sharma et al. (2022), $a_i^{(t,k)} = 1$ for all $i \in [n], t \in [T], k \in [\tau_i^{(t)}]$ and $\|\bar{\boldsymbol{a}}_i^{(t)}\|_1 = \tau_i^{(t)}$. Therefore, $\mathbf{g}_{\mathbf{x},i}^{(t)}, \mathbf{g}_{\mathbf{y},i}^{(t)}$ are simply the average of the stochastic gradients computed in the $t$-th round.

**Client Selection.** In each round $t$, the server samples $|\mathcal{C}^{(t)}|$ clients uniformly at random *without replacement* (WOR). While aggregating client updates at the server, client $i$ update is weighed by $\tilde{w}_i = w_i n / |\mathcal{C}^{(t)}|$, i.e.,

$$\mathbf{g}_{\mathbf{x}}^{(t)} = \sum_{i \in \mathcal{C}^{(t)}} \tilde{w}_i \mathbf{g}_{\mathbf{x},i}^{(t)}, \qquad \mathbf{g}_{\mathbf{y}}^{(t)} = \sum_{i \in \mathcal{C}^{(t)}} \tilde{w}_i \mathbf{g}_{\mathbf{y},i}^{(t)}. \tag{3}$$

Note that $\mathbb{E}_{\mathcal{C}^{(t)}}[\mathbf{g}_{\mathbf{x}}^{(t)}] = \sum_{i=1}^{n} w_i \mathbf{g}_{\mathbf{x},i}^{(t)}, \mathbb{E}_{\mathcal{C}^{(t)}}[\mathbf{g}_{\mathbf{y}}^{(t)}] = \sum_{i=1}^{n} w_i \mathbf{g}_{\mathbf{y},i}^{(t)}.$

## 5 Convergence Results

Next, we present the convergence results for different classes of nonconvex minimax problems. For simplicity, throughout this section we assume the parameters utilized in Algorithm 1 to be fixed across $t$. Therefore, $a_i^{(t,k)} \equiv a_i^{(k)}$, $\bar{\boldsymbol{a}}_i^{(t)} \equiv \boldsymbol{a}_i$, $\tau_i^{(t)} \equiv \tau_i$, $\tau_{\text{eff}}^{(t)} \equiv \tau_{\text{eff}}$ and $|\mathcal{C}^{(t)}| = P$, for all $t$.

### 5.1 Non-convex-Strongly-Concave (NC-SC) Case

**Assumption 5** ($\mu$-Strong-concavity (SC) in $\mathbf{y}$). A function $f$ is $\mu$-strong concave ($\mu > 0$) in $\mathbf{y}$ if

$$-f(\mathbf{x}, \tilde{\mathbf{y}}) \geq -f(\mathbf{x}, \bar{\mathbf{y}}) - \langle \nabla_y f(\mathbf{x}, \bar{\mathbf{y}}), \tilde{\mathbf{y}} - \bar{\mathbf{y}} \rangle + \frac{\mu}{2} \|\tilde{\mathbf{y}} - \bar{\mathbf{y}}\|^2, \qquad \text{for all } \mathbf{x} \in \mathbb{R}^{d_x}, \text{ and } \bar{\mathbf{y}}, \tilde{\mathbf{y}} \in \mathbb{R}^{d_y}.$$

**General Convergence Result.** We first show that the iterates of Algorithm 1 converge to the stationary point of a surrogate objective $\widetilde{F}$, where $\widetilde{F}(\mathbf{x}, \mathbf{y}) \triangleq \sum_{i=1}^{n} w_i f_i(\mathbf{x}, \mathbf{y})$. $\{w_i\}_{i=1}^{n}$ are the aggregation weights used by the server (Line 19). See Appendix B for the full statement and proof.

**Theorem 1.** *Suppose the local loss functions $\{f_i\}_i$ satisfy Assumptions 1, 2, 3, 4, 5. Suppose the server selects $|\mathcal{C}^{(t)}| = P$ clients in each round $t$. Given appropriate choices of client and server learning rates, $(\eta_x^c, \eta_y^c)$ and $(\gamma_x^s, \gamma_y^s)$ respectively (see Appendix B.2), the iterates generated by Fed-Norm-SGDA satisfy*

$$\min_{t \in [T]} \mathbb{E} \|\nabla \widetilde{\Phi}(\mathbf{x}^{(t)})\|^2 \leq \underbrace{\mathcal{O}\left(\kappa^2 \sigma_G \sqrt{\frac{n-P}{n-1} \frac{E_w}{PT}}\right)}_{\substack{\textit{Partial participation} \\ \textit{error}}} + \underbrace{\mathcal{O}\left(\kappa^2 \sqrt{\frac{\Delta_{\widetilde{\Phi}} + A_w \sigma_L^2 + B_w \beta_L^2 \sigma_G^2}{P \tau_{\textit{eff}} T}}\right)}_{\textit{Error with full synchronization}} + \underbrace{\mathcal{O}\left(\kappa^2 \frac{C_w \sigma_L^2 + D \sigma_G^2}{\bar{\tau}^2 T}\right)}_{\textit{Local updates error}}. \tag{4}$$

*where, $\kappa = L_f / \mu$ is the condition number, $\widetilde{\Phi}(\mathbf{x}) \triangleq \max_y \widetilde{F}(\mathbf{x}, \mathbf{y})$ is the envelope function, $\Delta_{\widetilde{\Phi}} \triangleq \widetilde{\Phi}(\mathbf{x}^{(0)}) - \min_{\mathbf{x}} \widetilde{\Phi}(\mathbf{x})$, $\bar{\tau} = \frac{1}{n} \sum_{i=1}^{n} \tau_i$, $\tau_{\textit{eff}} = \sum_{i=1}^{n} p_i \|\boldsymbol{a}_i\|_1$, $A_w \triangleq n \tau_{\textit{eff}} \sum_{i=1}^{n} \frac{w_i^2 \|\boldsymbol{a}_i\|_2^2}{\|\boldsymbol{a}_i\|_1^2}$, $B_w \triangleq n \tau_{\textit{eff}} \max_i \frac{w_i \|\boldsymbol{a}_i\|_2^2}{\|\boldsymbol{a}_i\|_1^2}$, $C_w \triangleq \sum_{i=1}^{n} w_i (\|\boldsymbol{a}_i\|_2^2 - [\alpha_i^{(t,\tau_i-1)}]^2)$, $D \triangleq \max_i (\beta_L^2 \|\mathbf{a}_{i,-1}\|_2^2 + \|\mathbf{a}_{i,-1}\|_1^2)$, where $\mathbf{a}_{i,-1} \triangleq [a_i^{(0)}, a_i^{(1)}, \dots, a_i^{(\tau_i-2)}]^\top$ and $E_w \triangleq n \max_i w_i.$*

*Remark* 1. The *first* term in (4) results from client subsampling ($P < n$). This explains its dependence on the data heterogeneity $\sigma_G$. The *second* term represents the optimization error for a centralized algorithm (see Appendix C.3 in Lin et al. (2020a)). The *last* term represents *client-drift*, the error if the client(s) run multiple local updates.

Theorem 1 states convergence for a surrogate objective $\widetilde{F}$. Next, we see convergence for the true objective $F$.

**Corollary 1.1** (Convergence in terms of $F$). *Given $\Phi(\mathbf{x}) \triangleq \max_y F(\mathbf{x}, \mathbf{y})$, under the conditions of Theorem 1,*

$$\min_{t \in [T]} \|\nabla \Phi(\mathbf{x}^{(t)})\|^2 \leq 2 \left(2 \chi_{\mathbf{p}\|\mathbf{w}}^2 \beta_H^2 + 1\right) \epsilon_{opt} + 4 \chi_{\mathbf{p}\|\mathbf{w}}^2 \sigma_G^2 + \frac{4 L_f^2}{T} \sum_{t=0}^{T-1} \|\mathbf{y}^*(\mathbf{x}^{(t)}) - \widetilde{\mathbf{y}}^*(\mathbf{x}^{(t)})\|^2. \tag{5}$$

*where $\chi_{\mathbf{p}\|\mathbf{w}}^2 \triangleq \sum_{i=1}^{n} \frac{(p_i - w_i)^2}{w_i}$, $\epsilon_{opt} \triangleq \frac{1}{T} \sum_{t=0}^{T-1} \|\nabla \widetilde{\Phi}(\mathbf{x}^{(t)})\|^2$ denotes the optimization error in (4). If $p_i = w_i$ for all $i \in [n]$, then $\chi_{\mathbf{p}\|\mathbf{w}}^2 = 0$. Also, then $\widetilde{F}(\mathbf{x}, \mathbf{y}) \equiv F(\mathbf{x}, \mathbf{y})$. Therefore, $\mathbf{y}^*(\mathbf{x}) = \arg\max_y F(\mathbf{x}, \mathbf{y})$ and $\widetilde{\mathbf{y}}^*(\mathbf{x}) = \arg\max_y \widetilde{F}(\mathbf{x}, \mathbf{y})$ are identical, for all $\mathbf{x}$. Hence, (5) yields $\min_{t \in [T]} \|\nabla \Phi(\mathbf{x}^{(t)})\|^2 \leq 2 \epsilon_{opt}.$*

It follows from Corollary 1.1 that if in Algorithm 1, the server aggregation weights $\{w_i\}$ (Line 19) are the same as $\{p_i\}$, we get convergence to a stationary point of the true objective $F$. For the rest of this subsection, we assume $w_i = p_i$ for all $i \in [n]$.

Table 2: Comparison of convergence rates of Fed-Norm-SGDA (Theorem 1) and Fed-Norm-SGDA+ (Theorem 2), if all the clients run SGDA/SGDA+ based local updates, i.e., $a_i^{(t,k)} = 1$, for all $i, k, t$. The results are stated for (i) $(p_i = 1/n, \tau_i = \tau, \forall\, i \in [n])$; and (ii) $(p_i \neq p_j)$, $(\tau_i \neq \tau_j)$. The additional factors in (ii) relative to (i) are highlighted in blue. We state the results under partial-client participation (PCP). FCP results follow by choosing $P = n$. For simplicity, we assume uniformly bounded local variance ($\beta_L = 0$ in Assumption 3).

Nonconvex-Strongly-concave (NC-SC)/Nonconvex-Polyak-Łojasiewicz (NC-PL): (Theorem 1, Remark 3)

| System Setting | Convergence Rate |
|---|---|
| $p_i = \frac{1}{n}, \forall\, i \in [n]$
$\tau_i = \tau, \forall\, i \in [n]$ | Sharma et al. (2022) with $P = n$: $\mathcal{O}\left(\frac{\kappa^2 \sigma_L^2}{\sqrt{n\tau T}} + \frac{\kappa^2 n\tau}{T}\left[\sigma_L^2 + \sigma_G^2\right]\right)$
Yang et al. (2022a) with $P < n$: $\mathcal{O}\left(\frac{\sigma_G^2}{\sqrt{PT}}\left(1 - \frac{P}{n}\right) + \frac{1}{\sqrt{PT}}\left[1 + \frac{\sigma_L^2}{\tau} + \sigma_G^2\right]\right)$
**Ours** with $P < n$: $\mathcal{O}\left(\kappa^2 \sigma_G\sqrt{\frac{(n-P)}{(n-1)PT}} + \frac{\kappa^2 \sigma_L}{\sqrt{P\tau T}} + \frac{\kappa}{T}\left[\frac{\sigma_L^2}{\tau} + \sigma_G^2\right]\right)$ |
| **Ours:**
$p_i \neq p_j,\ \tau_i \neq \tau_j$
$\bar{\tau} = \frac{1}{n}\sum_{i=1}^{n}\tau_i$ | $\mathcal{O}\left(\kappa^2 \sigma_G\sqrt{\frac{(n-P)n\max_i p_i}{(n-1)PT}} + \frac{\kappa^2 \sigma_L}{\sqrt{P\tau_{\text{eff}}T}}\sqrt{n\tau_{\text{eff}}\sum_{i=1}^{n}\frac{p_i^2}{\tau_i}} + \frac{\kappa}{T}\left[\frac{\sigma_L^2}{\bar{\tau}}\frac{\sum_{i=1}^{n}p_i\tau_i}{\bar{\tau}} + \sigma_G^2\max_i\frac{\tau_i^2}{\bar{\tau}^2}\right]\right)$ |

Nonconvex-Concave (NC-C)/Nonconvex-One-Point-Concave (NC-1PC): (Theorem 2, Remark 5)

| System Setting | Convergence Rate |
|---|---|
| $p_i = \frac{1}{n}, \forall\, i \in [n]$
$\tau_i = \tau, \forall\, i \in [n]$ | Sharma et al. (2022) with $P = n$: $\mathcal{O}\left(\frac{1}{(\tau PT)^{1/4}}\right) + \mathcal{O}\left(\frac{(n\tau)^{3/2}}{\sqrt{T}}\right)$
**Ours** with $P < n$: $\begin{cases}\mathcal{O}\left(\sqrt{\sigma_G}\left(\frac{n-P}{n-1}\frac{1}{PT}\right)^{1/4} + \sqrt{\sigma_L}\left(\frac{1}{\tau PT}\right)^{1/4}\right) + \\ \mathcal{O}\left(\frac{(\tau P)^{1/4}}{T^{3/4}}\left(1 + \tau\frac{n-P}{n-1}\right)^{-1/4}\right) + \mathcal{O}\left(\frac{1}{T^{3/4}}\left[\frac{\sigma_L^2}{\tau} + (G_{\mathbf{x}}^2 + \sigma_G^2)\right]\right)\end{cases}$ |
| **Ours:**
$p_i \neq p_j,\ \tau_i \neq \tau_j$
$\bar{\tau} = \frac{1}{n}\sum_{i=1}^{n}\tau_i$ | $\mathcal{O}\left(\left(\sqrt{\sigma_G}\left(\frac{n-P}{n-1}\frac{n\max_i p_i}{PT}\right)^{1/4} + \sqrt{\sigma_L}\left(\frac{n\tau_{\text{eff}}}{\tau_{\text{eff}}PT}\sum_{i=1}^{n}\frac{p_i^2}{\tau_i}\right)^{1/4}\right)\sqrt[8]{1 + \frac{n}{P}\|\mathbf{p}\|_2^2}\right)$
$+ \mathcal{O}\left(\frac{(\tau_{\text{eff}}P)^{1/4}}{T^{3/4}}\left(1 + \frac{n}{P}\|\mathbf{p}\|_2^2\right)\left(n\tau_{\text{eff}}\sum_{i=1}^{n}\frac{p_i^2}{\tau_i} + \tau_{\text{eff}}\frac{n-P}{n-1}n\max_i p_i\right)^{-1/4}\right)$
$+ \mathcal{O}\left(\frac{1}{T^{3/4}}\left[\sigma_L^2\frac{\sum_{i=1}^{n}p_i\tau_i}{\bar{\tau}^2} + (G_{\mathbf{x}}^2 + \sigma_G^2)\max_i\frac{\tau_i^2}{\bar{\tau}^2}\right]\right)$ |

In Table 2, we specialize the bound in (4) to SGDA-based local updates. We compare the bound under two cases: **Case 1**: equally-weighted clients ($p_i = 1/n$, for all $i$), all running $\tau_i \equiv \tau$ local updates; and **Case 2**: unequally weighted clients ($p_i \neq p_j$), running unequal local updates ($\tau_i \neq \tau_j$). The setting in **Case 1** has previously been considered in Sharma et al. (2022) (under full participation) and Yang et al. (2022a)[4]. Compared to Sharma et al. (2022), our bound has a smaller local-updates error term. This results in improved communication cost (see Corollary 1.2). The additional factors going from **Case 1** to the more general **Case 2** are highlighted in blue. The following insights can be drawn from Table 2.

- **Partial Client Participation Error:** $\mathcal{O}\left(\frac{\sigma_G}{\sqrt{PT}}\right)$ is the *most significant* component of convergence error. Unlike the other two errors, it does not decrease with increasing local updates $\tau_{\text{eff}}$. Consequently, we do not observe communication savings by performing multiple local updates at the clients. It remains an open problem to achieve speedup in terms of local updates in partial participation settings.
- **Unequal client weights:** if the clients are weighted disparately, we observe an increase in the stochastic gradient complexity. To see this, let $\tau_i \equiv \tau$. The resulting bound is $\mathcal{O}\left(\sigma_G\sqrt{\frac{(n-P)n\|\mathbf{p}\|_\infty}{(n-1)PT}} + \frac{\sigma_L\sqrt{n}\|\mathbf{p}\|_2}{\sqrt{P\tau_{\text{eff}}T}} + \frac{1}{T}\left[\frac{\sigma_L^2}{\bar{\tau}} + \sigma_G^2\right]\right)$. Since $\|\mathbf{p}\|_\infty, \|\mathbf{p}\|_2 \leq 1$, in the worst case (when only one of the clients has all the weight), the

---

[4]The condition number $\kappa$ dependence is not explicitly stated in the results in Yang et al. (2022a).

complexity is worse by a factor of $n$. This happens because the client sampling is not done in proportion to their weights. Rather, the server first samples the clients uniformly, and then scales their updates to get an unbiased estimator (3). We leave exploring non-uniform WOR sampling further as a future direction.

**Corollary 1.2** (Improved Communication Cost). *Suppose all the clients are weighted equally ($p_i = 1/n$ for all $i$), with each carrying out $\tau$ local steps of SGDA. Further, assume $\Phi$ is bounded from below. Then, to reach $\mathbf{x}$ such that $\mathbb{E}\|\nabla\Phi(\mathbf{x})\| \leq \epsilon$,*

- *Under full participation, the per-client gradient complexity of Fed-Norm-SGDA is $T\tau = \mathcal{O}(\kappa^4/(n\epsilon^4))$. The number of communication rounds required is $T = \mathcal{O}(\kappa^2/\epsilon^2)$.*
- *Under partial participation, the per-client gradient complexity of Fed-Norm-SGDA is $\mathcal{O}(\kappa^4/(P\epsilon^4))$. In general, running multiple local updates does not yield any communication savings. However, in the special case when inter-client data heterogeneity $\sigma_G = 0$, the communication cost is $\mathcal{O}(\kappa^2/\epsilon^2)$.*

*Remark* 2. The gradient complexity in Corollary 1.2 is optimal in $\epsilon$, and achieves linear speedup in the number of participating clients. The communication complexity improves the corresponding results in Deng & Mahdavi (2021); Sharma et al. (2022). We match the communication cost in the recent work Yang et al. (2022a). In addition, our work considers a more general FL setting with unequally weighted clients ($p_i \neq p_j$), running unequal local updates ($\tau_i \neq \tau_j$), using distinct local solvers ($\mathbf{a}_i \neq \mathbf{a}_j$).

**Extending the Results to Nonconvex-PL Functions**

**Assumption 6.** A function $f$ satisfies $\mu$-PL condition in $\mathbf{y}$ ($\mu > 0$), if for any fixed $\mathbf{x}$: 1) $\max_{\mathbf{y}'} f(\mathbf{x}, \mathbf{y}')$ has a nonempty solution set; and 2) for all $\mathbf{y}$

$$\|\nabla_y f(\mathbf{x}, \mathbf{y})\|^2 \geq 2\mu(\max_{\mathbf{y}'} f(\mathbf{x}, \mathbf{y}') - f(\mathbf{x}, \mathbf{y})).$$

*Remark* 3. If Assumptions 1, 2, 3, 4 hold, and the global function $F$ satisfies Assumption 6, then for appropriately chosen learning rates (Appendix B.5), the bound in Theorem 1 holds.

## 5.2 Non-convex-Concave (NC-C) Case

In this subsection, we consider smooth nonconvex functions which satisfy the following assumptions.

**Assumption 7** (Concavity). The function $f$ is concave in $\mathbf{y}$ if for a fixed $\mathbf{x} \in \mathbb{R}^{d_1}$, for all $\mathbf{y}, \mathbf{y}' \in \mathbb{R}^{d_2}$,

$$f(\mathbf{x}, \mathbf{y}) \leq f(\mathbf{x}, \mathbf{y}') + \langle \nabla_y f(\mathbf{x}, \mathbf{y}'), \mathbf{y} - \mathbf{y}' \rangle.$$

**Assumption 8** (Lipschitz continuity in $\mathbf{x}$). Given a function $f$, there exists a constant $G_{\mathbf{x}}$, such that for each $\mathbf{y} \in \mathbb{R}^{d_2}$, and all $\mathbf{x}, \mathbf{x}' \in \mathbb{R}^{d_1}$,

$$\|f(\mathbf{x}, \mathbf{y}) - f(\mathbf{x}', \mathbf{y})\| \leq G_{\mathbf{x}} \|\mathbf{x} - \mathbf{x}'\|.$$

The envelope function $\Phi(\mathbf{x}) = \max_{\mathbf{y}} f(\mathbf{x}, \mathbf{y})$ used so far, may no longer be smooth in the absence of a unique maximizer. However, $\Phi(\cdot)$ is weakly convex (Lin et al., 2020a, Lemma 4.7). Therefore, we use the alternate definition of stationarity, proposed in Davis & Drusvyatskiy (2019), utilizing the Moreau envelope of $\Phi$.

**Definition 4** (Moreau Envelope). The function $\phi_\lambda$ is the $\lambda$-Moreau envelope of $\phi$, for $\lambda > 0$, if for all $\mathbf{x} \in \mathbb{R}^{d_x}$,

$$\phi_\lambda(\mathbf{x}) = \min_{\mathbf{x}'} \phi(\mathbf{x}') + \frac{1}{2\lambda} \|\mathbf{x}' - \mathbf{x}\|^2.$$

Drusvyatskiy & Paquette (2019) showed that a small $\|\nabla\phi_\lambda(\mathbf{x})\|$ indicates the existence of some point $\widetilde{\mathbf{x}}$ in the vicinity of $\mathbf{x}$, that is *nearly stationary* for $\phi$. Hence, in our case, we focus on minimizing $\|\nabla\Phi_\lambda(\mathbf{x})\|$.

**Proposed Algorithm.** For nonconvex-concave functions, we use Fed-Norm-SGDA+. The **x**-updates are identical to Fed-Norm-SGDA. For the **y** updates however, the clients compute stochastic gradients $\nabla_y f_i(\hat{\mathbf{x}}^{(s)}, \mathbf{y}_i^{(t,k)}; \xi_i^{(t,k)})$ keeping the $x$-component fixed at $\hat{\mathbf{x}}^{(s)}$ for $S$ communication rounds. This *trick*, originally proposed in Deng & Mahdavi (2021), gives the analytical benefit of a double-loop algorithm (which updates **y** several times before updating **x** once) while also updating **x** simultaneously.

**Theorem 2.** *Suppose the local loss functions $\{f_i\}$ satisfy Assumptions 1, 2, 3, 4, 7, 8, the **y** iterates are bounded, and the server selects $|\mathcal{C}^{(t)}| = P$ clients for all $t$. With appropriate client and server learning rates, $(\eta_x^c, \eta_y^c)$ and $(\gamma_x^s, \gamma_y^s)$ respectively (see Appendix C.2), the iterates of Fed-Norm-SGDA+ satisfy*

$$
\min_{t\in[T]} \mathbb{E}\big\|\nabla\widetilde{\Phi}_{1/2L_f}(\mathbf{x}^{(t)})\big\|^2 \leq \underbrace{\mathcal{O}\left(\left(\sigma_G^2 \frac{n-P}{n-1}\bar{\Delta}_{\widetilde{\Phi}}\frac{E_w}{PT}\sqrt{1+F_w}\right)^{1/4}\right)}_{\textit{Partial participation error}} + \underbrace{\mathcal{O}\left(\frac{C_w\sigma_L^2 + D(G_\mathbf{x}^2 + \sigma_G^2)}{\bar{\tau}^2 T^{3/4}}\right)}_{\textit{Local updates error}}
$$
$$
+ \underbrace{\mathcal{O}\left(\left(\frac{\bar{\Delta}_{\widetilde{\Phi}}\sigma_L^2 A_w}{\tau_{\textit{eff}}PT}\sqrt{1+F_w}\right)^{1/4} + \frac{\bar{\Delta}_{\widetilde{\Phi}}(1+F_w)}{T^{3/4}}\left(\frac{\tau_{\textit{eff}}P}{A_w + \tau_{\textit{eff}}\frac{n-P}{n-1}E_w}\right)^{1/4}\right)}_{\textit{Error with full synchronization}},
$$

(6)

*where $\Phi_{1/2L_f}$ is the Moreau envelope of $\Phi$, and $\bar{\Delta}_{\widetilde{\Phi}} \triangleq \widetilde{\Phi}_{1/2L_f}(\mathbf{x}_0) - \min_\mathbf{x} \widetilde{\Phi}_{1/2L_f}(\mathbf{x})$. The constants $A_w, C_w, D, E_w, \bar{\tau}, \tau_{\textit{eff}}$ are defined in Theorem 1, and $F_w \triangleq \frac{n(n-P)}{P(n-1)}\sum_{i=1}^n w_i^2$.*

See Appendix C for the proof. Theorem 2 states convergence for a surrogate objective $\widetilde{F}$. Next, we see convergence for the true objective $F$.

**Corollary 2.1** (Convergence in terms of $F$). *Given envelope functions $\Phi(\mathbf{x}) \triangleq \max_\mathbf{y} F(\mathbf{x},\mathbf{y})$, $\widetilde{\Phi}(\mathbf{x}) \triangleq \max_\mathbf{y} \widetilde{F}(\mathbf{x},\mathbf{y})$, under the conditions of Theorem 2,*

$$
\min_{t\in[T]} \big\|\nabla\Phi_{1/2L_f}(\mathbf{x}^{(t)})\big\|^2 \leq \epsilon'_{opt} + \frac{8L_f^2}{T}\sum_{t=0}^{T-1}\big\|\widetilde{\mathbf{x}}^{(t)} - \bar{\mathbf{x}}^{(t)}\big\|^2,
$$

*where $\Phi_{1/2L_f}$ is the Moreau envelope of $\Phi$, $\widetilde{\mathbf{x}}^{(t)} \triangleq \arg\min_{\mathbf{x}'}\{\widetilde{\Phi}(\mathbf{x}') + L_f\|\mathbf{x}' - \mathbf{x}^{(t)}\|^2\}$, $\bar{\mathbf{x}}^{(t)} \triangleq \arg\min_{\mathbf{x}'}\{\Phi(\mathbf{x}') + L_f\|\mathbf{x}' - \mathbf{x}^{(t)}\|^2\}$, for all $t$, $\epsilon'_{opt}$ is the error bound in (6).*

Similar to Corollary 1.1, if we replace $\{w_i\}$ with $\{p_i\}$ for all $i \in [n]$ in the server updates in Algorithm 1, then $\widetilde{F} \equiv F$, and $\widetilde{\mathbf{x}}^{(t)}$ and $\bar{\mathbf{x}}^{(t)}$ are identical for all $t$. Consequently, Theorem 2 gives us convergence in terms of the true objective $F$. For the rest of this subsection, we assume $w_i = p_i$ for all $i \in [n]$.

*Remark* 4. Some existing works do not require Assumption 8 for NC-C functions, and also improve the convergence rate. However, these methods either have a double-loop structure Rafique et al. (2021); Zhang et al. (2022), or work with deterministic problems Xu et al. (2020); Zhang et al. (2020). Proposing a single-loop method for stochastic NC-C problems with the same advantages is an open problem.

Again, in Table 2, we specialize the bound in (6) to SGDA+ based local updates. As in the last section

- Partial client participation is the *most significant* source of convergence error.
- Unequal client weights ($p_i \neq p_j$) can increase the stochastic gradient complexity, due to the presence of $n\|\mathbf{p}\|_\infty, n\|\mathbf{p}\|_2^2$ factors.

**Corollary 2.2** (Improved Communication Cost). *Suppose all the clients are weighted equally ($p_i = 1/n$ for all $i$), with each carrying out $\tau$ local steps of SGDA+. Further, assume that $\Phi_{1/2L_f}$ is bounded from below. Then, to reach $\mathbf{x}$ such that $\mathbb{E}\|\nabla\Phi_{1/2L_f}(\mathbf{x})\| \leq \epsilon$,*

- *Under full participation, the per-client gradient complexity of Fed-Norm-SGDA+ is $T\tau = \mathcal{O}(1/(n\epsilon^8))$. The number of communication rounds required is $T = \mathcal{O}(1/\epsilon^4)$.*
- *Under partial participation, the per-client gradient complexity of Fed-Norm-SGDA+ is $\mathcal{O}(1/(P\epsilon^8))$. In general, running multiple local updates does not yield any communication savings. However, in the special case when inter-client data heterogeneity $\sigma_G = 0$, the communication cost is $\mathcal{O}(1/\epsilon^4)$.*

In terms of communication requirements, we achieve massive savings (compared to $\mathcal{O}(1/\epsilon^7)$ in Sharma et al. (2022)). Our gradient complexity results achieve linear speedup in the number of participating clients. Further, as stated earlier, our work considers a more general FL setting with unequally weighted clients ($p_i \neq p_j$), running unequal local updates ($\tau_i \neq \tau_j$), using distinct local solvers ($\boldsymbol{a}_i \neq \boldsymbol{a}_j$).

**Extending the Results to Nonconvex-One-Point-Concave Functions.** One-point-convexity has been observed in SGD dynamics during neural network training Li & Yuan (2017); Kleinberg et al. (2018).

**Assumption 9** (One-point-Concavity in **y**)**.** The function $f$ is said to be one-point-concave in **y** if fixing $\mathbf{x} \in \mathbb{R}^{d_1}$, for all $\mathbf{y} \in \mathbb{R}^{d_2}$,

$$\langle \nabla_y f(\mathbf{x}, \mathbf{y}'), \mathbf{y} - \mathbf{y}^*(\mathbf{x}) \rangle \leq f(\mathbf{x}, \mathbf{y}) - f(\mathbf{x}, \mathbf{y}^*(\mathbf{x})),$$

where $\mathbf{y}^*(\mathbf{x}) \in \arg\max_{\mathbf{y}} f(\mathbf{x}, \mathbf{y})$.

It turns out, Theorem 2 holds for the more general class of nonconvex-one-point-concave (NC-1PC) functions. See Appendix C.4 for more details.

*Remark* 5. Suppose Assumptions 1, 3, 2, 4, 8 hold. Suppose for all **x**, all the $f_i$'s satisfy Assumption 9 at a common global minimizer $\mathbf{y}^*(\mathbf{x})$. Then, the bound in Theorem 2 holds.

*Remark* 6. Hence, we settle the conjecture posed in Sharma et al. (2022) that *linear speedup* can be achieved for NC-1PC functions. As an intermediate step in our proof, we show convergence of Local SGD for one-point-convex functions. This extends the convex result for Local SGD to a larger class of functions.

## 6 Experiments

In this section, we evaluate the empirical performance of the proposed algorithms. We consider a robust neural training problem Sinha et al. (2017); Nouiehed et al. (2019), and a fair classification problem Mohri et al. (2019); Deng et al. (2020). Due to space constraints, additional details of our experiments, and some additional results are included in Appendix D. Our experiments were run on a network of $n = 15$ clients, each equipped with an NVIDIA TitanX GPU. We model data heterogeneity across clients using Dirichlet distribution Wang et al. (2019) with parameter $\alpha$, $\text{Dir}_n(\alpha)$. Small $\alpha \Rightarrow$ higher heterogeneity across clients.

**Robust NN training.** We consider the following robust neural network (NN) training problem.

$$\min_{\mathbf{x}} \max_{\|\mathbf{y}\|^2 \leq 1} \sum_{j=1}^{N} \ell\left(h_{\mathbf{x}}(\mathbf{a}_i + \mathbf{y}), b_i\right), \tag{7}$$

where **x** denotes the NN parameters, $(a_i, b_i)$ denote the feature and label of the $i$-th sample, **y** denotes the adversarially added feature perturbation, and $h_{\mathbf{x}}$ denotes the NN output.

**Impact of system heterogeneity.** In Figure 3, we compare the effect of heterogeneous number of local updates across clients, on the performance of our proposed Fed-Norm-SGDA+. We compare with Local SGDA+ Deng & Mahdavi (2021), and Local SGDA+ with momentum Sharma et al. (2022). Clients sample the number of epochs they run locally via uniform sampling over the set $\{2\ldots, E\}$, i.e., $\tau_i \sim Unif[2:E]$. We observe that Fed-Norm-SGDA+ adapts well to system heterogeneity and outperforms both existing methods.

**Impact of partial participation and heterogeneity.** Next, we compare the impact of different levels of partial client participation on performance. We compare the full participation setting ($n = 15$) with $P = 5, 10$. Clients sample the number of epochs they run locally via $\tau_i \sim Unif[2, 5]$. We plot the results for two different values of the data heterogeneity parameter $\alpha = 0.1, 1.0$. As seen in all our theoretical results where partial participation was the most significant component of convergence error, smaller values of $P$ result in performance loss. Further, higher inter-client heterogeneity (modeled by smaller values of $\alpha$) results in worse performance. We further explore the impact of $\alpha$ on performance in Appendix D.

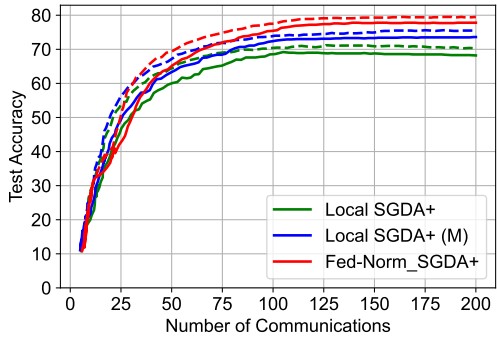

Figure 3: Comparison of the effect of heterogeneous number of local updates $\{\tau_i\}$ on the performance of Fed-Norm-SGDA+ (Algorithm 1), Local SGDA+, and Local SGDA+ with momentum, while solving (7) on CIFAR10 dataset, with VGG11 model. The solid (dashed) curves are for $E = 5$ ($E = 7$), and $\alpha = 0.1$.

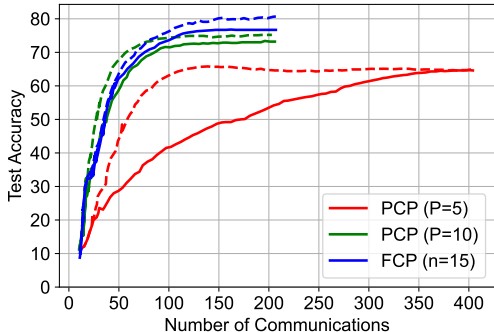

Figure 4: Comparison of the effects of partial client participation (PCP) on the performance of Fed-Norm-SGDA+, for the robust NN training problem on the CIFAR10 dataset, with the VGG11 model. The figure shows the robust test accuracy. The solid (dashed) curves are for $\alpha = 0.1$ ($\alpha = 1.0$).

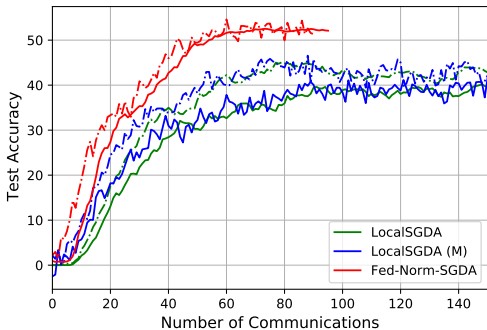

Figure 5: Comparison of Local SGDA, Local SGDA with momentum, and Fed-Norm-SGDA, for the fair classification task on the CIFAR10 dataset, with the VGG11 model. The solid (dashed) curves are for $E = 5$ ($E = 7$), $\alpha = 0.1$.

**Fair Classification.** We consider minimax formulation of the fair classification problem Mohri et al. (2019); Nouiehed et al. (2019).

$$\min_{\mathbf{x}} \max_{\mathbf{y} \in \Delta_C} \sum_{c=1}^{C} y_c F_c(\mathbf{x}) - \frac{\lambda}{2} \|\mathbf{y}\|^2, \tag{8}$$

where $\mathbf{x}$ denotes the parameters of the NN, $\{F_c\}_{c=1}^C$ denote the loss corresponding to class $c$, and $\Delta_C$ is the $C$-dimensional probability simplex. In Figure 5, we plot the worst distribution test accuracy achieved by Fed-Norm-SGDA, Local SGDA Deng & Mahdavi (2021) and Local SGDA with momentum Sharma et al. (2022). As in Figure 3, clients sample $\tau_i \sim Unif[2, E]$. We plot the test accuracy on the worst distribution in each case. Again, Fed-Norm-SGDA outperforms existing methods.

## 7    Conclusion

In this work, we considered nonconvex minimax problems in the federated setting, where in addition to inter-client data heterogeneity and partial client participation, there is system heterogeneity as well. Clients may run unequal number of local update steps, using different local solvers. In such settings, we observed that existing methods, such as Local SGDA, might converge to the stationary point of an objective quite different from the original intended objective. We showed that normalizing individual client contributions solves this problem. Using our generalized framework, we analyzed several classes of nonconvex minimax functions and significantly improved existing computation and communication complexity results. Potential future directions include analyzing federated systems with unpredictable client presence Yang et al. (2022b).

### Acknowledgments

This work was supported in part by NSF grants CCF 2045694, CNS-2112471, CPS-2111751, and ONR N00014-23-1- 2149. Jiarui Li helped with plotting figures for some experiments.

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

## Contents

# Appendix

The appendix are organized as follows. In Section A we mention some basic mathematical results and inequalities which are used throughout the paper. In Appendix B we prove the non-asymptotic convergence of Fed-Norm-SGDA (Algorithm 1) for smooth nonconvex-strongly-concave (and nonconvex-PŁ) functions, and derive gradient complexity and communication cost of the algorithm to achieve an $\epsilon$-stationary point. In Appendix C, we prove the non-asymptotic convergence of Fed-Norm-SGDA+ (Algorithm 1) for smooth nonconvex-concave and nonconvex-one-point-concave functions. Finally, in Appendix D we provide the details of the additional experiments we performed.

## A  Background

### A.1  Gradient Aggregation with Different Solvers at Clients

**Local SGDA.**  Suppose $\tau_i^{(t)} = \tau_{\text{eff}}^{(t)} = \tau$ for all $i \in [n]$, $t \in [T]$. Also, $a_i^{(t,k)} = 1$ for all $k \in [\tau], t$. Then, the local iterate updates in Algorithm 1-Fed-Norm-SGDA reduce to (the updates for Fed-Norm-SGDA+ are analogous)

$$\mathbf{x}_i^{(t,k+1)} = \mathbf{x}_i^{(t,k)} - \eta_x^c \nabla_x f_i(\mathbf{x}_i^{(t,k)}, \mathbf{y}_i^{(t,k)}; \xi_i^{(t,k)}),$$
$$\mathbf{y}_i^{(t,k+1)} = \mathbf{y}_i^{(t,k)} + \eta_y^c \nabla_y f_i(\mathbf{x}_i^{(t,k)}, \mathbf{y}_i^{(t,k)}; \xi_i^{(t,k)}),$$

for $k \in \{0, \ldots, \tau-1\}$ and the gradient aggregate vectors $(\mathbf{g}_{\mathbf{x},i}^{(t)}, \mathbf{g}_{\mathbf{y},i}^{(t)})$ are simply the average of individual gradients

$$\mathbf{g}_{\mathbf{x},i}^{(t)} = \frac{1}{\tau} \sum_{k=0}^{\tau-1} \nabla_x f_i(\mathbf{x}_i^{(t,k)}, \mathbf{y}_i^{(t,k)}; \xi_i^{(t,k)}), \quad \mathbf{g}_{\mathbf{y},i}^{(t)} = \frac{1}{\tau} \sum_{k=0}^{\tau-1} \nabla_y f_i(\mathbf{x}_i^{(t,k)}, \mathbf{y}_i^{(t,k)}; \xi_i^{(t,k)})$$

Note that these are precisely the iterates of LocalSGDA proposed in Deng & Mahdavi (2021); Sharma et al. (2022), with the only difference that in LocalSGDA, the clients communicate the iterates $\{\mathbf{x}_i^{(t,\tau)}, \mathbf{y}_i^{(t,\tau)}\}$ to the server, which averages them to compute $\{\mathbf{x}^{(t+1)}, \mathbf{y}^{(t+1)}\}$. While here, the clients communicate $\{\mathbf{g}_{\mathbf{x},i}^{(t)}, \mathbf{g}_{\mathbf{y},i}^{(t)}\}$. Also, in Fed-Norm-SGDA, the clients and server use separate learning rates, which results in tighter bounds on the local-updates error.

**With Momentum in Local Updates.**  Suppose each local client uses a momentum buffer with momentum scale $\rho$. Then, for $k \in \{0, \ldots, \tau_i^{(t)} - 1\}$

$$\mathbf{d}_{\mathbf{x},i}^{t,k+1} = \rho \mathbf{d}_{\mathbf{x},i}^{t,k} + \nabla_x f_i(\mathbf{x}_i^{(t,k)}, \mathbf{y}_i^{(t,k)}; \xi_i^{(t,k)}), \quad \mathbf{x}_i^{(t,k+1)} = \mathbf{x}_i^{(t,k)} - \eta_x^c \mathbf{d}_{\mathbf{x},i}^{t,k+1}$$
$$\mathbf{d}_{\mathbf{y},i}^{t,k+1} = \rho \mathbf{d}_{\mathbf{y},i}^{t,k} + \nabla_y f_i(\mathbf{x}_i^{(t,k)}, \mathbf{y}_i^{(t,k)}; \xi_i^{(t,k)}), \quad \mathbf{y}_i^{(t,k+1)} = \mathbf{y}_i^{(t,k)} + \eta_y^c \mathbf{d}_{\mathbf{y},i}^{t,k+1},$$

Simple calculations show that the coefficient of $\nabla_x f_i(\mathbf{x}_i^{(t,k)}, \mathbf{y}_i^{(t,k)}; \xi_i^{(t,k)})$ and $\nabla_y f_i(\mathbf{x}_i^{(t,k)}, \mathbf{y}_i^{(t,k)}; \xi_i^{(t,k)})$ in the gradient aggregate vectors $(\mathbf{g}_{\mathbf{x},i}^{(t)}, \mathbf{g}_{\mathbf{y},i}^{(t)})$ is

$$\sum_{j \geq k}^{\tau_i^{(t)}-1} = 1 + \rho + \cdots + \rho^{\tau_i^{(t)}-1-k} = \frac{1 - \rho^{\tau_i^{(t)}-k}}{1 - \rho}.$$

Therefore, the aggregation vector is $\bar{\mathbf{a}}_i^{(t)} = \frac{1}{1-\rho}[1 - \rho^{\tau_i^{(t)}}, 1 - \rho^{\tau_i^{(t)}-1}, \ldots, 1 - \rho]$, and

$$\|\bar{\mathbf{a}}_i^{(t)}\|_1 = \sum_{k=0}^{\tau_i^{(t)}-1} \frac{1 - \rho^{\tau_i^{(t)}-k}}{1 - \rho} = \frac{1}{1 - \rho} \left[ \tau_i^{(t)} - \frac{\rho(1 - \rho^{\tau_i^{(t)}})}{1 - \rho} \right].$$

### A.2 Auxiliary Results

*Remark* 7 (Impact of heterogeneity $\sigma_G$ even with $\tau = 1$). Consider two simple minimization problems:

$$\textbf{(P1):} \quad \min_{\mathbf{x}} \frac{1}{n} \sum_{i=1}^{n} f_i(\mathbf{x}) \qquad \text{and} \qquad \textbf{(P2):} \quad \min_{\mathbf{x}} f(x).$$

**(P1)** is a simple distributed minimization problem, with $n$ clients, which we solve using synchronous distributed SGD. At iteration $t$, each client $i$ computes stochastic gradient $\nabla f_i(\mathbf{x}^{(t)}; \xi_i^{(t)})$, and sends it to the server, which averages these, and takes a step in the direction $\frac{1}{n} \sum_{i=1}^{n} \nabla f_i(\mathbf{x}^{(t)}; \xi_i^{(t)})$. On the other hand, **(P1)** is a centralized minimization problem, where at each iteration $t$, the agent computes a stochastic gradient estimator with batch-size $n$, $\frac{1}{n} \sum_{i=1}^{n} \nabla f(\mathbf{x}^{(t)}; \xi_i^{(t)})$. We compare the variance of the two global gradient estimators as follows.

$$
\begin{array}{ccc}
\textbf{(P1)} & & \textbf{(P2)} \\[4pt]
\mathbb{E}\left\| \frac{1}{n} \sum_{i=1}^{n} \nabla f_i(\mathbf{x}^{(t)}; \xi_i^{(t)}) - \nabla f(\mathbf{x}^{(t)}) \right\|^2 & & \mathbb{E}\left\| \frac{1}{n} \sum_{i=1}^{n} \nabla f(\mathbf{x}^{(t)}; \xi_i^{(t)}) - \nabla f(\mathbf{x}^{(t)}) \right\|^2 \\[8pt]
\leq \frac{1}{n^2} \sum_{i=1}^{n} \left[ \sigma_L^2 + \beta_L^2 \mathbb{E}\|\nabla f_i(\mathbf{x}^{(t)})\|^2 \right] & \text{vs} & = \frac{1}{n^2} \sum_{i=1}^{n} \mathbb{E}\left\| \nabla f(\mathbf{x}^{(t)}; \xi_i^{(t)}) - \nabla f(\mathbf{x}^{(t)}) \right\|^2 \\[8pt]
\leq \frac{\sigma_L^2}{n} + \frac{\beta_L^2}{n} \left[ \beta_H^2 \mathbb{E}\|\nabla f(\mathbf{x}^{(t)})\|^2 + \sigma_G^2 \right]. & & \leq \frac{\sigma_L^2}{n} + \frac{\beta_L^2}{n} \mathbb{E}\|\nabla f(\mathbf{x}^{(t)})\|^2.
\end{array}
$$

Since almost all the existing works consider the local variance bound (Assumption 3) with $\beta_L = 0$, the global gradient estimator in both synchronous distributed SGD **(P1)** and single-agent minibatch SGD **(P2)** have the same $\frac{\sigma_L^2}{n}$ variance bound. Therefore, in most existing federated works on minimization Wang et al. (2020); Yang et al. (2021) and minimax problems Sharma et al. (2022), the full synchronization error only depends on the local variance $\sigma_L^2$. However, as seen above, for $\beta_L > 0$, this *apparent equivalence* breaks down. Koloskova et al. (2020), which considers similar local variance assumption as ours for minimization problems, also show similar dependence on heterogeneity $\sigma_G$.

**Lemma A.1** (Young's inequality). *Given two same-dimensional vectors* $\mathbf{u}, \mathbf{v} \in \mathbb{R}^d$, *the Euclidean inner product can be bounded as follows:*

$$\langle \mathbf{u}, \mathbf{v} \rangle \leq \frac{\|\mathbf{u}\|^2}{2\gamma} + \frac{\gamma \|\mathbf{v}\|^2}{2}$$

*for every constant* $\gamma > 0$.

**Lemma A.2** (Strong Concavity). *A function* $g : \mathcal{X} \times \mathcal{Y}$ *is strongly concave in* $\mathbf{y}$, *if there exists a constant* $\mu > 0$, *such that for all* $\mathbf{x} \in \mathcal{X}$, *and for all* $\mathbf{y}, \mathbf{y}' \in \mathcal{Y}$, *the following inequality holds.*

$$g(\mathbf{x}, \mathbf{y}) \leq g(\mathbf{x}, \mathbf{y}') + \langle \nabla_y g(\mathbf{x}, \mathbf{y}'), \mathbf{y}' - \mathbf{y} \rangle - \frac{\mu}{2} \|\mathbf{y} - \mathbf{y}'\|^2.$$

**Lemma A.3** (Jensen's inequality). *Given a convex function* $f$ *and a random variable* $X$, *the following holds.*

$$f(\mathbb{E}[X]) \leq \mathbb{E}[f(X)].$$

**Lemma A.4** (Sum of squares). *For a positive integer* $K$, *and a set of vectors* $\mathbf{x}_1, \ldots, \mathbf{x}_K$, *the following holds:*

$$\left\| \sum_{k=1}^{K} \mathbf{x}_k \right\|^2 \leq K \sum_{k=1}^{K} \|\mathbf{x}_k\|^2.$$

**Lemma A.5** (Quadratic growth condition Karimi et al. (2016)). *If function* $g$ *satisfies Assumptions 1, 5, then for all* $\mathbf{x}$, *the following conditions holds*

$$g(\mathbf{x}) - \min_{\mathbf{z}} g(\mathbf{z}) \geq \frac{\mu}{2} \|\mathbf{x}_p - \mathbf{x}\|^2,$$

$$\|\nabla g(\mathbf{x})\|^2 \geq 2\mu \left( g(\mathbf{x}) - \min_{\mathbf{z}} g(\mathbf{z}) \right).$$

**Lemma A.6.** *For L-smooth, convex function g, the following inequality holds*

$$\mathbb{E}\left\|\nabla g(\mathbf{y}) - \nabla g(\mathbf{x})\right\|^2 \leq 2L\left[g(\mathbf{y}) - g(\mathbf{x}) - \nabla g(\mathbf{x})^\top(\mathbf{y} - \mathbf{x})\right]. \tag{9}$$

**Lemma A.7** (Proposition 6 in Cho & Yun (2022))**.** *For L-smooth function g which is bounded below by $g^*$, the following inequality holds for all $\mathbf{x}$*

$$\mathbb{E}\left\|\nabla g(\mathbf{x})\right\|^2 \leq 2L\left[g(\mathbf{x}) - g^*\right]. \tag{10}$$

# B Convergence of Fed-Norm-SGDA for Nonconvex-Strongly-Concave Functions (Theorem 1)

We organize this section as follows. First, in Appendix B.1 we present some intermediate results, which we use to prove the main theorem. Next, in Appendix B.2, we present the proof of Theorem 1, which is followed by the proofs of the intermediate results in Appendix B.3. Appendix B.4 contains some auxiliary results. Finally, in Appendix B.5 we discuss the convergence result for nonconvex-PL functions.

The problem we solve is

$$\min_{\mathbf{x}}\max_{\mathbf{y}}\left\{\widetilde{F}(\mathbf{x},\mathbf{y}) \triangleq \sum_{i=1}^{n} w_i f_i(\mathbf{x},\mathbf{y})\right\}.$$

We define $\widetilde{\Phi}(\mathbf{x}) \triangleq \max_{\mathbf{y}}\widetilde{F}(\mathbf{x},\mathbf{y})$ and $\widetilde{\mathbf{y}}^*(\mathbf{x}) \in \arg\max_{\mathbf{y}}\widetilde{F}(\mathbf{x},\mathbf{y})$. Since $\widetilde{F}(\mathbf{x},\cdot)$ is $\mu$-strongly concave, $\widetilde{\mathbf{y}}^*(\mathbf{x})$ is unique. In Fed-Norm-SGDA (Algorithm 1), the client updates are given by

$$\begin{aligned}
\mathbf{x}_i^{(t,k)} &= \mathbf{x}^{(t)} - \eta_x^c \sum_{j=0}^{k-1} a_i^{(j)}(k)\nabla_x f_i(\mathbf{x}_i^{(t,j)},\mathbf{y}_i^{(t,j)};\xi_i^{(t,j)}), \\
\mathbf{y}_i^{(t,k)} &= \mathbf{y}^{(t)} + \eta_y^c \sum_{j=0}^{k-1} a_i^{(j)}(k)\nabla_y f_i(\mathbf{x}_i^{(t,j)},\mathbf{y}_i^{(t,j)};\xi_i^{(t,j)}),
\end{aligned} \tag{11}$$

where $1 \leq k \leq \tau_i$. These client updates are then aggregated to compute $\{\mathbf{g}_{\mathbf{x},i}^{(t)}, \mathbf{g}_{\mathbf{y},i}^{(t)}\}$

$$\begin{aligned}
\mathbf{g}_{\mathbf{x},i}^{(t)} &= \frac{1}{\|\boldsymbol{a}_i\|}\sum_{k=0}^{\tau_i-1} a_i^{(k)}(\tau_i)\nabla_x f_i\left(\mathbf{x}_i^{(t,k)},\mathbf{y}_i^{(t,k)};\xi_i^{(t,k)}\right); \quad \mathbf{h}_{\mathbf{x},i}^{(t)} = \frac{1}{\|\boldsymbol{a}_i\|}\sum_{k=0}^{\tau_i-1} a_i^{(k)}(\tau_i)\nabla_x f_i\left(\mathbf{x}_i^{(t,k)},\mathbf{y}_i^{(t,k)}\right) \\
\mathbf{g}_{\mathbf{y},i}^{(t)} &= \frac{1}{\|\boldsymbol{a}_i\|}\sum_{k=0}^{\tau_i-1} a_i^{(k)}(\tau_i)\nabla_y f_i\left(\mathbf{x}_i^{(t,k)},\mathbf{y}_i^{(t,k)};\xi_i^{(t,k)}\right); \quad \mathbf{h}_{\mathbf{y},i}^{(t)} = \frac{1}{\|\boldsymbol{a}_i\|}\sum_{k=0}^{\tau_i-1} a_i^{(k)}(\tau_i)\nabla_y f_i\left(\mathbf{x}_i^{(t,k)},\mathbf{y}_i^{(t,k)}\right).
\end{aligned}$$

*Remark* 8. Note that we have made explicit, the dependence on $k$ in $a_i^{(j)}(k)$ above. This was omitted in the main paper to avoid tedious notation. However, for some local optimizers, such as local momentum at the clients (Appendix A.1), the coefficients $a_i^{(j)}(k)$ change with $k$. We assume in our subsequent analysis that $a_i^{(j)}(k) \leq \alpha$ for all $j \in \{0,1,\ldots,k-1\}$ and for all $k \in \{1,2,\ldots,\tau_i\}$. We also use the notation $\|\boldsymbol{a}_i\| \triangleq \|\boldsymbol{a}_i(\tau_i)\|$.

At iteration $t$, the server samples $|\mathcal{C}^{(t)}|$ clients *without* replacement **(WOR)** uniformly at random. While aggregating at the server, client $i$ update is weighed by $\tilde{w}_i = w_i n/|\mathcal{C}^{(t)}|$. The aggregates $(\mathbf{g}_{\mathbf{x}}^{(t)},\mathbf{g}_{\mathbf{y}}^{(t)})$ computed at the server are of the form

$$\begin{aligned}
\mathbf{g}_{\mathbf{x}}^{(t)} &= \sum_{i\in\mathcal{C}^{(t)}}\tilde{w}_i\mathbf{g}_{\mathbf{x},i}^{(t)}, \quad \text{such that} \quad \mathbb{E}_{\mathcal{C}^{(t)}}[\mathbf{g}_{\mathbf{x}}^{(t)}] = \mathbb{E}_{\mathcal{C}^{(t)}}\left[\sum_{i=1}^{n}\mathbb{I}(i\in\mathcal{C}^{(t)})\tilde{w}_i\mathbf{g}_{\mathbf{x},i}^{(t)}\right] = \sum_{i=1}^{n}w_i\mathbf{g}_{\mathbf{x},i}^{(t)} \\
\mathbf{g}_{\mathbf{y}}^{(t)} &= \sum_{i\in\mathcal{C}^{(t)}}\tilde{w}_i\mathbf{g}_{\mathbf{y},i}^{(t)}, \quad \text{such that} \quad \mathbb{E}_{\mathcal{C}^{(t)}}[\mathbf{g}_{\mathbf{y}}^{(t)}] = \mathbb{E}_{\mathcal{C}^{(t)}}\left[\sum_{i=1}^{n}\mathbb{I}(i\in\mathcal{C}^{(t)})\tilde{w}_i\mathbf{g}_{\mathbf{y},i}^{(t)}\right] = \sum_{i=1}^{n}w_i\mathbf{g}_{\mathbf{y},i}^{(t)}
\end{aligned} \tag{12}$$

For simplicity of analysis, unless stated otherwise, we assume that $|\mathcal{C}^{(t)}| = P$ for all $t$. Finally, server updates the $\mathbf{x}, \mathbf{y}$ variables as

$$\mathbf{x}^{(t+1)} = \mathbf{x}^{(t)} - \tau_{\text{eff}} \gamma_x^s \mathbf{g}_{\mathbf{x}}^{(t)}, \qquad \mathbf{y}^{(t+1)} = \mathbf{y}^{(t)} + \tau_{\text{eff}} \gamma_y^s \mathbf{g}_{\mathbf{y}}^{(t)}.$$

We define by $\mathcal{F}(t')$ the $\sigma$-algebra generated by $\{\{\mathbf{x}_i^{(t,k)}, \mathbf{y}_i^{(t,k)}\}_{i,k}\}_{t=0}^{t'-1}$. Throughout, we denote the conditional expectation $\mathbb{E}[\cdot | \mathcal{F}(t)]$ by the shorthand $\mathbb{E}_t[\cdot]$.

## B.1 Intermediate Lemmas

We begin with the following result from Nouiehed et al. (2019) about the smoothness of $\widetilde{\Phi}(\cdot)$.

**Lemma B.1.** *If a function $f(\cdot, \cdot)$ satisfies Assumptions 1, 5 ($L_f$-smoothness and $\mu$-strong concavity in $\mathbf{y}$), then $\phi(\cdot) \triangleq \max_{\mathbf{y}} f(\cdot, \mathbf{y})$ is $L_\Phi$-smooth with $L_\Phi = \kappa L_f / 2 + L_f$, where $\kappa = L_f / \mu$ is the condition number.*

**Lemma B.2.** *Suppose the local client loss functions $\{f_i\}$ satisfy Assumptions 1, 4, and the stochastic oracles for the local functions satisfy Assumption 3. Suppose the server selects $P$ clients in each round. Then the iterates generated by Fed-Norm-SGDA (Algorithm 1) satisfy*

$$
\mathbb{E}_t \left\| \mathbf{g}_{\mathbf{x}}^{(t)} \right\|^2 = \mathbb{E}_t \left\| \sum_{i \in \mathcal{C}^{(t)}} \tilde{w}_i \mathbf{g}_{\mathbf{x},i}^{(t)} \right\|^2
$$

$$
\leq \frac{n}{P} \left( \frac{P-1}{n-1} \right) \mathbb{E}_t \left\| \sum_{i=1}^n w_i \mathbf{h}_{\mathbf{x},i}^{(t)} \right\|^2 + \frac{n}{P} \sum_{i=1}^n \frac{w_i^2}{\|\boldsymbol{a}_i\|_1^2} \sum_{k=0}^{\tau_i - 1} [a_i^{(k)}(\tau_i)]^2 \left[ \sigma_L^2 + \beta_L^2 \mathbb{E}_t \left\| \nabla_x f_i(\mathbf{x}_i^{(t,k)}, \mathbf{y}_i^{(t,k)}) \right\|^2 \right] \tag{13}
$$

$$
+ \frac{n(n-P)}{n-1} \left[ \frac{2 L_f^2}{P} \sum_{i=1}^n \frac{w_i^2}{\|\boldsymbol{a}_i\|_1} \sum_{k=0}^{\tau_i - 1} a_i^{(k)}(\tau_i) \Delta_{\mathbf{x},\mathbf{y}}^{(t,k)}(i) + (\max_i w_i) \frac{2}{P} \left( \beta_G^2 \left\| \nabla_x \widetilde{F}(\mathbf{x}^{(t)}, \mathbf{y}^{(t)}) \right\|^2 + \sigma_G^2 \right) \right],
$$

*where, $\Delta_{\mathbf{x},\mathbf{y}}^{(t,k)}(i) \triangleq \mathbb{E}_t \left[ \|\mathbf{x}_i^{(t,k)} - \mathbf{x}^{(t)}\|^2 + \|\mathbf{y}_i^{(t,k)} - \mathbf{y}^{(t)}\|^2 \right]$ is the iterate drift for client $i$, at local iteration $k$ in the $t$-th communication round.*

**Lemma B.3.** *Suppose the local client loss functions $\{f_i\}$ satisfy Assumptions 1, 4, 5, and the stochastic oracles for the local functions satisfy Assumption 3. Also, the server learning rate $\gamma_x^s$ satisfies $64 \tau_{eff} \gamma_x^s L_\Phi \beta_L^2 \beta_G^2 \frac{n}{P} (\max_i w_i \|\boldsymbol{a}_i\|_2^2 / \|\boldsymbol{a}_i\|_1^2) \leq 1$, $8 \tau_{eff} \gamma_x^s L_\Phi (\max_i w_i) \frac{n}{P} \left( \frac{n-P}{n-1} \right) \max\{8\beta_G^2, 1\} \leq 1$, and $8 \tau_{eff} \gamma_x^s L_\Phi \beta_L^2 \frac{n}{P} (\max_{i,k} w_i a_i^{(k)}(\tau_i) / \|\boldsymbol{a}_i\|_1) \leq 1$. Then the iterates generated by Algorithm 1 satisfy*

$$
\mathbb{E}_t \left[ \widetilde{\Phi}(\mathbf{x}^{(t+1)}) - \widetilde{\Phi}(\mathbf{x}^{(t)}) \right] \leq -\frac{7 \tau_{eff} \gamma_x^s}{16} \left\| \nabla \widetilde{\Phi}(\mathbf{x}^{(t)}) \right\|^2 - \frac{\tau_{eff} \gamma_x^s}{2} \left( 1 - \frac{n(P-1)}{P(n-1)} \tau_{eff} \gamma_x^s L_\Phi \right) \mathbb{E}_t \left\| \sum_{i=1}^n w_i \mathbf{h}_{\mathbf{x},i}^{(t)} \right\|^2
$$

$$
+ \frac{5}{4} \tau_{eff} \gamma_x^s L_f^2 \sum_{i=1}^n \frac{w_i}{\|\boldsymbol{a}_i\|_1} \sum_{k=0}^{\tau_i - 1} a_i^{(k)}(\tau_i) \Delta_{\mathbf{x},\mathbf{y}}^{(t,k)}(i) + \frac{9 \tau_{eff} \gamma_x^s L_f^2}{4\mu} \left[ \widetilde{\Phi}(\mathbf{x}^{(t)}) - \widetilde{F}(\mathbf{x}^{(t)}, \mathbf{y}^{(t)}) \right] \tag{14}
$$

$$
+ \frac{\tau_{eff}^2 [\gamma_x^s]^2 L_\Phi}{2} \frac{n}{P} \left[ \sigma_L^2 \sum_{i=1}^n \frac{w_i^2 \|\boldsymbol{a}_i\|_2^2}{\|\boldsymbol{a}_i\|_1^2} + \sigma_G^2 \left( 2(\max_i w_i) \frac{n-P}{n-1} + 2\beta_L^2 \max_i \frac{w_i \|\boldsymbol{a}_i\|_2^2}{\|\boldsymbol{a}_i\|_1^2} \right) \right].
$$

*Remark.* The bound in Equation (14) looks very similar to the corresponding one-step decay bound for simple smooth minimization problems. The major difference is the presence of $\left[ \widetilde{\Phi}(\mathbf{x}^{(t)}) - \widetilde{F}(\mathbf{x}^{(t)}, \mathbf{y}^{(t)}) \right]$, which quantifies the inaccuracy of $\mathbf{y}^{(t)}$ in solving the *max* problem $\max_{\mathbf{y}} \widetilde{F}(\mathbf{x}^{(t)}, \mathbf{y})$. The term $\sum_{i=1}^n \frac{w_i}{\|\boldsymbol{a}_i\|_1} \sum_{k=0}^{\tau_i - 1} a_i^{(k)}(\tau_i) \Delta_{\mathbf{x},\mathbf{y}}^{(t,k)}(i)$ is the client drift and is bounded in Lemma B.4 below.

**Lemma B.4.** *Suppose the local loss functions $\{f_i\}$ satisfy Assumptions 1, 4, 5, and the stochastic oracles for the local functions satisfy Assumption 3. Further, in Algorithm 1, we choose learning rates $\eta_x^c, \eta_y^c$ such that $\max\{\eta_x^c, \eta_y^c\} \leq \frac{1}{2 L_f (\max_i \|\boldsymbol{a}_i\|_1) \sqrt{2(1+\beta_L^2)}}$. Then, the iterates $\{\mathbf{x}_i^{(t)}, \mathbf{y}_i^{(t)}\}$ generated by Fed-Norm-SGDA*

*(Algorithm 1) satisfy*

$$L_f^2 \sum_{i=1}^n \frac{w_i}{\|\boldsymbol{a}_i\|_1} \sum_{k=0}^{\tau_i-1} a_i^{(k)}(\tau_i) \Delta_{\mathbf{x},\mathbf{y}}^{(t,k)}(i) \leq 2\left([\eta_x^c]^2 + [\eta_y^c]^2\right) L_f^2 \sigma_L^2 \sum_{i=1}^n w_i \|\boldsymbol{a}_{i,-1}\|_2^2 + 4L_f^2 M_{\mathbf{a}_{-1}} \left([\eta_x^c]^2 + [\eta_y^c]^2\right) \sigma_G^2$$
$$+ 8L_f^2 M_{\mathbf{a}_{-1}} \beta_G^2 [\eta_x^c]^2 \left\|\nabla\widetilde{\Phi}(\mathbf{x}^{(t)})\right\|^2 + 8L_f^3 M_{\mathbf{a}_{-1}} \beta_G^2 \left(2\kappa[\eta_x^c]^2 + [\eta_y^c]^2\right) \left[\widetilde{\Phi}(\mathbf{x}^{(t)}) - \widetilde{F}(\mathbf{x}^{(t)}, \mathbf{y}^{(t)})\right],$$

*where* $M_{\mathbf{a}_{-1}} \triangleq \max_i \left(\|\boldsymbol{a}_{i,-1}\|_1^2 + \beta_L^2 \|\boldsymbol{a}_{i,-1}\|_2^2\right)$.

**Lemma B.5.** *Suppose the local loss functions* $\{f_i\}$ *satisfy Assumptions 1, 4, 5, and the stochastic oracles for the local functions satisfy Assumption 3. The server learning rates* $\gamma_x^s, \gamma_y^s$ *satisfy the following conditions:*

$$\tau_{\text{eff}} \gamma_y^s \kappa L_f \beta_G^2 \frac{n}{P} \max\left\{\beta_L^2 \max_i \frac{w_i \|\boldsymbol{a}_i\|_2^2}{\|\boldsymbol{a}_i\|_1^2}, \frac{n-P}{n-1} \max_i w_i\right\} \leq \frac{1}{64}, \gamma_x^s \leq \frac{\gamma_y^s}{81\kappa^2},$$

$$8\tau_{\text{eff}} L_f \gamma_y^s \frac{n}{P} \max\left\{\frac{n-P}{n-1} \max_i w_i, \beta_L^2 \max_{i,k} \frac{w_i a_i^{(k)}(\tau_i)}{\|\boldsymbol{a}_i\|_1}\right\} \leq 1$$

*The client learning rates* $\eta_x^c, \eta_y^c$ *satisfy* $\eta_y^c L_f \beta_G \leq \frac{1}{16\sqrt{\kappa M_{\mathbf{a}_{-1}}}}$ *and* $\eta_x^c$: $\eta_x^c L_f \beta_G \leq \frac{1}{64\kappa\sqrt{M_{\mathbf{a}_{-1}}}}$, *respectively. Then the iterates generated by Fed-Norm-SGDA (Algorithm 1) satisfy*

$$\frac{1}{T} \sum_{t=0}^{T-1} \mathbb{E}\left[\widetilde{\Phi}(\mathbf{x}^{(t)}) - \widetilde{F}(\mathbf{x}^{(t)}, \mathbf{y}^{(t)})\right]$$
$$\leq \frac{4\left[\widetilde{\Phi}(\mathbf{x}^{(0)}) - \widetilde{F}(\mathbf{x}^{(0)}, \mathbf{y}^{(0)})\right]}{\tau_{\text{eff}} \gamma_y^s \mu T} + \frac{1}{12\mu\kappa^2} \frac{1}{T} \sum_{t=0}^{T-1} \mathbb{E}\left\|\nabla\widetilde{\Phi}(\mathbf{x}^{(t)})\right\|^2 + \frac{4\tau_{\text{eff}}[\gamma_x^s]^2 L_\Phi}{\gamma_y^s \mu} \frac{n(P-1)}{P(n-1)} \mathbb{E}\left\|\sum_{i=1}^n w_i \mathbf{h}_{\mathbf{x},i}^{(t)}\right\|^2$$
$$+ 8\tau_{\text{eff}} \gamma_y^s \kappa \frac{n}{P}\left[\sigma_L^2 \sum_{i=1}^n \frac{w_i^2 \|\boldsymbol{a}_i\|_2^2}{\|\boldsymbol{a}_i\|_1^2} + 2\sigma_G^2 \left(\frac{n-P}{n-1} \max_i w_i + \beta_L^2 \max_i \frac{w_i \|\boldsymbol{a}_i\|_2^2}{\|\boldsymbol{a}_i\|_1^2}\right)\right]$$
$$+ 8\kappa L_f \left([\eta_x^c]^2 + [\eta_y^c]^2\right)\left[\sigma_L^2 \sum_{i=1}^n w_i \|\boldsymbol{a}_{i,-1}\|_2^2 + 2\sigma_G^2 M_{\mathbf{a}_{-1}}\right]. \tag{15}$$

*Remark* 9. The proof of Lemma B.5 differs from similar results in the existing literature Sharma et al. (2022); Yang et al. (2022a). As in these works, if all the clients are running the same number of local steps ($\tau_i = \tau$, for all $i$), we can define virtual sequences of average iterates $\mathbf{x}^{(t,k)} = \frac{1}{P} \sum_{i \in \mathcal{C}^{(t)}} \mathbf{x}_i^{(t,k)}$, $\mathbf{y}^{(t,k)} = \frac{1}{P} \sum_{i \in \mathcal{C}^{(t)}} \mathbf{y}_i^{(t,k)}$, for all $k \in [0, \tau-1], t$. Define $\mathcal{F}'(t,k)$ as the $\sigma$-algebra generated by

$$\mathcal{F}'(t,k) \triangleq \sigma\left\{\{\{\mathbf{x}_i^{(s,k)}, \mathbf{x}_i^{(s,k)}\}_{i,k}\}_{s=0}^{t-1} \bigcup \{\{\mathbf{x}_i^{(t,j)}, \mathbf{x}_i^{(t,j)}\}_i\}_{j=0}^{k-1}\right\}.$$

Since, conditioned on $\mathcal{F}'(t,k)$, $\mathbf{x}^{t,k+1} \perp \{\nabla_y f_i(\mathbf{x}_i^{(t,k)}, \mathbf{y}_i^{(t,k)}; \xi_i^{(t,k)})\}_{i=1}^n$, using $\{\mathbf{x}^{(t,k)}, \mathbf{y}^{(t,k)}\}$ considerably simplifies the analysis. However, with $\tau_i \neq \tau_j$, the virtual sequences $\{\mathbf{x}^{(t,k)}, \mathbf{y}^{(t,k)}\}$ can no longer be defined for all $k$. Hence, we need an alternate proof strategy.

## B.2 Proof of Theorem 1

For the sake of completeness, we first state the full statement of Theorem 1 here.

**Theorem.** *Suppose the local loss functions* $\{f_i\}_i$ *satisfy Assumptions 1, 3, 4, 5. Suppose the server selects clients using without-replacement sampling scheme (WOR). Also, the server learning rates* $\gamma_x^s, \gamma_y^s$ *and the client learning rates* $\eta_x^c, \eta_y^c$ *satisfy the conditions specified in Lemma B.5. Then the iterates generated by*

*Fed-Norm-SGDA (Algorithm 1) satisfy*

$$\min_{t\in[0:T-1]}\mathbb{E}\left\|\nabla\widetilde{\Phi}(\mathbf{x}^{(t)})\right\|^2 \leq \frac{1}{T}\sum_{t=0}^{T-1}\mathbb{E}\left\|\nabla\widetilde{\Phi}(\mathbf{x}^{(t)})\right\|^2 \leq \underbrace{\mathcal{O}\left(\kappa^2\left[\frac{\Delta_{\widetilde{\Phi}}}{\tau_{eff}\gamma_y^s T} + \frac{\gamma_y^s L_f}{P}\left(A_w\sigma_L^2 + B_w\beta_L^2\sigma_G^2\right)\right]\right)}_{\textit{Error with full synchronization}}$$

$$+\underbrace{\mathcal{O}\left(\kappa^2\left([\eta_x^c]^2+[\eta_y^c]^2\right)L_f^2\left[C_w\sigma_L^2 + D\sigma_G^2\right]\right)}_{\textit{Error due to local updates}} + \underbrace{\mathcal{O}\left(\kappa^2\frac{n-P}{n-1}\frac{\gamma_y^s L_f E_w\tau_{eff}\sigma_G^2}{P}\right)}_{\textit{Partial Participation Error}},$$

*where $\kappa = L_f/\mu$ is the condition number, $\widetilde{\Phi}(\mathbf{x}) \triangleq \max_{\mathbf{y}}\widetilde{F}(\mathbf{x},\mathbf{y})$ is the envelope function, $\Delta_{\widetilde{\Phi}} \triangleq \widetilde{\Phi}(\mathbf{x}^{(0)}) - \min_{\mathbf{x}}\widetilde{\Phi}(\mathbf{x})$, $A_w \triangleq n\tau_{eff}\sum_{i=1}^n \frac{w_i^2\|\boldsymbol{a}_i\|_2^2}{\|\boldsymbol{a}_i\|_1^2}$, $B_w \triangleq n\tau_{eff}\max_i\frac{w_i\|\boldsymbol{a}_i\|_2^2}{\|\boldsymbol{a}_i\|_1^2}$, $C_w \triangleq \sum_{i=1}^n w_i(\|\boldsymbol{a}_i\|_2^2 - [\alpha_i^{(t,\tau_i-1)}]^2)$, $D \triangleq \max_i(\beta_L^2\|\boldsymbol{a}_{i,-1}\|_2^2 + \|\boldsymbol{a}_{i,-1}\|_1^2)$, where $\boldsymbol{a}_{i,-1} \triangleq [a_i^{(0)}, a_i^{(1)}, \ldots, a_i^{(\tau_i-2)}]^\top$ for all $i$ and $E_w \triangleq n\max_i w_i$.*

*Using $\gamma_y^s = \Theta\left(\sqrt{\frac{P}{\tau_{eff}L_f T\left[\Delta_{\widetilde{\Phi}}+A_w\sigma_L^2+\left(B_w\beta_L^2+\frac{n-P}{n-1}E_w\tau_{eff}\right)\sigma_G^2\right]}}\right)$ and $\eta_x^c \leq \eta_y^c = \Theta(\frac{1}{L_f\bar{\tau}\sqrt{T}})$, where $\bar{\tau} = \frac{1}{n}\sum_{i=1}^n \tau_i$ in the bounds above, we get*

$$\min_{t\in[T]}\mathbb{E}\left\|\nabla\widetilde{\Phi}(\mathbf{x}^{(t)})\right\|^2 \leq \underbrace{\mathcal{O}\left(\kappa^2\sqrt{\frac{\Delta_{\widetilde{\Phi}}+A_w\sigma_L^2+B_w\beta_L^2\sigma_G^2}{P\tau_{eff}T}}\right)}_{\textit{Error with full synchronization}} + \underbrace{\mathcal{O}\left(\kappa^2\sqrt{\frac{n-P}{n-1}\cdot\frac{E_w\sigma_G^2}{PT}}\right)}_{\substack{\textit{Partial participation}\\\textit{error}}} + \underbrace{\mathcal{O}\left(\kappa^2\frac{C_w\sigma_L^2+D\sigma_G^2}{\bar{\tau}^2 T}\right)}_{\textit{Local updates error}}.$$

*Proof.* Using Lemma B.3, and substituting in the bound on iterates' drift from Lemma B.4, we can bound

$$\mathbb{E}_t\left[\widetilde{\Phi}(\mathbf{x}^{(t+1)}) - \widetilde{\Phi}(\mathbf{x}^{(t)})\right] \leq -\frac{7\tau_{\text{eff}}\gamma_x^s}{16}\left\|\nabla\widetilde{\Phi}(\mathbf{x}^{(t)})\right\|^2 - \frac{\tau_{\text{eff}}\gamma_x^s}{2}\left(1 - \frac{n(P-1)}{P(n-1)}\tau_{\text{eff}}\gamma_x^s L_\Phi\right)\mathbb{E}_t\left\|\sum_{i=1}^n w_i\mathbf{h}_{\mathbf{x},i}^{(t)}\right\|^2$$

$$+\frac{9\tau_{\text{eff}}\gamma_x^s L_f^2}{4\mu}\left[\widetilde{\Phi}(\mathbf{x}^{(t)}) - \widetilde{F}(\mathbf{x}^{(t)},\mathbf{y}^{(t)})\right]$$

$$+\frac{\tau_{\text{eff}}^2[\gamma_x^s]^2 L_\Phi}{2}\frac{n}{P}\left[\sigma_L^2\sum_{i=1}^n\frac{w_i^2\|\boldsymbol{a}_i\|_2^2}{\|\boldsymbol{a}_i\|_1^2} + \sigma_G^2\left(2(\max_i w_i)\frac{n-P}{n-1} + 2\beta_L^2\max_i\frac{w_i\|\boldsymbol{a}_i\|_2^2}{\|\boldsymbol{a}_i\|_1^2}\right)\right]$$

$$+\frac{5}{2}\tau_{\text{eff}}\gamma_x^s\left([\eta_x^c]^2+[\eta_y^c]^2\right)L_f^2\left[\sigma_L^2\sum_{i=1}^n w_i\|\boldsymbol{a}_{i,-1}\|_2^2 + 2\sigma_G^2 M_{\mathbf{a}_{-1}}\right]$$

$$+10\tau_{\text{eff}}\gamma_x^s L_f^2 M_{\mathbf{a}_{-1}}\beta_G^2\left[[\eta_x^c]^2\left\|\nabla\widetilde{\Phi}(\mathbf{x}^{(t)})\right\|^2 + L_f\left(2\kappa[\eta_x^c]^2+[\eta_y^c]^2\right)\left[\widetilde{\Phi}(\mathbf{x}^{(t)}) - \widetilde{F}(\mathbf{x}^{(t)},\mathbf{y}^{(t)})\right]\right]. \tag{16}$$

Summing (16) over $t = 0, \ldots, T-1$, substituting the bound on $\mathbb{E}\left[\widetilde{\Phi}(\mathbf{x}^{(t)}) - \widetilde{F}(\mathbf{x}^{(t)},\mathbf{y}^{(t)})\right]$ from Lemma B.5, and rearranging the terms, we get

$$\frac{1}{T}\sum_{t=0}^{T-1}\mathbb{E}\left\|\nabla\widetilde{\Phi}(\mathbf{x}^{(t)})\right\|^2$$

$$= \mathcal{O}\left(\frac{\kappa^2\Delta_{\widetilde{\Phi}}}{\tau_{\text{eff}}\gamma_y^s T} + \tau_{\text{eff}}\gamma_y^s L_f\kappa^2\frac{n}{P}\left[\sigma_L^2\sum_{i=1}^n\frac{w_i^2\|\boldsymbol{a}_i\|_2^2}{\|\boldsymbol{a}_i\|_1^2} + \sigma_G^2\left(\frac{n-P}{n-1}\max_i w_i + \beta_L^2\max_i\frac{w_i\|\boldsymbol{a}_i\|_2^2}{\|\boldsymbol{a}_i\|_1^2}\right)\right]\right)$$

$$+ \mathcal{O}\left(\kappa^2\left([\eta_x^c]^2+[\eta_y^c]^2\right)L_f^2\left[\sigma_L^2\sum_{i=1}^n w_i\left(\|\boldsymbol{a}_i\|_2^2 - [a_i^{(t,\tau_i-1)}]^2\right) + \sigma_G^2\max_i\left(\|\boldsymbol{a}_{i,-1}\|_1^2 + \beta_L^2\|\boldsymbol{a}_{i,-1}\|_2^2\right)\right]\right) \tag{17}$$

Consequently, using constants $A_w, B_w, C_w, D, E_w$, (17) can be simplified to

$$\frac{1}{T}\sum_{t=0}^{T-1}\mathbb{E}\left\|\nabla\widetilde{\Phi}(\mathbf{x}^{(t)})\right\|^2 \leq \mathcal{O}\left(\kappa^2\left[\frac{\Delta_{\widetilde{\Phi}}}{\tau_{\text{eff}}\gamma_y^s T} + \frac{\gamma_y^s L_f}{P}\left(A_w\sigma_L^2 + \left(B_w\beta_L^2 + \frac{n-P}{n-1}E_w\tau_{\text{eff}}\right)\sigma_G^2\right)\right]\right)$$

$$+ \mathcal{O}\left(\kappa^2 \left([\eta_x^c]^2 + [\eta_y^c]^2\right) L_f^2 \left[C_w \sigma_L^2 + D\sigma_G^2\right]\right),$$

which completes the proof. □

**Convergence in terms of $F$**

*Proof of Corollary 1.1.* According to the definition of $F(\mathbf{x})$ and $\widetilde{F}(\mathbf{x})$, we have

$$\nabla\Phi(\mathbf{x}) - \nabla\widetilde{\Phi}(\mathbf{x}) = \sum_{i=1}^{n} \left[p_i \nabla_x f_i(\mathbf{x}, \mathbf{y}^*(\mathbf{x})) - w_i \nabla_x f_i(\mathbf{x}, \widetilde{\mathbf{y}}^*(\mathbf{x}))\right] \qquad (\mathbf{y}^*(\mathbf{x}) \in \arg\max_{\mathbf{y}} F(\mathbf{x}, \mathbf{y}))$$

$$= \sum_{i=1}^{n} p_i \left[\nabla_x f_i(\mathbf{x}, \mathbf{y}^*(\mathbf{x})) - \nabla_x f_i(\mathbf{x}, \widetilde{\mathbf{y}}^*(\mathbf{x}))\right] + \sum_{i=1}^{n} (p_i - w_i) \nabla_x f_i(\mathbf{x}, \widetilde{\mathbf{y}}^*(\mathbf{x}))$$

$$= \left[\nabla_x F(\mathbf{x}, \mathbf{y}^*(\mathbf{x})) - \nabla_x F(\mathbf{x}, \widetilde{\mathbf{y}}^*(\mathbf{x}))\right] + \sum_{i=1}^{n} \frac{p_i - w_i}{\sqrt{w_i}} \cdot \sqrt{w_i} \nabla_x f_i(\mathbf{x}, \widetilde{\mathbf{y}}^*(\mathbf{x})).$$

Taking norm, using $L_f$-smoothness and applying Cauchy–Schwarz inequality, we get

$$\left\|\nabla\Phi(\mathbf{x}) - \nabla\widetilde{\Phi}(\mathbf{x})\right\|^2 \leq 2L_f^2 \left\|\mathbf{y}^*(\mathbf{x}) - \widetilde{\mathbf{y}}^*(\mathbf{x})\right\|^2 + 2\left[\sum_{i=1}^{n} \frac{(p_i - w_i)^2}{w_i}\right]\left[\sum_{i=1}^{n} w_i \left\|\nabla_x f_i(\mathbf{x}, \widetilde{\mathbf{y}}^*(\mathbf{x}))\right\|^2\right]$$

$$\leq 2L_f^2 \left\|\mathbf{y}^*(\mathbf{x}) - \widetilde{\mathbf{y}}^*(\mathbf{x})\right\|^2 + 2\chi_{\mathbf{p}\|\mathbf{w}}^2 \left[\beta_G^2 \left\|\nabla\widetilde{\Phi}(\mathbf{x})\right\|^2 + \sigma_G^2\right],$$

where the last inequality uses Assumption 4. Next, note that

$$\left\|\nabla\Phi(\mathbf{x})\right\|^2 \leq 2\left\|\nabla\Phi(\mathbf{x}) - \nabla\widetilde{\Phi}(\mathbf{x})\right\|^2 + 2\left\|\nabla\widetilde{\Phi}(\mathbf{x})\right\|^2.$$

Therefore, we obtain

$$\min_{t\in[T]} \left\|\nabla\Phi(\mathbf{x}^{(t)})\right\|^2 \leq \frac{1}{T}\sum_{t=0}^{T-1} \left\|\nabla\Phi(\mathbf{x}^{(t)})\right\|^2$$

$$\leq 2\left[2\chi_{\mathbf{p}\|\mathbf{w}}^2\beta_G^2 + 1\right]\frac{1}{T}\sum_{t=0}^{T-1}\left\|\nabla\widetilde{\Phi}(\mathbf{x}^{(t)})\right\|^2 + 4\left[\chi_{\mathbf{p}\|\mathbf{w}}^2\sigma_G^2 + L_f^2\frac{1}{T}\sum_{t=0}^{T-1}\left\|\mathbf{y}^*(\mathbf{x}^{(t)}) - \widetilde{\mathbf{y}}^*(\mathbf{x}^{(t)})\right\|^2\right]$$

$$= 2\left[2\chi_{\mathbf{p}\|\mathbf{w}}^2\beta_G^2 + 1\right]\epsilon_{\mathrm{opt}} + 4\left[\chi_{\mathbf{p}\|\mathbf{w}}^2\sigma_G^2 + L_f^2\frac{1}{T}\sum_{t=0}^{T-1}\left\|\mathbf{y}^*(\mathbf{x}^{(t)}) - \widetilde{\mathbf{y}}^*(\mathbf{x}^{(t)})\right\|^2\right].$$

where $\epsilon_{\mathrm{opt}}$ denotes the optimization error in the right hand side of (4) in Theorem 1. □

*Proof of Corollary 1.2.* If clients are weighted equally ($w_i = p_i = 1/n$ for all $i$), with each carrying out $\tau$ steps of local SGDA, from (4) we get

$$\min_{t\in[T]}\left\|\nabla\Phi(\mathbf{x}^{(t)})\right\|^2 \leq \mathcal{O}\left(\sqrt{\frac{n-P}{n-1}}\frac{\kappa^2\sigma_G}{\sqrt{PT}} + \kappa^2\left(\frac{\sigma_L + \beta_L\sigma_G}{\sqrt{P\tau T}} + \frac{\sigma_L^2 + \tau\sigma_G^2}{\tau T}\right)\right).$$

- For full client participation, this reduces to

$$\min_{t\in[T]}\mathbb{E}\left\|\nabla\widetilde{\Phi}(\mathbf{x}^{(t)})\right\|^2 \leq \mathcal{O}\left(\frac{1}{\sqrt{n\tau T}} + \frac{1}{T}\right).$$

To reach an $\epsilon$-stationary point, assuming $n\tau \leq T$, the per-client gradient complexity is $T\tau = \mathcal{O}\left(\frac{\kappa^4}{n\epsilon^4}\right)$.

Since $\tau \leq T/n$, the minimum number of communication rounds required is $T = \mathcal{O}\left(\frac{\kappa^2}{\epsilon^2}\right)$.

- For partial participation, $\mathcal{O}\left(\left(\frac{n-P}{n-1}\right)\sigma_G^2\sqrt{\frac{\tau}{PT}}\right)$ is the dominant term, and we do not get any convergence benefit of multiple local updates. Consequently, per-gradient client complexity and number of communication rounds are both $T\tau = \mathcal{O}\left(\frac{\kappa^4}{P\epsilon^4}\right)$, for $\tau = \mathcal{O}(1)$. However, if the data across clients comes from identical distributions ($\sigma_G = 0$), then we recover per-client gradient complexity of $\mathcal{O}\left(\frac{\kappa^4}{P\epsilon^4}\right)$, and number of communication rounds $= \mathcal{O}\left(\frac{\kappa^2}{\epsilon^2}\right)$.

$\square$

## B.3 Proofs of the Intermediate Lemmas

*Proof of Lemma B.2.*

$$
\mathbb{E}_t\left\|\sum_{i\in\mathcal{C}^{(t)}}\tilde{w}_i\mathbf{g}_{\mathbf{x},i}^{(t)}\right\|^2 = \mathbb{E}_t\left\|\sum_{i\in\mathcal{C}^{(t)}}\tilde{w}_i\left(\mathbf{g}_{\mathbf{x},i}^{(t)}-\mathbf{h}_{\mathbf{x},i}^{(t)}+\mathbf{h}_{\mathbf{x},i}^{(t)}\right)\right\|^2
$$

$$
\overset{(a)}{=} \mathbb{E}_t\left\|\sum_{i\in\mathcal{C}^{(t)}}\tilde{w}_i\left(\mathbf{g}_{\mathbf{x},i}^{(t)}-\mathbf{h}_{\mathbf{x},i}^{(t)}\right)\right\|^2 + \mathbb{E}_t\left\|\sum_{i\in\mathcal{C}^{(t)}}\tilde{w}_i\mathbf{h}_{\mathbf{x},i}^{(t)}\right\|^2
$$

$$
= \mathbb{E}_t\left[\sum_{i\in\mathcal{C}^{(t)}}\tilde{w}_i^2\left\|\mathbf{g}_{\mathbf{x},i}^{(t)}-\mathbf{h}_{\mathbf{x},i}^{(t)}\right\|^2\right] + \mathbb{E}_t\left\|\sum_{i\in\mathcal{C}^{(t)}}\tilde{w}_i\mathbf{h}_{\mathbf{x},i}^{(t)}\right\|^2 \qquad (\because \mathbb{E}_t[\mathbf{g}_{\mathbf{x},i}^{(t)}] = \mathbf{h}_{\mathbf{x},i}^{(t)} \text{ for all clients } i \in \mathcal{C}^{(t)})
$$

$$
= \frac{n}{P}\sum_{i=1}^n w_i^2\mathbb{E}_t\left\|\mathbf{g}_{\mathbf{x},i}^{(t)}-\mathbf{h}_{\mathbf{x},i}^{(t)}\right\|^2 + \mathbb{E}_t\left\|\sum_{i\in\mathcal{C}^{(t)}}\tilde{w}_i\mathbf{h}_{\mathbf{x},i}^{(t)}\right\|^2 \qquad (\because \tilde{w}_i = w_i n/P \text{ and } \mathbb{P}(i \in \mathcal{C}^{(t)}) = P/n)
$$

$$
\leq \frac{n}{P}\sum_{i=1}^n\frac{w_i^2}{\|\boldsymbol{a}_i\|_1^2}\sum_{k=0}^{\tau_i-1}[a_i^{(k)}(\tau_i)]^2\left[\sigma_L^2 + \beta_L^2\left\|\nabla_x f_i(\mathbf{x}_i^{(t,k)},\mathbf{y}_i^{(t,k)})\right\|^2\right] + \mathbb{E}_t\left\|\sum_{i\in\mathcal{C}^{(t)}}\tilde{w}_i\mathbf{h}_{\mathbf{x},i}^{(t)}\right\|^2. \qquad (18)
$$

Here, $(a)$ follows from the following reasoning.

$$
\mathbb{E}_t\left[\sum_{i,j\in\mathcal{C}^{(t)}}\tilde{w}_i\tilde{w}_j\left\langle\mathbf{h}_{\mathbf{x},i}^{(t)},\mathbf{g}_{\mathbf{x},j}^{(t)}-\mathbf{h}_{\mathbf{x},j}^{(t)}\right\rangle\right]
$$

$$
= \mathbb{E}_t\left[\sum_{i\in\mathcal{C}^{(t)}}\tilde{w}_i^2\mathbb{E}\left[\left\langle\mathbf{h}_{\mathbf{x},i}^{(t)},\mathbf{g}_{\mathbf{x},i}^{(t)}-\mathbf{h}_{\mathbf{x},i}^{(t)}\right\rangle \mid \mathcal{F}(t),\mathcal{C}^{(t)}\right] + \sum_{i\neq j}\tilde{w}_i\tilde{w}_j\underbrace{\mathbb{E}\left[\left\langle\mathbf{h}_{\mathbf{x},i}^{(t)},\mathbf{g}_{\mathbf{x},j}^{(t)}-\mathbf{h}_{\mathbf{x},j}^{(t)}\right\rangle \mid \mathcal{F}(t),\mathcal{C}^{(t)}\right]}_{=0}\right]
$$

$$
\text{(Assumption 3; independence of stochastic gradients across clients)}
$$

$$
= \mathbb{E}_t\left[\sum_{i\in\mathcal{C}^{(t)}}\tilde{w}_i^2\mathbb{E}\left[\left\langle\mathbf{h}_{\mathbf{x},i}^{(t)},\mathbf{g}_{\mathbf{x},i}^{(t)}-\mathbf{h}_{\mathbf{x},i}^{(t)}\right\rangle \mid \mathcal{F}(t),\mathcal{C}^{(t)}\right]\right]
$$

$$
= \mathbb{E}_t\left[\sum_{i\in\mathcal{C}^{(t)}}\frac{\tilde{w}_i^2}{\|\boldsymbol{a}_i\|_1^2}\sum_{k=0}^{\tau_i-1}\sum_{j=0}^{\tau_i-1}a_i^{(k)}(\tau_i)a_i^{(j)}(\tau_i)\,\mathbb{E}\left[\left\langle\nabla_x f_i\left(\mathbf{x}_i^{(t,k)},\mathbf{y}_i^{(t,k)};\xi_i^{(t,k)}\right)-\nabla_x f_i\left(\mathbf{x}_i^{(t,k)},\mathbf{y}_i^{(t,k)}\right),\right.\right.\right.
$$

$$
\left.\left.\left.\nabla_x f_i\left(\mathbf{x}_i^{(t,j)},\mathbf{y}_i^{(t,j)}\right)\right\rangle \mid \mathcal{F}(t),\mathcal{C}^{(t)}\right]\right]
$$

$$
= \mathbb{E}_t\left[\sum_{i\in\mathcal{C}^{(t)}}\frac{\tilde{w}_i^2}{\|\boldsymbol{a}_i\|_1^2}\left[\sum_{k=0}^{\tau_i-1}[a_i^{(k)}(\tau_i)]^2\,\mathbb{E}\left[\left\langle\underbrace{\mathbb{E}\left[\nabla_x f_i(\mathbf{x}_i^{(t,k)},\mathbf{y}_i^{(t,k)};\xi_i^{(t,k)})-\nabla_x f_i(\mathbf{x}_i^{(t,k)},\mathbf{y}_i^{(t,k)})\Big|\mathbf{x}_i^{(t,k)},\mathbf{y}_i^{(t,k)}\right]}_{=0},\right.\right.\right.\right.
$$

$$\nabla_x f_i(\mathbf{x}_i^{(t,k)}, \mathbf{y}_i^{(t,k)}) \Big\rangle \Big| \mathcal{F}(t), \mathcal{C}^{(t)} \Big]\Big]$$

$$+ 2\sum_{j<k} a_i^{(k)}(\tau_i) a_i^{(j)}(\tau_i) \mathbb{E}\Big[\Big\langle \underbrace{\mathbb{E}\Big[\nabla_x f_i(\mathbf{x}_i^{(t,k)}, \mathbf{y}_i^{(t,k)}; \xi_i^{(t,k)}) - \nabla_x f_i(\mathbf{x}_i^{(t,k)}, \mathbf{y}_i^{(t,k)}) \Big| \mathbf{x}_i^{(t,j)}, \mathbf{y}_i^{(t,j)}\Big]}_{=0},$$

$$\nabla_x f_i(\mathbf{x}_i^{(t,j)}, \mathbf{y}_i^{(t,j)}) \Big\rangle \Big| \mathcal{F}(t), \mathcal{C}^{(t)} \Big]\Big]$$

$$= 0. \qquad\qquad (\because \text{Assumption 3})$$

The last inequality in (18) follows from Assumption 1 and 3. Further, we can bound the second term in (18) as follows.

$$\mathbb{E}_t \left\| \sum_{i \in \mathcal{C}^{(t)}} \tilde{w}_i \mathbf{h}_{\mathbf{x},i}^{(t)} - \sum_{i=1}^n w_i \mathbf{h}_{\mathbf{x},i}^{(t)} + \sum_{i=1}^n w_i \mathbf{h}_{\mathbf{x},i}^{(t)} \right\|^2$$

$$= \mathbb{E}_t \left\| \sum_{i=1}^n w_i \mathbf{h}_{\mathbf{x},i}^{(t)} \right\|^2 + \mathbb{E}_t \left\| \sum_{i=1}^n \mathbb{I}(i \in \mathcal{C}^{(t)}) \tilde{w}_i \mathbf{h}_{\mathbf{x},i}^{(t)} - \sum_{i=1}^n w_i \mathbf{h}_{\mathbf{x},i}^{(t)} \right\|^2 \qquad \text{((WOR) sampling)}$$

$$= \mathbb{E}_t \left\| \sum_{i=1}^n w_i \mathbf{h}_{\mathbf{x},i}^{(t)} \right\|^2 + \sum_{i=1}^n \mathbb{E}_t \left[ \Big( (\mathbb{I}(i \in \mathcal{C}^{(t)}))^2 \tilde{w}_i^2 + w_i^2 - 2\mathbb{I}(i \in \mathcal{C}^{(t)}) \tilde{w}_i w_i \Big) \left\| \mathbf{h}_{\mathbf{x},i}^{(t)} \right\|^2 \right]$$

$$\quad + \sum_{i \neq j} \mathbb{E}_t \left\langle (\mathbb{I}(i \in \mathcal{C}^{(t)}) \tilde{w}_i - w_i) \mathbf{h}_{\mathbf{x},i}^{(t)}, (\mathbb{I}(j \in \mathcal{C}^{(t)}) \tilde{w}_j - w_j) \mathbf{h}_{\mathbf{x},j}^{(t)} \right\rangle$$

$$= \mathbb{E}_t \left\| \sum_{i=1}^n w_i \mathbf{h}_{\mathbf{x},i}^{(t)} \right\|^2 + \sum_{i=1}^n \mathbb{E}_t \left[ w_i^2 \left( \frac{n}{P} - 1 \right) \left\| \mathbf{h}_{\mathbf{x},i}^{(t)} \right\|^2 \right]$$

$$\quad + \sum_{i \neq j} \mathbb{E}_t \left[ \Big( \mathbb{I}(i \in \mathcal{C}^{(t)}) \cdot \mathbb{I}(j \in \mathcal{C}^{(t)}) \tilde{w}_i \tilde{w}_j - \mathbb{I}(j \in \mathcal{C}^{(t)}) \tilde{w}_j w_i - \mathbb{I}(i \in \mathcal{C}^{(t)}) \tilde{w}_i w_j + w_i w_j \Big) \left\langle \mathbf{h}_{\mathbf{x},i}^{(t)}, \mathbf{h}_{\mathbf{x},j}^{(t)} \right\rangle \right]$$

$$= \mathbb{E}_t \left\| \sum_{i=1}^n w_i \mathbf{h}_{\mathbf{x},i}^{(t)} \right\|^2 + \left( \frac{n}{P} - 1 \right) \sum_{i=1}^n \mathbb{E}_t \left[ w_i^2 \left\| \mathbf{h}_{\mathbf{x},i}^{(t)} \right\|^2 \right] + \sum_{i \neq j} \mathbb{E}_t \left[ w_i w_j \left( \frac{n}{P} \left( \frac{P-1}{n-1} \right) - 1 \right) \left\langle \mathbf{h}_{\mathbf{x},i}^{(t)}, \mathbf{h}_{\mathbf{x},j}^{(t)} \right\rangle \right]$$

$$= \frac{n}{P} \left( \frac{P-1}{n-1} \right) \mathbb{E}_t \left\| \sum_{i=1}^n w_i \mathbf{h}_{\mathbf{x},i}^{(t)} \right\|^2 + \frac{n}{P} \frac{n-P}{n-1} \sum_{i=1}^n w_i^2 \mathbb{E}_t \left\| \mathbf{h}_{\mathbf{x},i}^{(t)} \right\|^2, \qquad (19)$$

Next, we bound the second term in (19).

$$\sum_{i=1}^n w_i^2 \mathbb{E}_t \left\| \mathbf{h}_{\mathbf{x},i}^{(t)} - \nabla_x f_i(\mathbf{x}^{(t)}, \mathbf{y}^{(t)}) + \nabla_x f_i(\mathbf{x}^{(t)}, \mathbf{y}^{(t)}) \right\|^2$$

$$\leq 2L_f^2 \sum_{i=1}^n \frac{w_i^2}{\|\boldsymbol{a}_i\|_1} \sum_{k=0}^{\tau_i-1} a_i^{(k)}(\tau_i) \Delta_{\mathbf{x},\mathbf{y}}^{(t,k)}(i) + 2(\max_i w_i) \left( \beta_G^2 \left\| \nabla_x \widetilde{F}(\mathbf{x}^{(t)}, \mathbf{y}^{(t)}) \right\|^2 + \sigma_G^2 \right), \qquad (20)$$

using Assumption 4. $\Delta_{\mathbf{x},\mathbf{y}}^{(t,k)}(i) \triangleq \mathbb{E}_t \left[ \|\mathbf{x}_i^{(t,k)} - \mathbf{x}^{(t)}\|^2 + \|\mathbf{y}_i^{(t,k)} - \mathbf{y}^{(t)}\|^2 \right]$. Substituting (20) in (19), and using the resulting bound in (18) we get the bound in (13). $\qquad \square$

*Proof of Lemma B.3.* Since the local functions $\{f_i\}$ satisfy Assumption 5, $F(\mathbf{x}, \cdot)$ is $\mu$-SC for any $\mathbf{x}$. In the proof, we use the quadratic growth property of $\mu$-SC function $F(\mathbf{x}, \cdot)$, i.e., for any given $\mathbf{x}$

$$\frac{\mu}{2} \|\mathbf{y} - \mathbf{y}^*(\mathbf{x})\|^2 \leq F(\mathbf{x}, \mathbf{y}^*(\mathbf{x})) - F(\mathbf{x}, \mathbf{y}), \quad \text{for all } \mathbf{y}, \qquad (21)$$

where $\mathbf{y}^*(\mathbf{x}) = \arg\max_{\mathbf{y}'} F(\mathbf{x}, \mathbf{y}')$. Using $L_\Phi$-smoothness of $\widetilde{\Phi}(\cdot)$,

$$
\mathbb{E}_t\left[\widetilde{\Phi}(\mathbf{x}^{(t+1)})\right] \le \mathbb{E}_t\widetilde{\Phi}(\mathbf{x}^{(t)}) - \mathbb{E}_t\left\langle \nabla\widetilde{\Phi}(\mathbf{x}^{(t)}), \tau_{\text{eff}}\gamma_x^s \sum_{i\in\mathcal{C}^{(t)}} \tilde{w}_i \mathbf{g}_{\mathbf{x},i}^{(t)}\right\rangle + \frac{\tau_{\text{eff}}^2[\gamma_x^s]^2 L_\Phi}{2}\mathbb{E}_t\left\|\mathbf{g}_{\mathbf{x}}^{(t)}\right\|^2
$$

$$
= \widetilde{\Phi}(\mathbf{x}^{(t)}) - \tau_{\text{eff}}\gamma_x^s\mathbb{E}_t\left\langle \nabla\widetilde{\Phi}(\mathbf{x}^{(t)}), \sum_{i=1}^n w_i\mathbf{h}_{\mathbf{x},i}^{(t)}\right\rangle + \frac{\tau_{\text{eff}}^2[\gamma_x^s]^2 L_\Phi}{2}\mathbb{E}_t\left\|\mathbf{g}_{\mathbf{x}}^{(t)}\right\|^2 \qquad \text{(using Assumption 3 and (3))}
$$

$$
= \widetilde{\Phi}(\mathbf{x}^{(t)}) - \frac{\tau_{\text{eff}}\gamma_x^s}{2}\left[\left\|\nabla\widetilde{\Phi}(\mathbf{x}^{(t)})\right\|^2 + \mathbb{E}_t\left\|\sum_{i=1}^n w_i\mathbf{h}_{\mathbf{x},i}^{(t)}\right\|^2\right] + \frac{\tau_{\text{eff}}\gamma_x^s}{2}\mathbb{E}_t\left\|\nabla\widetilde{\Phi}(\mathbf{x}^{(t)}) - \sum_{i=1}^n w_i\mathbf{h}_{\mathbf{x},i}^{(t)}\right\|^2
$$

$$
+ \frac{\tau_{\text{eff}}^2[\gamma_x^s]^2 L_\Phi}{2}\mathbb{E}_t\left\|\mathbf{g}_{\mathbf{x}}^{(t)}\right\|^2. \tag{22}
$$

The last term is bounded in Lemma B.2. Next, we bound the third term above.

$$
\mathbb{E}_t\left\|\nabla\widetilde{\Phi}(\mathbf{x}^{(t)}) - \sum_{i=1}^n w_i\mathbf{h}_{\mathbf{x},i}^{(t)}\right\|^2
$$

$$
= \mathbb{E}_t\left\|\sum_{i=1}^n w_i\left(\nabla_x f_i(\mathbf{x}^{(t)}, \widetilde{\mathbf{y}}^*(\mathbf{x}^{(t)})) - \nabla_x f_i(\mathbf{x}^{(t)}, \mathbf{y}^{(t)}) + \nabla_x f_i(\mathbf{x}^{(t)}, \mathbf{y}^{(t)}) - \mathbf{h}_{\mathbf{x},i}^{(t)}\right)\right\|^2
$$

$$
\text{(since } \widetilde{\mathbf{y}}^*(\mathbf{x}) = \arg\max_{\mathbf{y}'} \widetilde{F}(\mathbf{x}, \mathbf{y}'))
$$

$$
\le 2L_f^2\mathbb{E}_t\left\|\widetilde{\mathbf{y}}^*(\mathbf{x}^{(t)}) - \mathbf{y}^{(t)}\right\|^2 + 2\mathbb{E}_t\left\|\sum_{i=1}^n w_i\left(\nabla_x f_i(\mathbf{x}^{(t)}, \mathbf{y}^{(t)}) - \frac{1}{\|\boldsymbol{a}_i\|_1}\sum_{k=0}^{\tau_i-1} a_i^{(k)}(\tau_i)\nabla_x f_i\left(\mathbf{x}_i^{(t,k)}, \mathbf{y}_i^{(t,k)}\right)\right)\right\|^2
$$

$$
\text{(}L_f\text{-smoothness; Young's inequality)}
$$

$$
\le \frac{4L_f^2}{\mu}\left[\widetilde{\Phi}(\mathbf{x}^{(t)}) - \widetilde{F}(\mathbf{x}^{(t)}, \mathbf{y}^{(t)})\right] + 2\sum_{i=1}^n w_i\frac{1}{\|\boldsymbol{a}_i\|_1}\sum_{k=0}^{\tau_i-1} a_i^{(k)}(\tau_i)\mathbb{E}_t\left\|\nabla_x f_i(\mathbf{x}^{(t)}, \mathbf{y}^{(t)}) - \nabla_x f_i(\mathbf{x}_i^{(t,k)}, \mathbf{y}_i^{(t,k)})\right\|^2
$$

$$
\text{(Quadratic growth of } \mu\text{-SC functions (21); Jensen's inequality)}
$$

$$
\le \frac{4L_f^2}{\mu}\left[\widetilde{\Phi}(\mathbf{x}^{(t)}) - \widetilde{F}(\mathbf{x}^{(t)}, \mathbf{y}^{(t)})\right] + 2L_f^2\sum_{i=1}^n \frac{w_i}{\|\boldsymbol{a}_i\|_1}\sum_{k=0}^{\tau_i-1} a_i^{(k)}(\tau_i)\mathbb{E}_t\left[\left\|\mathbf{x}_i^{(t,k)} - \mathbf{x}^{(t)}\right\|^2 + \left\|\mathbf{y}_i^{(t,k)} - \mathbf{y}^{(t)}\right\|^2\right]
$$

$$
\text{(}L_f\text{-smoothness)}
$$

$$
= \frac{4L_f^2}{\mu}\left[\widetilde{\Phi}(\mathbf{x}^{(t)}) - \widetilde{F}(\mathbf{x}^{(t)}, \mathbf{y}^{(t)})\right] + 2L_f^2\sum_{i=1}^n \frac{w_i}{\|\boldsymbol{a}_i\|_1}\sum_{k=0}^{\tau_i-1} a_i^{(k)}(\tau_i)\Delta_{\mathbf{x},\mathbf{y}}^{(t,k)}(i). \tag{23}
$$

where $\Delta_{\mathbf{x},\mathbf{y}}^{(t,k)}(i) \triangleq \mathbb{E}_t\left[\left\|\mathbf{x}_i^{(t,k)} - \mathbf{x}^{(t)}\right\|^2 + \left\|\mathbf{y}_i^{(t,k)} - \mathbf{y}^{(t)}\right\|^2\right]$. Further, the term containing $\left\|\nabla_x f_i(\mathbf{x}_i^{(t,k)}, \mathbf{y}_i^{(t,k)})\right\|^2$ in (13) is bounded in Lemma B.7. Substituting the bounds from (23), (13) and Lemma B.7 into (22), we get

$$
\mathbb{E}_t\left[\widetilde{\Phi}(\mathbf{x}^{(t+1)})\right] \le \widetilde{\Phi}(\mathbf{x}^{(t)}) - \frac{\tau_{\text{eff}}\gamma_x^s}{2}\left[\left\|\nabla\widetilde{\Phi}(\mathbf{x}^{(t)})\right\|^2 + \mathbb{E}_t\left\|\sum_{i=1}^n w_i\mathbf{h}_{\mathbf{x},i}^{(t)}\right\|^2\right]
$$

$$
+ \frac{\tau_{\text{eff}}\gamma_x^s}{2}\left[\frac{4L_f^2}{\mu}\left[\widetilde{\Phi}(\mathbf{x}^{(t)}) - \widetilde{F}(\mathbf{x}^{(t)}, \mathbf{y}^{(t)})\right] + 2L_f^2\sum_{i=1}^n \frac{w_i}{\|\boldsymbol{a}_i\|_1}\sum_{k=0}^{\tau_i-1} a_i^{(k)}(\tau_i)\Delta_{\mathbf{x},\mathbf{y}}^{(t,k)}(i)\right]
$$

$$
+ \frac{\tau_{\text{eff}}^2[\gamma_x^s]^2 L_\Phi}{2}\left[\frac{n}{P}\left(\frac{P-1}{n-1}\right)\mathbb{E}_t\left\|\sum_{i=1}^n w_i\mathbf{h}_{\mathbf{x},i}^{(t)}\right\|^2 + \frac{n}{P}\left(\frac{n-P}{n-1}\right)2L_f^2\sum_{i=1}^n \frac{w_i^2}{\|\boldsymbol{a}_i\|_1}\sum_{k=0}^{\tau_i-1} a_i^{(k)}(\tau_i)\Delta_{\mathbf{x},\mathbf{y}}^{(t,k)}(i)\right]
$$

$$
+ \frac{\tau_{\text{eff}}^2[\gamma_x^s]^2 L_\Phi}{2}\frac{n\sigma_L^2}{P}\sum_{i=1}^n \frac{w_i^2\|\boldsymbol{a}_i\|_2^2}{\|\boldsymbol{a}_i\|_1^2} + \frac{\tau_{\text{eff}}^2[\gamma_x^s]^2 L_\Phi}{2}\frac{n}{P}\left(\frac{n-P}{n-1}\right)2(\max_i w_i)\left(\beta_G^2\left\|\nabla_x\widetilde{F}(\mathbf{x}^{(t)}, \mathbf{y}^{(t)})\right\|^2 + \sigma_G^2\right)
$$

$$+ \frac{\tau_{\text{eff}}^2 [\gamma_x^s]^2 L_\Phi}{2} \frac{n}{P} \beta_L^2 \left[ 2L_f^2 \sum_{i=1}^n \frac{w_i^2}{\|\boldsymbol{a}_i\|_1^2} \sum_{k=0}^{\tau_i-1} [a_i^{(k)}(\tau_i)]^2 \Delta_{\mathbf{x},\mathbf{y}}^{(t,k)}(i) + 2\sigma_G^2 \left( \max_i \frac{w_i \|\boldsymbol{a}_i\|_2^2}{\|\boldsymbol{a}_i\|_1^2} \right) \right]$$

$$+ \frac{\tau_{\text{eff}}^2 [\gamma_x^s]^2 L_\Phi}{2} \frac{n}{P} \beta_L^2 4\beta_G^2 \left( \max_i \frac{w_i \|\boldsymbol{a}_i\|_2^2}{\|\boldsymbol{a}_i\|_1^2} \right) \left[ \frac{2L_f^2}{\mu} \left( \widetilde{\Phi}(\mathbf{x}^{(t)}) - \widetilde{F}(\mathbf{x}^{(t)}, \mathbf{y}^{(t)}) \right) + \left\| \nabla_x \widetilde{\Phi}(\mathbf{x}^{(t)}) \right\|^2 \right]$$

$$\leq \widetilde{\Phi}(\mathbf{x}^{(t)}) - \frac{7\tau_{\text{eff}} \gamma_x^s}{16} \left\| \nabla \widetilde{\Phi}(\mathbf{x}^{(t)}) \right\|^2 - \frac{\tau_{\text{eff}} \gamma_x^s}{2} \left( 1 - \frac{n}{P} \left( \frac{P-1}{n-1} \right) \tau_{\text{eff}} \gamma_x^s L_\Phi \right) \mathbb{E}_t \left\| \sum_{i=1}^n w_i \mathbf{h}_{\mathbf{x},i}^{(t)} \right\|^2$$

$$+ \frac{9\tau_{\text{eff}} \gamma_x^s L_f^2}{4\mu} \left[ \widetilde{\Phi}(\mathbf{x}^{(t)}) - \widetilde{F}(\mathbf{x}^{(t)}, \mathbf{y}^{(t)}) \right] + \frac{5}{4} \tau_{\text{eff}} \gamma_x^s L_f^2 \sum_{i=1}^n \frac{w_i}{\|\boldsymbol{a}_i\|_1} \sum_{k=0}^{\tau_i-1} a_i^{(k)}(\tau_i) \Delta_{\mathbf{x},\mathbf{y}}^{(t,k)}(i)$$

$$+ \frac{\tau_{\text{eff}}^2 [\gamma_x^s]^2 L_\Phi}{2} \frac{n}{P} \left[ \sigma_L^2 \sum_{i=1}^n \frac{w_i^2 \|\boldsymbol{a}_i\|_2^2}{\|\boldsymbol{a}_i\|_1^2} + \sigma_G^2 \left( 2(\max_i w_i) \frac{n-P}{n-1} + 2\beta_L^2 \max_i \frac{w_i \|\boldsymbol{a}_i\|_2^2}{\|\boldsymbol{a}_i\|_1^2} \right) \right],$$

where, the coefficients are simplified, using assumptions on the learning rate $\gamma_x^s$

$$\tau_{\text{eff}} \gamma_x^s L_\Phi \left[ (\max_i w_i) \left( \frac{n-P}{n-1} \right) + \beta_L^2 \max_{i,k} \frac{w_i a_i^{(k)}(\tau_i)}{\|\boldsymbol{a}_i\|_1} \right] \leq \frac{P}{4n}$$

$$\tau_{\text{eff}} \gamma_x^s L_\Phi \left[ (\max_i w_i) \left( \frac{n-P}{n-1} \right) + \beta_L^2 \max_i \frac{w_i \|\boldsymbol{a}_i\|_2^2}{\|\boldsymbol{a}_i\|_1^2} \right] \leq \frac{P}{32\beta_G^2 n}.$$

This finishes the proof. $\qquad\square$

*Proof of Lemma B.4.* We use the client update equations for individual iterates (11). To bound $\Delta_{\mathbf{x},\mathbf{y}}^{(t,k)}(i)$, first we bound a single component term $\mathbb{E}_t \left\| \mathbf{x}_i^{(t,k)} - \mathbf{x}^{(t)} \right\|^2$. For $1 \leq k \leq \tau_i$,

$$\mathbb{E}_t \left\| \mathbf{x}_i^{(t,k)} - \mathbf{x}^{(t)} \right\|^2 = [\eta_x^c]^2 \mathbb{E}_t \left\| \sum_{j=0}^{k-1} a_i^{(j)}(k) \left( \nabla_x f_i \left( \mathbf{x}_i^{(t,j)}, \mathbf{y}_i^{(t,j)}; \xi_i^{(t,j)} \right) - \nabla_x f_i \left( \mathbf{x}_i^{(t,j)}, \mathbf{y}_i^{(t,j)} \right) + \nabla_x f_i \left( \mathbf{x}_i^{(t,j)}, \mathbf{y}_i^{(t,j)} \right) \right) \right\|^2$$

$$= [\eta_x^c]^2 \left[ \mathbb{E}_t \left\| \sum_{j=0}^{k-1} a_i^{(j)}(k) \left( \nabla_x f_i \left( \mathbf{x}_i^{(t,j)}, \mathbf{y}_i^{(t,j)}; \xi_i^{(t,j)} \right) - \nabla_x f_i \left( \mathbf{x}_i^{(t,j)}, \mathbf{y}_i^{(t,j)} \right) \right) \right\|^2 + \mathbb{E}_t \left\| \sum_{j=0}^{k-1} a_i^{(j)}(k) \nabla_x f_i \left( \mathbf{x}_i^{(t,j)}, \mathbf{y}_i^{(t,j)} \right) \right\|^2 \right]$$

(using unbiasedness in Assumption 3)

$$= [\eta_x^c]^2 \left[ \sum_{j=0}^{k-1} [a_i^{(j)}(k)]^2 \mathbb{E}_t \left\| \nabla_x f_i \left( \mathbf{x}_i^{(t,j)}, \mathbf{y}_i^{(t,j)}; \xi_i^{(t,j)} \right) - \nabla_x f_i \left( \mathbf{x}_i^{(t,j)}, \mathbf{y}_i^{(t,j)} \right) \right\|^2 + \mathbb{E}_t \left\| \sum_{j=0}^{k-1} a_i^{(j)}(k) \nabla_x f_i \left( \mathbf{x}_i^{(t,j)}, \mathbf{y}_i^{(t,j)} \right) \right\|^2 \right]$$

$$\leq [\eta_x^c]^2 \left[ \sum_{j=0}^{k-1} [a_i^{(j)}(k)]^2 \left( \sigma_L^2 + \beta_L^2 \mathbb{E}_t \left\| \nabla_x f_i(\mathbf{x}_i^{(t,j)}, \mathbf{y}_i^{(t,j)}) \right\|^2 \right) + \left( \sum_{j=0}^{k-1} a_i^{(j)}(k) \right) \sum_{j=0}^{k-1} a_i^{(j)}(k) \mathbb{E}_t \left\| \nabla_x f_i \left( \mathbf{x}_i^{(t,j)}, \mathbf{y}_i^{(t,j)} \right) \right\|^2 \right].$$

$$(24)$$

where the last inequality follows from Jensen's inequality (Lemma A.3). Next, note that

$$\frac{1}{\|\boldsymbol{a}_i\|_1} \sum_{k=0}^{\tau_i-1} a_i^{(k)}(\tau_i) \sum_{j=0}^{k-1} [a_i^{(j)}(k)]^2 \leq \frac{1}{\|\boldsymbol{a}_i\|_1} \sum_{k=0}^{\tau_i-1} a_i^{(k)}(\tau_i) \sum_{k=0}^{\tau_i-2} [a_i^{(j)}(k)]^2 = \sum_{k=0}^{\tau_i-2} [a_i^{(j)}(k)]^2 \leq \|\boldsymbol{a}_i\|_2^2 - [a_i^{(t,\tau_i-1)}(\tau_i)]^2,$$

$$\frac{1}{\|\boldsymbol{a}_i\|_1} \sum_{k=0}^{\tau_i-1} a_i^{(k)}(\tau_i) \sum_{j=0}^{k-1} a_i^{(j)}(k) \leq \frac{1}{\|\boldsymbol{a}_i\|_1} \sum_{k=0}^{\tau_i-1} a_i^{(k)}(\tau_i) \sum_{k=0}^{\tau_i-2} a_i^{(j)}(k) = \sum_{k=0}^{\tau_i-2} a_i^{(j)}(k) \leq \|\boldsymbol{a}_i\|_1 - [a_i^{(t,\tau_i-1)}(\tau_i)],$$

$$\frac{1}{\|\boldsymbol{a}_i\|_1} \sum_{k=0}^{\tau_i-1} a_i^{(k)}(\tau_i) \sum_{j=0}^{k-1} [a_i^{(j)}(k)]^2 \leq \frac{1}{\|\boldsymbol{a}_i\|_1} \sum_{k=0}^{\tau_i-1} a_i^{(k)}(\tau_i) \sum_{j=0}^{\tau_i-2} [a_i^{(j)}(k)]^2 \leq \alpha \cdot \sum_{k=0}^{\tau_i-2} a_i^{(k)}(\tau_i),$$

$$(25)$$

where $a_i^{(j)}(k) \leq \alpha$ for all $j, k$. We define $\|\boldsymbol{a}_{i,-1}\|_2^2 \triangleq \|\boldsymbol{a}_i\|_2^2 - [a_i^{(t,\tau_i-1)}(\tau_i)]^2$, $\|\boldsymbol{a}_{i,-1}\|_1 \triangleq \|\boldsymbol{a}_i\|_1 - [a_i^{(t,\tau_i-1)}(\tau_i)]$ for the sake of brevity. Using (25), we bound the individual terms in (24).

$$\frac{1}{\|\boldsymbol{a}_i\|_1} \sum_{k=0}^{\tau_i-1} a_i^{(k)}(\tau_i) \sum_{j=0}^{k-1} [a_i^{(j)}(k)]^2 \beta_L^2 \left\| \nabla_x f_i(\mathbf{x}_i^{(t,j)}, \mathbf{y}_i^{(t,j)}) \right\|^2$$

$$\leq 2\beta_L^2 \alpha L_f^2 \sum_{j=0}^{\tau_i-2} [a_i^{(j)}(k)] \Delta_{\mathbf{x},\mathbf{y}}^{(t,j)} + 2\beta_L^2 \|\boldsymbol{a}_{i,-1}\|_2^2 \left\| \nabla_x f_i(\mathbf{x}^{(t)}, \mathbf{y}^{(t)}) \right\|^2. \qquad (26)$$

Similarly,

$$\frac{1}{\|\boldsymbol{a}_i\|_1} \sum_{k=0}^{\tau_i-1} a_i^{(k)}(\tau_i) \left(\sum_{j=0}^{k-1} a_i^{(j)}(k)\right) \sum_{j=0}^{k-1} a_i^{(j)}(k) \mathbb{E}_t \left\| \nabla_x f_i \left( \mathbf{x}_i^{(t,j)}, \mathbf{y}_i^{(t,j)} \right) \right\|^2$$

$$\leq \frac{2}{\|\boldsymbol{a}_i\|_1} \sum_{k=0}^{\tau_i-1} a_i^{(k)}(\tau_i) \left(\sum_{j=0}^{\tau_i-2} a_i^{(j)}(k)\right) \sum_{j=0}^{\tau_i-2} a_i^{(j)}(k) \left[ L_f^2 \Delta_{\mathbf{x},\mathbf{y}}^{(t,j)} + \left\| \nabla_x f_i(\mathbf{x}^{(t)}, \mathbf{y}^{(t)}) \right\|^2 \right]$$

$$\leq 2 \|\boldsymbol{a}_{i,-1}\|_1 L_f^2 \sum_{j=0}^{\tau_i-2} a_i^{(j)}(k) \Delta_{\mathbf{x},\mathbf{y}}^{(t,j)} + 2 \|\boldsymbol{a}_{i,-1}\|_1^2 \left\| \nabla_x f_i(\mathbf{x}^{(t)}, \mathbf{y}^{(t)}) \right\|^2. \qquad (27)$$

Substituting (26), (27) in (24), we get

$$\frac{1}{\|\boldsymbol{a}_i\|_1} \sum_{k=0}^{\tau_i-1} a_i^{(k)}(\tau_i) \mathbb{E}_t \left\| \mathbf{x}_i^{(t,k)} - \mathbf{x}^{(t)} \right\|^2 \leq [\eta_x^c]^2 \sigma_L^2 \|\boldsymbol{a}_{i,-1}\|_2^2 + 2[\eta_x^c]^2 L_f^2 \left(\|\boldsymbol{a}_{i,-1}\|_1 + \beta_L^2 \alpha\right) \sum_{k=0}^{\tau_i-1} a_i^{(k)}(\tau_i) \Delta_{\mathbf{x},\mathbf{y}}^{(t,k)}(i)$$

$$+ 2[\eta_x^c]^2 \left(\|\boldsymbol{a}_{i,-1}\|_1^2 + \beta_L^2 \|\boldsymbol{a}_{i,-1}\|_2^2\right) \left\| \nabla_x f_i(\mathbf{x}^{(t)}, \mathbf{y}^{(t)}) \right\|^2. \qquad (28)$$

Similarly, we can bound the $\mathbf{y}$ error

$$\frac{1}{\|\boldsymbol{a}_i\|_1} \sum_{k=0}^{\tau_i-1} a_i^{(k)}(\tau_i) \mathbb{E}_t \left\| \mathbf{y}_i^{(t,k)} - \mathbf{y}^{(t)} \right\|^2 \leq [\eta_y^c]^2 \sigma_L^2 \|\boldsymbol{a}_{i,-1}\|_2^2 + 2[\eta_y^c]^2 L_f^2 \left(\|\boldsymbol{a}_{i,-1}\|_1 + \beta_L^2 \alpha\right) \sum_{k=0}^{\tau_i-1} a_i^{(k)}(\tau_i) \Delta_{\mathbf{x},\mathbf{y}}^{(t,k)}(i)$$

$$+ 2[\eta_y^c]^2 \left(\|\boldsymbol{a}_{i,-1}\|_1^2 + \beta_L^2 \|\boldsymbol{a}_{i,-1}\|_2^2\right) \left\| \nabla_y f_i(\mathbf{x}^{(t)}, \mathbf{y}^{(t)}) \right\|^2. \qquad (29)$$

Combining the two bounds in (28) and (29), we get

$$\frac{1}{\|\boldsymbol{a}_i\|_1} \sum_{k=0}^{\tau_i-1} a_i^{(k)}(\tau_i) \mathbb{E}_t \left[ \left\| \mathbf{x}_i^{(t,k)} - \mathbf{x}^{(t)} \right\|^2 + \left\| \mathbf{y}_i^{(t,k)} - \mathbf{y}^{(t)} \right\|^2 \right] \leq \left([\eta_x^c]^2 + [\eta_y^c]^2\right) \sigma_L^2 \|\boldsymbol{a}_{i,-1}\|_2^2$$

$$+ 2 \left([\eta_x^c]^2 + [\eta_y^c]^2\right) L_f^2 \|\boldsymbol{a}_i\|_1 \left(\|\boldsymbol{a}_{i,-1}\|_1 + \beta_L^2 \alpha\right) \frac{1}{\|\boldsymbol{a}_i\|_1} \sum_{k=0}^{\tau_i-1} a_i^{(k)}(\tau_i) \Delta_{\mathbf{x},\mathbf{y}}^{(t,k)}(i)$$

$$+ 2 \left( \|\boldsymbol{a}_{i,-1}\|_1^2 + \beta_L^2 \|\boldsymbol{a}_{i,-1}\|_2^2 \right) \left[ [\eta_x^c]^2 \left\| \nabla_x f_i \left( \mathbf{x}^{(t)}, \mathbf{y}^{(t)} \right) \right\|^2 + [\eta_y^c]^2 \left\| \nabla_y f_i \left( \mathbf{x}^{(t)}, \mathbf{y}^{(t)} \right) \right\|^2 \right]. \tag{30}$$

Define $A_m \triangleq 2L_f^2 \left( [\eta_x^c]^2 + [\eta_y^c]^2 \right) \max_i \|\boldsymbol{a}_i\|_1 \left( \|\boldsymbol{a}_{i,-1}\|_1 + \beta_L^2 \alpha \right)$. Rearranging the terms in (30), and taking the weighted sum over agents, we get

$$L_f^2 \sum_{i=1}^n \frac{w_i}{\|\boldsymbol{a}_i\|_1} \sum_{k=0}^{\tau_i-1} a_i^{(k)}(\tau_i) \Delta_{\mathbf{x},\mathbf{y}}^{(t,k)}(i)$$

$$\leq \frac{\left( [\eta_x^c]^2 + [\eta_y^c]^2 \right) L_f^2 \sigma_L^2}{1 - A_m} \sum_{i=1}^n w_i \|\boldsymbol{a}_{i,-1}\|_2^2$$

$$+ \frac{2L_f^2}{1 - A_m} \sum_{i=1}^n w_i \left( \|\boldsymbol{a}_{i,-1}\|_1^2 + \beta_L^2 \|\boldsymbol{a}_{i,-1}\|_2^2 \right) \left[ [\eta_x^c]^2 \left\| \nabla_x f_i \left( \mathbf{x}^{(t)}, \mathbf{y}^{(t)} \right) \right\|^2 + [\eta_y^c]^2 \left\| \nabla_y f_i \left( \mathbf{x}^{(t)}, \mathbf{y}^{(t)} \right) \right\|^2 \right]$$

$$\leq \frac{\left( [\eta_x^c]^2 + [\eta_y^c]^2 \right) L_f^2}{1 - A_m} \sigma_L^2 \sum_{i=1}^n w_i \|\boldsymbol{a}_{i,-1}\|_2^2 + \frac{2L_f^2 M_{\mathbf{a}_{-1}}}{1 - A_m} [\eta_x^c]^2 \left( \beta_G^2 \left\| \sum_{i=1}^n w_i \nabla_x f_i \left( \mathbf{x}^{(t)}, \mathbf{y}^{(t)} \right) \right\|^2 + \sigma_G^2 \right)$$

$$+ \frac{2L_f^2 M_{\mathbf{a}_{-1}}}{1 - A_m} [\eta_y^c]^2 \left( \beta_G^2 \left\| \sum_{i=1}^n w_i \nabla_y f_i \left( \mathbf{x}^{(t)}, \mathbf{y}^{(t)} \right) \right\|^2 + \sigma_G^2 \right). \tag{31}$$

where (31) follows from Assumption 4, and we define $M_{\mathbf{a}_{-1}} \triangleq \max_i \left( \|\boldsymbol{a}_{i,-1}\|_1^2 + \beta_L^2 \|\boldsymbol{a}_{i,-1}\|_2^2 \right)$. We bounded $\left\| \nabla_x F \left( \mathbf{x}^{(t)}, \mathbf{y}^{(t)} \right) \right\|^2$ in Lemma B.6. Similarly, we can bound $\left\| \nabla_y F \left( \mathbf{x}^{(t)}, \mathbf{y}^{(t)} \right) \right\|^2$ as follows.

$$\left\| \nabla_y F \left( \mathbf{x}^{(t)}, \mathbf{y}^{(t)} \right) \right\|^2 = \left\| \nabla_y F \left( \mathbf{x}^{(t)}, \mathbf{y}^{(t)} \right) - \nabla_y F \left( \mathbf{x}^{(t)}, \mathbf{y}^*(\mathbf{x}^{(t)}) \right) \right\|^2 \qquad (\because \mathbf{y}^*(\mathbf{x}) = \arg\max_{\mathbf{y}'} F(\mathbf{x}, \mathbf{y}'))$$

$$\leq 2L_f \left[ \widetilde{\Phi}(\mathbf{x}^{(t)}) - \widetilde{F}(\mathbf{x}^{(t)}, \mathbf{y}^{(t)}) \right]. \tag{32}$$

using $L_f$-smoothness and concavity of $F(\mathbf{x}, \cdot)$ (Lemma A.6). Also, for the choice of $\eta_x^c, \eta_y^c$, we get $A_m \leq 1/2$. Consequently, substituting the two bounds in (31), we complete the proof. $\square$

*Proof of Lemma B.5.* First we see that

$$\mathbb{E}_t \left[ \widetilde{\Phi}(\mathbf{x}^{(t+1)}) - \widetilde{F}(\mathbf{x}^{(t+1)}, \mathbf{y}^{(t+1)}) \right]$$

$$= \mathbb{E}_t \left[ \widetilde{\Phi}(\mathbf{x}^{(t+1)}) - \widetilde{\Phi}(\mathbf{x}^{(t)}) \right] + \left[ \widetilde{\Phi}(\mathbf{x}^{(t)}) - \widetilde{F}(\mathbf{x}^{(t)}, \mathbf{y}^{(t)}) \right] + \mathbb{E}_t \left[ \widetilde{F}(\mathbf{x}^{(t)}, \mathbf{y}^{(t)}) - \widetilde{F}(\mathbf{x}^{(t+1)}, \mathbf{y}^{(t+1)}) \right]. \tag{33}$$

$\mathbb{E}_t \left[ \widetilde{\Phi}(\mathbf{x}^{(t+1)}) - \widetilde{\Phi}(\mathbf{x}^{(t)}) \right]$ is already bounded in Lemma B.3. We bound $\mathbb{E}_t \left[ \widetilde{F}(\mathbf{x}^{(t)}, \mathbf{y}^{(t)}) - \widetilde{F}(\mathbf{x}^{(t+1)}, \mathbf{y}^{(t+1)}) \right]$ as follows. Using the notation $\mathbf{z}^{(t)} = (\mathbf{x}^{(t)}, \mathbf{y}^{(t)})$ and using $L_f$-smoothness (Assumption 1) of $\widetilde{F}(\cdot, \cdot)$,

$$-\widetilde{F}(\mathbf{z}^{(t+1)}) \leq -\widetilde{F}(\mathbf{z}^{(t)}) - \left\langle \nabla_{\mathbf{z}} \widetilde{F}(\mathbf{z}^{(t)}), \mathbf{z}^{(t+1)} - \mathbf{z}^{(t)} \right\rangle + \frac{L_f}{2} \left\| \mathbf{z}^{(t+1)} - \mathbf{z}^{(t)} \right\|^2$$

$$= -\widetilde{F}(\mathbf{z}^{(t)}) - \left\langle \nabla_y \widetilde{F}(\mathbf{x}^{(t)}, \mathbf{y}^{(t)}), \mathbf{y}^{(t+1)} - \mathbf{y}^{(t)} \right\rangle + \frac{L_f}{2} \left\| \mathbf{y}^{(t+1)} - \mathbf{y}^{(t)} \right\|^2$$

$$- \left\langle \nabla_x \widetilde{F}(\mathbf{x}^{(t)}, \mathbf{y}^{(t)}), \mathbf{x}^{(t+1)} - \mathbf{x}^{(t)} \right\rangle + \frac{L_f}{2} \left\| \mathbf{x}^{(t+1)} - \mathbf{x}^{(t)} \right\|^2$$

$$\Rightarrow -\mathbb{E}_t \left[ \widetilde{F}(\mathbf{x}^{(t+1)}, \mathbf{y}^{(t+1)}) \right] \leq -\widetilde{F}(\mathbf{x}^{(t)}, \mathbf{y}^{(t)}) - \tau_{\text{eff}} \gamma_y^s \mathbb{E}_t \left\langle \nabla_y \widetilde{F}(\mathbf{x}^{(t)}, \mathbf{y}^{(t)}), \sum_{i=1}^n w_i \mathbf{h}_{\mathbf{y},i}^{(t)} \right\rangle + \frac{\tau_{\text{eff}}^2 [\gamma_y^s]^2 L_f}{2} \mathbb{E}_t \left\| \mathbf{g}_{\mathbf{y}}^{(t)} \right\|^2$$

$$+ \tau_{\text{eff}} \gamma_x^s \mathbb{E}_t \left\langle \nabla_x \widetilde{F}(\mathbf{x}^{(t)}, \mathbf{y}^{(t)}), \sum_{i=1}^n w_i \mathbf{h}_{\mathbf{x},i}^{(t)} \right\rangle + \frac{\tau_{\text{eff}}^2 [\gamma_x^s]^2 L_f}{2} \mathbb{E}_t \left\| \mathbf{g}_{\mathbf{x}}^{(t)} \right\|^2.$$

Rearranging the terms a bit, we get

$$
\mathbb{E}_t \left[ \widetilde{F}(\mathbf{x}^{(t)}, \mathbf{y}^{(t)}) - \widetilde{F}(\mathbf{x}^{(t+1)}, \mathbf{y}^{(t+1)}) \right]
$$

$$
\leq -\frac{\tau_{\text{eff}} \gamma_y^s}{2} \mathbb{E}_t \left[ \left\| \nabla_y \widetilde{F}(\mathbf{x}^{(t)}, \mathbf{y}^{(t)}) \right\|^2 + \left\| \sum_{i=1}^n w_i \mathbf{h}_{\mathbf{y},i}^{(t)} \right\|^2 - \left\| \nabla_y \widetilde{F}(\mathbf{x}^{(t)}, \mathbf{y}^{(t)}) - \sum_{i=1}^n w_i \mathbf{h}_{\mathbf{y},i}^{(t)} \right\|^2 \right] + \frac{\tau_{\text{eff}}^2 [\gamma_y^s]^2 L_f}{2} \mathbb{E}_t \left\| \mathbf{g}_{\mathbf{y}}^{(t)} \right\|^2
$$

$$
+ \frac{\tau_{\text{eff}} \gamma_x^s}{2} \mathbb{E}_t \left[ \left\| \nabla_x \widetilde{F}(\mathbf{x}^{(t)}, \mathbf{y}^{(t)}) \right\|^2 + \left\| \sum_{i=1}^n w_i \mathbf{h}_{\mathbf{x},i}^{(t)} \right\|^2 - \left\| \nabla_x \widetilde{F}(\mathbf{x}^{(t)}, \mathbf{y}^{(t)}) - \sum_{i=1}^n w_i \mathbf{h}_{\mathbf{x},i}^{(t)} \right\|^2 \right] + \frac{\tau_{\text{eff}}^2 [\gamma_x^s]^2 L_f}{2} \mathbb{E}_t \left\| \mathbf{g}_{\mathbf{x}}^{(t)} \right\|^2 .
$$

$$(34)$$

Next, we bound the individual terms in (34). Using quadratic growth of $\mu$-SC functions (Lemma A.5),

$$
\left\| \nabla_y \widetilde{F}(\mathbf{x}^{(t)}, \mathbf{y}^{(t)}) \right\|^2 \geq 2\mu \left[ \widetilde{\Phi}(\mathbf{x}^{(t)}) - \widetilde{F}(\mathbf{x}^{(t)}, \mathbf{y}^{(t)}) \right] .
$$

To bound $\left\| \nabla_y \widetilde{F}(\mathbf{x}^{(t)}, \mathbf{y}^{(t)}) - \sum_{i=1}^n w_i \mathbf{h}_{\mathbf{y},i}^{(t)} \right\|^2$, we use similar reasoning as in (23).

$$
\mathbb{E}_t \left\| \nabla_y \widetilde{F}(\mathbf{x}^{(t)}, \mathbf{y}^{(t)}) - \sum_{i=1}^n w_i \mathbf{h}_{\mathbf{y},i}^{(t)} \right\|^2 = \left\| \sum_{i=1}^n w_i \left( \nabla_y f_i(\mathbf{x}^{(t)}, \mathbf{y}^{(t)}) - \mathbf{h}_{\mathbf{y},i}^{(t)} \right) \right\|^2
$$

$$
\leq L_f^2 \sum_{i=1}^n \frac{w_i}{\|\boldsymbol{a}_i\|_1} \sum_{k=0}^{\tau_i - 1} a_i^{(k)}(\tau_i) \Delta_{\mathbf{x},\mathbf{y}}^{(t,k)}(i), \qquad (35)
$$

Substituting the bounds in (35) and Lemma B.6 into (34) we get

$$
\mathbb{E}_t \left[ \widetilde{F}(\mathbf{x}^{(t)}, \mathbf{y}^{(t)}) - \widetilde{F}(\mathbf{x}^{(t+1)}, \mathbf{y}^{(t+1)}) \right]
$$

$$
\leq -\tau_{\text{eff}} \mu \gamma_y^s \left( 1 - \frac{2\kappa^2 \gamma_x^s}{\gamma_y^s} \right) \left[ \widetilde{\Phi}(\mathbf{x}^{(t)}) - \widetilde{F}(\mathbf{x}^{(t)}, \mathbf{y}^{(t)}) \right] - \frac{\tau_{\text{eff}} \gamma_y^s}{2} \mathbb{E}_t \left\| \sum_{i=1}^n w_i \mathbf{h}_{\mathbf{y},i}^{(t)} \right\|^2
$$

$$
+ \tau_{\text{eff}} \gamma_x^s \left\| \nabla \widetilde{\Phi}(\mathbf{x}^{(t)}) \right\|^2 + \frac{\tau_{\text{eff}} \gamma_x^s}{2} \mathbb{E}_t \left\| \sum_{i=1}^n w_i \mathbf{h}_{\mathbf{x},i}^{(t)} \right\|^2 + \frac{\tau_{\text{eff}}^2 L_f}{2} \left[ [\gamma_y^s]^2 \mathbb{E}_t \left\| \mathbf{g}_{\mathbf{y}}^{(t)} \right\|^2 + [\gamma_x^s]^2 \mathbb{E}_t \left\| \mathbf{g}_{\mathbf{x}}^{(t)} \right\|^2 \right]
$$

$$
+ \frac{\tau_{\text{eff}} \gamma_y^s L_f^2}{2} \sum_{i=1}^n \frac{w_i}{\|\boldsymbol{a}_i\|_1} \sum_{k=0}^{\tau_i - 1} a_i^{(k)}(\tau_i) \Delta_{\mathbf{x},\mathbf{y}}^{(t,k)}(i). \qquad (36)
$$

$\mathbb{E}_t \left\| \mathbf{g}_{\mathbf{x}}^{(t)} \right\|^2$ is already bounded in Lemma B.2. We further substitute the bound on $\mathbb{E}_t \left\| \nabla_x f_i(\mathbf{x}_i^{(t,k)}, \mathbf{y}_i^{(t,k)}) \right\|^2$ from Lemma B.7 to get

$$
\mathbb{E}_t \left\| \mathbf{g}_{\mathbf{x}}^{(t)} \right\|^2 \leq \frac{n(P-1)}{P(n-1)} \mathbb{E}_t \left\| \sum_{i=1}^n w_i \mathbf{h}_{\mathbf{x},i}^{(t)} \right\|^2 + \frac{\sigma_L^2 n}{P} \sum_{i=1}^n \frac{w_i^2 \|\boldsymbol{a}_i\|_2^2}{\|\boldsymbol{a}_i\|_1^2} + \frac{2\sigma_G^2}{P} \left( \frac{n(n-P)}{n-1} \max_i w_i \right)
$$

$$
+ \frac{\beta_L^2 n}{P} \left[ 2 L_f^2 \sum_{i=1}^n \frac{w_i^2}{\|\boldsymbol{a}_i\|_1^2} \sum_{k=0}^{\tau_i - 1} [a_i^{(k)}(\tau_i)]^2 \Delta_{\mathbf{x},\mathbf{y}}^{(t,k)}(i) + 4\beta_G^2 \left( \max_i \frac{w_i \|\boldsymbol{a}_i\|_2^2}{\|\boldsymbol{a}_i\|_1^2} \right) \left\| \nabla_x \widetilde{\Phi}(\mathbf{x}^{(t)}) \right\|^2 \right]
$$

$$
+ \frac{\beta_L^2 n}{P} 2 \max_i \frac{w_i \|\boldsymbol{a}_i\|_2^2}{\|\boldsymbol{a}_i\|_1^2} \left[ \sigma_G^2 + 4\beta_G^2 L_f \kappa \left[ \widetilde{\Phi}(\mathbf{x}^{(t)}) - \widetilde{F}(\mathbf{x}^{(t)}, \mathbf{y}^{(t)}) \right] \right]
$$

$$
+ \frac{n(n-P)}{n-1} \frac{2 L_f^2}{P} \sum_{i=1}^n \frac{w_i^2}{\|\boldsymbol{a}_i\|_1} \sum_{k=0}^{\tau_i - 1} a_i^{(k)}(\tau_i) \Delta_{\mathbf{x},\mathbf{y}}^{(t,k)}(i)
$$

$$
+ \left( \frac{n(n-P)}{n-1} \max_i w_i \right) \frac{2\beta_G^2}{P} \left[ 4 L_f \kappa \left[ \widetilde{\Phi}(\mathbf{x}^{(t)}) - \widetilde{F}(\mathbf{x}^{(t)}, \mathbf{y}^{(t)}) \right] + 2 \left\| \nabla_x \widetilde{\Phi}(\mathbf{x}^{(t)}) \right\|^2 \right]
$$

$$\leq \frac{n(P-1)}{P(n-1)}\mathbb{E}_t\left\|\sum_{i=1}^n w_i\mathbf{h}_{\mathbf{x},i}^{(t)}\right\|^2 + \frac{\sigma_L^2 n}{P}\sum_{i=1}^n \frac{w_i^2\|\boldsymbol{a}_i\|_2^2}{\|\boldsymbol{a}_i\|_1^2} + \frac{2n\sigma_G^2}{P}\left(\frac{n-P}{n-1}\max_i w_i + \beta_L^2\max_i\frac{w_i\|\boldsymbol{a}_i\|_2^2}{\|\boldsymbol{a}_i\|_1^2}\right)$$

$$+ \left(\frac{n-P}{n-1}\max_i w_i + \beta_L^2\max_{i,k}\frac{w_i a_i^{(k)}(\tau_i)}{\|\boldsymbol{a}_i\|_1}\right)\frac{2nL_f^2}{P}\sum_{i=1}^n\frac{w_i}{\|\boldsymbol{a}_i\|_1}\sum_{k=0}^{\tau_i-1}a_i^{(k)}(\tau_i)\Delta_{\mathbf{x},\mathbf{y}}^{(t,k)}(i)$$

$$+ \frac{4n\beta_G^2}{P}\left(\frac{n-P}{n-1}\max_i w_i + \beta_L^2\max_i\frac{w_i\|\boldsymbol{a}_i\|_2^2}{\|\boldsymbol{a}_i\|_1^2}\right)\left\|\nabla_x\widetilde{\Phi}(\mathbf{x}^{(t)})\right\|^2$$

$$+ \frac{8n\beta_G^2 L_f\kappa}{P}\left(\frac{n-P}{n-1}\max_i w_i + \beta_L^2\max_i\frac{w_i\|\boldsymbol{a}_i\|_2^2}{\|\boldsymbol{a}_i\|_1^2}\right)\left[\widetilde{\Phi}(\mathbf{x}^{(t)}) - \widetilde{F}(\mathbf{x}^{(t)},\mathbf{y}^{(t)})\right]. \tag{37}$$

Similarly, we can bound $\mathbb{E}_t\left\|\mathbf{g}_{\mathbf{y}}^{(t)}\right\|^2$ to get

$$\mathbb{E}_t\left\|\mathbf{g}_{\mathbf{y}}^{(t)}\right\|^2 \leq \frac{n(P-1)}{P(n-1)}\mathbb{E}_t\left\|\sum_{i=1}^n w_i\mathbf{h}_{\mathbf{y},i}^{(t)}\right\|^2 + \frac{\sigma_L^2 n}{P}\sum_{i=1}^n \frac{w_i^2\|\boldsymbol{a}_i\|_2^2}{\|\boldsymbol{a}_i\|_1^2} + \frac{2\sigma_G^2 n}{P}\left(\frac{n-P}{n-1}\max_i w_i + \beta_L^2\max_i\frac{w_i\|\boldsymbol{a}_i\|_2^2}{\|\boldsymbol{a}_i\|_1^2}\right)$$

$$+ \left(\frac{n-P}{n-1}\max_i w_i + \beta_L^2\max_{i,k}\frac{w_i a_i^{(k)}(\tau_i)}{\|\boldsymbol{a}_i\|_1}\right)\frac{2nL_f^2}{P}\sum_{i=1}^n\frac{w_i}{\|\boldsymbol{a}_i\|_1}\sum_{k=0}^{\tau_i-1}a_i^{(k)}(\tau_i)\Delta_{\mathbf{x},\mathbf{y}}^{(t,k)}(i)$$

$$+ \frac{4\beta_G^2 L_f n}{P}\left(\frac{n-P}{n-1}\max_i w_i + \beta_L^2\max_i\frac{w_i\|\boldsymbol{a}_i\|_2^2}{\|\boldsymbol{a}_i\|_1^2}\right)\left[\widetilde{\Phi}(\mathbf{x}^{(t)}) - \widetilde{F}(\mathbf{x}^{(t)},\mathbf{y}^{(t)})\right]. \tag{38}$$

Substituting (37), (38) in (36), and again substituting the resulting bound in (33), we get

$$\mathbb{E}_t\left[\widetilde{\Phi}(\mathbf{x}^{(t+1)}) - \widetilde{F}(\mathbf{x}^{(t+1)},\mathbf{y}^{(t+1)})\right]$$

$$\overset{(a)}{\leq}\left(1 - \tau_{\text{eff}}\mu\gamma_y^s\left(1 - \frac{\kappa^2\gamma_x^s}{\gamma_y^s}\left[2+\frac{9}{4}\right] - \tau_{\text{eff}}\kappa\gamma_y^s\frac{4n\beta_G^2 L_f}{P}\left(\frac{n-P}{n-1}\max_i w_i + \beta_L^2\max_i\frac{w_i\|\boldsymbol{a}_i\|_2^2}{\|\boldsymbol{a}_i\|_1^2}\right)\right)\right)\left[\widetilde{\Phi}(\mathbf{x}^{(t)}) - \widetilde{F}(\mathbf{x}^{(t)},\mathbf{y}^{(t)})\right]$$

$$+ \tau_{\text{eff}}\gamma_x^s\left(\frac{9}{16} + \tau_{\text{eff}}L_f\gamma_x^s\frac{2n\beta_G^2}{P}\left(\frac{n-P}{n-1}\max_i w_i + \beta_L^2\max_i\frac{w_i\|\boldsymbol{a}_i\|_2^2}{\|\boldsymbol{a}_i\|_1^2}\right)\right)\left\|\nabla\widetilde{\Phi}(\mathbf{x}^{(t)})\right\|^2$$

$$+ \frac{\tau_{\text{eff}}^2[\gamma_x^s]^2 L_\Phi}{2}\frac{n}{P}\left[\sigma_L^2\sum_{i=1}^n\frac{w_i^2\|\boldsymbol{a}_i\|_2^2}{\|\boldsymbol{a}_i\|_1^2} + \sigma_G^2\left(2(\max_i w_i)\frac{n-P}{n-1} + 2\beta_L^2\max_i\frac{w_i\|\boldsymbol{a}_i\|_2^2}{\|\boldsymbol{a}_i\|_1^2}\right)\right]$$

$$+ \frac{\tau_{\text{eff}}^2 L_f}{2}\left([\gamma_y^s]^2 + [\gamma_x^s]^2\right)\left[\frac{\sigma_L^2 n}{P}\sum_{i=1}^n\frac{w_i^2\|\boldsymbol{a}_i\|_2^2}{\|\boldsymbol{a}_i\|_1^2} + \frac{2n\sigma_G^2}{P}\left(\frac{n-P}{n-1}\max_i w_i + \beta_L^2\max_i\frac{w_i\|\boldsymbol{a}_i\|_2^2}{\|\boldsymbol{a}_i\|_1^2}\right)\right]$$

$$- \frac{\tau_{\text{eff}}\gamma_x^s}{2}\left(1 - \frac{n(P-1)}{P(n-1)}\tau_{\text{eff}}\gamma_x^s L_\Phi - 1 - \frac{n(P-1)}{P(n-1)}\tau_{\text{eff}}\gamma_x^s L_f\right)\mathbb{E}_t\left\|\sum_{i=1}^n w_i\mathbf{h}_{\mathbf{x},i}^{(t)}\right\|^2$$

$$- \frac{\tau_{\text{eff}}\gamma_y^s}{2}\left(1 - \frac{n(P-1)}{P(n-1)}\tau_{\text{eff}}\gamma_y^s L_f\right)\mathbb{E}_t\left\|\sum_{i=1}^n w_i\mathbf{h}_{\mathbf{y},i}^{(t)}\right\|^2$$

$$+ \tau_{\text{eff}}\left(\frac{3\gamma_x^s}{2} + \frac{3\gamma_y^s}{4}\right)L_f^2\sum_{i=1}^n\frac{w_i}{\|\boldsymbol{a}_i\|_1}\sum_{k=0}^{\tau_i-1}a_i^{(k)}(\tau_i)\Delta_{\mathbf{x},\mathbf{y}}^{(t,k)}(i)$$

$$\overset{(b)}{\leq}\left(1 - \tau_{\text{eff}}\mu\gamma_y^s\left(1 - \frac{\kappa^2\gamma_x^s}{\gamma_y^s}\left[2+\frac{9}{4}\right] - \tau_{\text{eff}}\kappa\gamma_y^s\frac{4n\beta_G^2 L_f}{P}\left(\frac{n-P}{n-1}\max_i w_i + \beta_L^2\max_i\frac{w_i\|\boldsymbol{a}_i\|_2^2}{\|\boldsymbol{a}_i\|_1^2}\right)\right)\right)\left[\widetilde{\Phi}(\mathbf{x}^{(t)}) - \widetilde{F}(\mathbf{x}^{(t)},\mathbf{y}^{(t)})\right]$$

$$+ \tau_{\text{eff}}\gamma_x^s\left(\frac{9}{16} + \tau_{\text{eff}}L_f\gamma_x^s\frac{2n\beta_G^2}{P}\left(\frac{n-P}{n-1}\max_i w_i + \beta_L^2\max_i\frac{w_i\|\boldsymbol{a}_i\|_2^2}{\|\boldsymbol{a}_i\|_1^2}\right)\right)\left\|\nabla\widetilde{\Phi}(\mathbf{x}^{(t)})\right\|^2$$

$$+ \frac{\tau_{\text{eff}}^2[\gamma_x^s]^2}{2}\frac{n(P-1)}{P(n-1)}(L_\Phi + L_f)\mathbb{E}_t\left\|\sum_{i=1}^n w_i\mathbf{h}_{\mathbf{x},i}^{(t)}\right\|^2$$

$$- \frac{\tau_{\text{eff}}\gamma_x^s}{2}\left(1 - \frac{n(P-1)}{P(n-1)}\tau_{\text{eff}}\frac{[\gamma_y^s]^2}{\gamma_x^s}L_f\right)\mathbb{E}_t\left\|\sum_{i=1}^n w_i\mathbf{h}_{\mathbf{y},i}^{(t)}\right\|^2$$

$$+ \frac{\tau_{\text{eff}}^2[\gamma_x^s]^2 L_\Phi}{2}\frac{n}{P}\left[\sigma_L^2\sum_{i=1}^n\frac{w_i^2\|\boldsymbol{a}_i\|_2^2}{\|\boldsymbol{a}_i\|_1^2} + \sigma_G^2\left(2(\max_i w_i)\frac{n-P}{n-1} + 2\beta_L^2\max_i\frac{w_i\|\boldsymbol{a}_i\|_2^2}{\|\boldsymbol{a}_i\|_1^2}\right)\right]$$

$$+ \frac{\tau_{\text{eff}}^2 L_f}{2}\left([\gamma_y^s]^2 + [\gamma_x^s]^2\right)\left[\frac{\sigma_L^2 n}{P}\sum_{i=1}^n\frac{w_i^2\|\boldsymbol{a}_i\|_2^2}{\|\boldsymbol{a}_i\|_1^2} + \frac{2n\sigma_G^2}{P}\left(\frac{n-P}{n-1}\max_i w_i + \beta_L^2\max_i\frac{w_i\|\boldsymbol{a}_i\|_2^2}{\|\boldsymbol{a}_i\|_1^2}\right)\right]$$

$$+ \tau_{\text{eff}}\gamma_y^s\left[2\left([\eta_x^c]^2 + [\eta_y^c]^2\right)L_f^2\sigma_L^2\sum_{i=1}^n w_i\|\boldsymbol{a}_{i,-1}\|_2^2 + 4L_f^2 M_{\mathbf{a}_{-1}}\left([\eta_x^c]^2 + [\eta_y^c]^2\right)\sigma_G^2\right]$$

$$+ \tau_{\text{eff}}\gamma_y^s\left[8L_f^2 M_{\mathbf{a}_{-1}}\beta_G^2[\eta_x^c]^2\left\|\nabla\widetilde{\Phi}(\mathbf{x}^{(t)})\right\|^2 + 8L_f^3 M_{\mathbf{a}_{-1}}\beta_G^2\left(2\kappa[\eta_x^c]^2 + [\eta_y^c]^2\right)\left[\widetilde{\Phi}(\mathbf{x}^{(t)}) - \widetilde{F}(\mathbf{x}^{(t)}, \mathbf{y}^{(t)})\right]\right], \quad (39)$$

where ($a$) follows since

$$\tau_{\text{eff}}L_f\gamma_y^s\left(\frac{n-P}{n-1}\max_i w_i + \beta_L^2\max_{i,k}\frac{w_i a_i^{(k)}(\tau_i)}{\|\boldsymbol{a}_i\|_1}\right)\frac{n}{P} \leq \frac{1}{4},$$

and ($b$) follows by using the bound in Lemma B.4. Next, we simplify the coefficients of different terms in (39).

- Coefficient of $\mathbb{E}\left[\widetilde{\Phi}(\mathbf{x}^{(t)}) - \widetilde{F}(\mathbf{x}^{(t)}, \mathbf{y}^{(t)})\right]$ can be simplified to

$$\left(1 - \tau_{\text{eff}}\mu\gamma_y^s\left(1 - \frac{\kappa^2\gamma_x^s}{\gamma_y^s}\frac{17}{4} - \tau_{\text{eff}}\kappa\gamma_y^s\frac{4n\beta_G^2 L_f}{P}\left(\frac{n-P}{n-1}\max_i w_i + \beta_L^2\max_i\frac{w_i\|\boldsymbol{a}_i\|_2^2}{\|\boldsymbol{a}_i\|_1^2}\right)\right)\right)$$

$$+ \tau_{\text{eff}}\gamma_y^s 8L_f^3 M_{\mathbf{a}_{-1}}\beta_G^2\left(2\kappa[\eta_x^c]^2 + [\eta_y^c]^2\right)$$

$$\leq 1 - \frac{\tau_{\text{eff}}\gamma_y^s\mu}{4}.$$

  using $\gamma_x^s \leq \frac{\gamma_y^s}{17\kappa^2}$, $\tau_{\text{eff}}\gamma_y^s\kappa L_f\beta_G^2\frac{n}{P}\max\left\{\beta_L^2\max_i\frac{w_i\|\boldsymbol{a}_i\|_2^2}{\|\boldsymbol{a}_i\|_1^2}, \frac{n-P}{n-1}\max_i w_i\right\} \leq \frac{1}{64}$, $\kappa L_f\beta_G\eta_x^c \leq \frac{1}{16\sqrt{2M_{\mathbf{a}_{-1}}}}$ and $L_f\beta_G\eta_y^c \leq \frac{1}{16\sqrt{\kappa M_{\mathbf{a}_{-1}}}}$.

- Coefficient of $\mathbb{E}\left\|\sum_{i=1}^n w_i\mathbf{h}_{\mathbf{x},i}^{(t)}\right\|^2$ can be simplified to

$$\frac{\tau_{\text{eff}}^2[\gamma_x^s]^2}{2}(L_\Phi + L_f)\frac{n(P-1)}{P(n-1)} \leq \tau_{\text{eff}}^2[\gamma_x^s]^2 L_\Phi\frac{n(P-1)}{P(n-1)}. \qquad (\because L_f \leq L_\Phi)$$

- Coefficient of $\mathbb{E}\left\|\nabla\widetilde{\Phi}(\mathbf{x}^{(t)})\right\|^2$ can be simplified to

$$\tau_{\text{eff}}\gamma_x^s\left(\frac{9}{16} + \tau_{\text{eff}}L_f\gamma_x^s\frac{2n\beta_G^2}{P}\left(\frac{n-P}{n-1}\max_i w_i + \beta_L^2\max_i\frac{w_i\|\boldsymbol{a}_i\|_2^2}{\|\boldsymbol{a}_i\|_1^2}\right)\right) + \tau_{\text{eff}}\gamma_y^s 8L_f^2 M_{\mathbf{a}_{-1}}\beta_G^2[\eta_x^c]^2$$

$$\leq \tau_{\text{eff}}\gamma_x^s\left(\frac{9}{16} + \frac{4\gamma_x^s}{\gamma_y^s}\frac{1}{64\kappa}\right) + \frac{\tau_{\text{eff}}\gamma_y^s}{64\kappa^2}$$

$$\leq \frac{\tau_{\text{eff}}\gamma_y^s}{48\kappa^2}. \qquad (\frac{\gamma_x^s}{\gamma_y^s} \leq \frac{1}{81\kappa^2})$$

  using $\gamma_x^s \leq \frac{\gamma_y^s}{81\kappa^2}$, $\eta_x^c L_f\beta_G \leq \frac{1}{64\kappa\sqrt{M_{\mathbf{a}_{-1}}}}$ and $\tau_{\text{eff}}\gamma_y^s\kappa L_f\beta_G^2\frac{n}{P}\max\left\{\frac{n-P}{n-1}\max_i w_i, \beta_L^2\max_i\frac{w_i\|\boldsymbol{a}_i\|_2^2}{\|\boldsymbol{a}_i\|_1^2}\right\} \leq \frac{1}{64}$.

- Coefficient of $\sigma_L^2$ can be simplified to

$$\frac{\tau_{\text{eff}}^2[\gamma_x^s]^2 L_\Phi}{2} \frac{n}{P} \left[ \sum_{i=1}^n \frac{w_i^2 \|\boldsymbol{a}_i\|_2^2}{\|\boldsymbol{a}_i\|_1^2} \right] + \frac{\tau_{\text{eff}}^2 L_f}{2} \left([\gamma_y^s]^2 + [\gamma_x^s]^2\right) \left[ \frac{n}{P} \sum_{i=1}^n \frac{w_i^2 \|\boldsymbol{a}_i\|_2^2}{\|\boldsymbol{a}_i\|_1^2} \right]$$

$$+ \tau_{\text{eff}} \gamma_y^s \left[ 2\left([\eta_x^c]^2 + [\eta_y^c]^2\right) L_f^2 \sum_{i=1}^n w_i \|\boldsymbol{a}_{i,-1}\|_2^2 \right]$$

$$\leq \tau_{\text{eff}}^2 \left([\gamma_y^s]^2 L_f + [\gamma_x^s]^2 L_\Phi\right) \frac{n}{P} \sum_{i=1}^n \frac{w_i^2 \|\boldsymbol{a}_i\|_2^2}{\|\boldsymbol{a}_i\|_1^2} + 2\tau_{\text{eff}} \gamma_y^s \left([\eta_x^c]^2 + [\eta_y^c]^2\right) L_f^2 \sum_{i=1}^n w_i \|\boldsymbol{a}_{i,-1}\|_2^2.$$

- Coefficient of $\sigma_G^2$ can be simplified to

$$\frac{\tau_{\text{eff}}^2[\gamma_x^s]^2 L_\Phi}{2} \frac{n}{P} \left[ 2(\max_i w_i)\frac{n-P}{n-1} + 2\beta_L^2 \max_i \frac{w_i\|\boldsymbol{a}_i\|_2^2}{\|\boldsymbol{a}_i\|_1^2} \right]$$

$$+ \frac{\tau_{\text{eff}}^2 L_f}{2} \left([\gamma_y^s]^2 + [\gamma_x^s]^2\right) \frac{2n}{P} \left( \frac{n-P}{n-1} \max_i w_i + \beta_L^2 \max_i \frac{w_i\|\boldsymbol{a}_i\|_2^2}{\|\boldsymbol{a}_i\|_1^2} \right)$$

$$+ \tau_{\text{eff}} \gamma_y^s 4 L_f^2 M_{\mathbf{a}_{-1}} \left([\eta_x^c]^2 + [\eta_y^c]^2\right)$$

$$\leq \tau_{\text{eff}}^2 \left([\gamma_y^s]^2 L_f + [\gamma_x^s]^2 L_\Phi\right) \frac{2n}{P} \left( \frac{n-P}{n-1} \max_i w_i + \beta_L^2 \max_i \frac{w_i\|\boldsymbol{a}_i\|_2^2}{\|\boldsymbol{a}_i\|_1^2} \right) + 4\tau_{\text{eff}} \gamma_y^s \left([\eta_x^c]^2 + [\eta_y^c]^2\right) L_f^2 M_{\mathbf{a}_{-1}}.$$

Substituting these simplified coefficients in (39), and rearranging the terms, we get

$$\mathbb{E}_t \left[ \widetilde{\Phi}(\mathbf{x}^{(t+1)}) - \widetilde{F}(\mathbf{x}^{(t+1)}, \mathbf{y}^{(t+1)}) \right]$$

$$\leq \left(1 - \frac{\tau_{\text{eff}} \gamma_y^s \mu}{4}\right) \left[ \widetilde{\Phi}(\mathbf{x}^{(t)}) - \widetilde{F}(\mathbf{x}^{(t)}, \mathbf{y}^{(t)}) \right] + \frac{\tau_{\text{eff}} \gamma_y^s}{48\kappa^2} \left\| \nabla\widetilde{\Phi}(\mathbf{x}^{(t)}) \right\|^2 + \tau_{\text{eff}}^2[\gamma_x^s]^2 L_\Phi \frac{n(P-1)}{P(n-1)} \mathbb{E}_t \left\| \sum_{i=1}^n w_i \mathbf{h}_{\mathbf{x},i}^{(t)} \right\|^2$$

$$+ \tau_{\text{eff}}^2 \left([\gamma_y^s]^2 L_f + [\gamma_x^s]^2 L_\Phi\right) \frac{n}{P} \left[ \sigma_L^2 \sum_{i=1}^n \frac{w_i^2 \|\boldsymbol{a}_i\|_2^2}{\|\boldsymbol{a}_i\|_1^2} + 2\sigma_G^2 \left( \frac{n-P}{n-1} \max_i w_i + \beta_L^2 \max_i \frac{w_i\|\boldsymbol{a}_i\|_2^2}{\|\boldsymbol{a}_i\|_1^2} \right) \right]$$

$$+ 2\tau_{\text{eff}} \gamma_y^s \left([\eta_x^c]^2 + [\eta_y^c]^2\right) L_f^2 \left[ \sigma_L^2 \sum_{i=1}^n w_i \|\boldsymbol{a}_{i,-1}\|_2^2 + 2\sigma_G^2 M_{\mathbf{a}_{-1}} \right]. \tag{40}$$

Summing both sides of (40) over $t = 0, \ldots, T-1$ and rearranging the terms, we get

$$\frac{1}{T} \sum_{t=0}^{T-1} \mathbb{E} \left[ \widetilde{\Phi}(\mathbf{x}^{(t)}) - \widetilde{F}(\mathbf{x}^{(t)}, \mathbf{y}^{(t)}) \right]$$

$$\leq \frac{4}{\tau_{\text{eff}} \gamma_y^s \mu} \left[ \frac{\widetilde{\Phi}(\mathbf{x}^{(0)}) - \widetilde{F}(\mathbf{x}^{(0)}, \mathbf{y}^{(0)})}{T} - \frac{\mathbb{E}\left(\widetilde{\Phi}(\mathbf{x}^{(T)}) - \widetilde{F}(\mathbf{x}^{(T)}, \mathbf{y}^{(T)})\right)}{T} \right] + \frac{1}{12\mu\kappa^2} \frac{1}{T} \sum_{t=0}^{T-1} \mathbb{E} \left\| \nabla\widetilde{\Phi}(\mathbf{x}^{(t)}) \right\|^2$$

$$+ \frac{4\tau_{\text{eff}}[\gamma_x^s]^2 L_\Phi}{\gamma_y^s \mu} \frac{n(P-1)}{P(n-1)} \mathbb{E} \left\| \sum_{i=1}^n w_i \mathbf{h}_{\mathbf{x},i}^{(t)} \right\|^2 + 8\kappa L_f \left([\eta_x^c]^2 + [\eta_y^c]^2\right) \left[ \sigma_L^2 \sum_{i=1}^n w_i \|\boldsymbol{a}_{i,-1}\|_2^2 + 2\sigma_G^2 M_{\mathbf{a}_{-1}} \right]$$

$$+ 8\tau_{\text{eff}} \gamma_y^s \kappa \frac{n}{P} \left[ \sigma_L^2 \sum_{i=1}^n \frac{w_i^2 \|\boldsymbol{a}_i\|_2^2}{\|\boldsymbol{a}_i\|_1^2} + 2\sigma_G^2 \left( \frac{n-P}{n-1} \max_i w_i + \beta_L^2 \max_i \frac{w_i\|\boldsymbol{a}_i\|_2^2}{\|\boldsymbol{a}_i\|_1^2} \right) \right], \tag{41}$$

where, we use $[\gamma_y^s]^2 L_f \geq [\gamma_x^s]^2 L_\Phi$. This concludes the proof. □

### B.4 Auxiliary Lemmas

**Lemma B.6.** *If the local client function $f_i(\mathbf{x}, \cdot)$ satisfy Assumptions 1, 5 ($L_f$-smoothness and $\mu$-strong concavity in $\mathbf{y}$), then the function $F$ satisfies*

$$\left\|\nabla_x \widetilde{F}(\mathbf{x}, \mathbf{y})\right\|^2 \le 2\left\|\nabla \widetilde{\Phi}(\mathbf{x})\right\|^2 + \frac{4L_f^2}{\mu}\left[\widetilde{\Phi}(\mathbf{x}) - \widetilde{F}(\mathbf{x}, \mathbf{y})\right].$$

*Proof.*

$$
\begin{aligned}
\left\|\nabla_x \widetilde{F}(\mathbf{x}, \mathbf{y})\right\|^2 &\le 2\left\|\nabla \widetilde{\Phi}(\mathbf{x})\right\|^2 + 2\left\|\nabla_x F(\mathbf{x}, \mathbf{y}) - \nabla \widetilde{\Phi}(\mathbf{x})\right\|^2 \\
&\le 2\left\|\nabla \widetilde{\Phi}(\mathbf{x})\right\|^2 + 2L_f^2\left\|\widetilde{\mathbf{y}}^*(\mathbf{x}) - \mathbf{y}\right\|^2 && (L_f\text{-smoothness (Assumption 1)}) \\
&\le 2\left\|\nabla \widetilde{\Phi}(\mathbf{x})\right\|^2 + \frac{4L_f^2}{\mu}\left[\widetilde{\Phi}(\mathbf{x}) - \widetilde{F}(\mathbf{x}, \mathbf{y})\right]. && (\text{Assumption 5})
\end{aligned}
$$

$\square$

**Lemma B.7.** *If the local client function $f_i(\mathbf{x}, \cdot)$ satisfy Assumptions 1, 4 and 5, then the iterates $\{\mathbf{x}_i^{(t,k)}, \mathbf{y}_i^{(t,k)}\}_{i,(t,k)}$ generated by Algorithm 1 satisfy*

$$
\begin{aligned}
&\sum_{i=1}^n \frac{w_i^2}{\|\boldsymbol{a}_i\|_1^2} \sum_{k=0}^{\tau_i-1} [a_i^{(k)}(\tau_i)]^2 \mathbb{E}\left\|\nabla_x f_i(\mathbf{x}_i^{(t,k)}, \mathbf{y}_i^{(t,k)})\right\|^2 \\
&\le 2\sum_{i=1}^n \frac{w_i^2}{\|\boldsymbol{a}_i\|_1^2} \sum_{k=0}^{\tau_i-1} [a_i^{(k)}(\tau_i)]^2 L_f^2 \Delta_{\mathbf{x},\mathbf{y}}^{(t,k)}(i) + 2\sigma_G^2 \left(\max_i \frac{w_i\|\boldsymbol{a}_i\|_2^2}{\|\boldsymbol{a}_i\|_1^2}\right) \\
&\quad + 4\beta_G^2 \left(\max_i \frac{w_i\|\boldsymbol{a}_i\|_2^2}{\|\boldsymbol{a}_i\|_1^2}\right)\left[\frac{2L_f^2}{\mu}\mathbb{E}\left(\widetilde{\Phi}(\mathbf{x}^{(t)}) - \widetilde{F}(\mathbf{x}^{(t)}, \mathbf{y}^{(t)})\right) + \left\|\nabla_x\widetilde{\Phi}(\mathbf{x}^{(t)})\right\|^2\right].
\end{aligned}
$$

*Proof.*

$$
\begin{aligned}
&\sum_{i=1}^n \frac{w_i^2}{\|\boldsymbol{a}_i\|_1^2} \sum_{k=0}^{\tau_i-1} [a_i^{(k)}(\tau_i)]^2 \mathbb{E}\left\|\nabla_x f_i(\mathbf{x}_i^{(t,k)}, \mathbf{y}_i^{(t,k)}) \pm \nabla_x f_i(\mathbf{x}^{(t)}, \mathbf{y}^{(t)})\right\|^2 \\
&\le 2\sum_{i=1}^n \frac{w_i^2}{\|\boldsymbol{a}_i\|_1^2} \sum_{k=0}^{\tau_i-1} [a_i^{(k)}(\tau_i)]^2 L_f^2 \mathbb{E}\left[\left\|\mathbf{x}_i^{(t,k)} - \mathbf{x}^{(t)}\right\|^2 + \left\|\mathbf{y}_i^{(t,k)} - \mathbf{y}^{(t)}\right\|^2\right] + 2\sum_{i=1}^n \frac{w_i^2\|\boldsymbol{a}_i\|_2^2}{\|\boldsymbol{a}_i\|_1^2}\mathbb{E}\left\|\nabla_x f_i(\mathbf{x}^{(t)}, \mathbf{y}^{(t)})\right\|^2 \\
&\hspace{11cm} (L_f\text{-smoothness}) \\
&\le 2\sum_{i=1}^n \frac{w_i^2}{\|\boldsymbol{a}_i\|_1^2} \sum_{k=0}^{\tau_i-1} [a_i^{(k)}(\tau_i)]^2 L_f^2 \Delta_{\mathbf{x},\mathbf{y}}^{(t,k)}(i) + 2\left(\max_i \frac{w_i\|\boldsymbol{a}_i\|_2^2}{\|\boldsymbol{a}_i\|_1^2}\right)\left[\beta_G^2\left\|\nabla_x\widetilde{F}(\mathbf{x}^{(t)}, \mathbf{y}^{(t)})\right\|^2 + \sigma_G^2\right] \\
&\hspace{11cm} (\text{Assumption 4})
\end{aligned}
$$

Using Lemma B.6 gives the result. $\square$

**Lemma B.8.** *If the local client function $f_i(\mathbf{x}, \cdot)$ satisfy Assumptions 1, 4 and 5, then the iterates $\{\mathbf{x}_i^{(t,k)}, \mathbf{y}_i^{(t,k)}\}_{i,(t,k)}$ generated by Algorithm 1 satisfy*

$$
\begin{aligned}
&\sum_{i=1}^n \frac{w_i^2}{\|\boldsymbol{a}_i\|_1^2} \sum_{k=0}^{\tau_i-1} [a_i^{(k)}(\tau_i)]^2 \mathbb{E}\left\|\nabla_y f_i(\mathbf{x}_i^{(t,k)}, \mathbf{y}_i^{(t,k)})\right\|^2 \\
&\le 2\sum_{i=1}^n \frac{w_i^2}{\|\boldsymbol{a}_i\|_1^2} \sum_{k=0}^{\tau_i-1} [a_i^{(k)}(\tau_i)]^2 L_f^2 \Delta_{\mathbf{x},\mathbf{y}}^{(t,k)}(i) + 2\left(\max_i \frac{w_i\|\boldsymbol{a}_i\|_2^2}{\|\boldsymbol{a}_i\|_1^2}\right)\left[\sigma_G^2 + 2\beta_G^2 L_f \mathbb{E}\left(\widetilde{\Phi}(\mathbf{x}^{(t)}) - \widetilde{F}(\mathbf{x}^{(t)}, \mathbf{y}^{(t)})\right)\right].
\end{aligned}
$$

*Proof.* Following closely the proof of Lemma B.7,

$$\sum_{i=1}^{n} \frac{w_i^2}{\|\boldsymbol{a}_i\|_1^2} \sum_{k=0}^{\tau_i-1} [a_i^{(k)}(\tau_i)]^2 \mathbb{E} \left\| \nabla_y f_i(\mathbf{x}_i^{(t,k)}, \mathbf{y}_i^{(t,k)}) \pm \nabla_y f_i(\mathbf{x}^{(t)}, \mathbf{y}^{(t)}) \right\|^2$$

$$\leq 2 \sum_{i=1}^{n} \frac{w_i^2}{\|\boldsymbol{a}_i\|_1^2} \sum_{k=0}^{\tau_i-1} [a_i^{(k)}(\tau_i)]^2 L_f^2 \mathbb{E} \left[ \left\| \mathbf{x}_i^{(t,k)} - \mathbf{x}^{(t)} \right\|^2 + \left\| \mathbf{y}_i^{(t,k)} - \mathbf{y}^{(t)} \right\|^2 \right] + 2 \sum_{i=1}^{n} \frac{w_i^2 \|\boldsymbol{a}_i\|_2^2}{\|\boldsymbol{a}_i\|_1^2} \mathbb{E} \left\| \nabla_y f_i(\mathbf{x}^{(t)}, \mathbf{y}^{(t)}) \right\|^2$$
$$(L_f\text{-smoothness})$$

$$\leq 2 \sum_{i=1}^{n} \frac{w_i^2}{\|\boldsymbol{a}_i\|_1^2} \sum_{k=0}^{\tau_i-1} [a_i^{(k)}(\tau_i)]^2 L_f^2 \Delta_{\mathbf{x},\mathbf{y}}^{(t,k)}(i) + 2 \left( \max_i \frac{w_i \|\boldsymbol{a}_i\|_2^2}{\|\boldsymbol{a}_i\|_1^2} \right) \left[ \beta_G^2 \left\| \nabla_y \widetilde{F}(\mathbf{x}^{(t)}, \mathbf{y}^{(t)}) \right\|^2 + \sigma_G^2 \right]$$
$$(\text{Assumption } 4)$$

$$\leq 2 \sum_{i=1}^{n} \frac{w_i^2}{\|\boldsymbol{a}_i\|_1^2} \sum_{k=0}^{\tau_i-1} [a_i^{(k)}(\tau_i)]^2 L_f^2 \Delta_{\mathbf{x},\mathbf{y}}^{(t,k)}(i) + 2 \left( \max_i \frac{w_i \|\boldsymbol{a}_i\|_2^2}{\|\boldsymbol{a}_i\|_1^2} \right) \left[ \sigma_G^2 + 2\beta_G^2 L_f \mathbb{E} \left( \widetilde{\Phi}(\mathbf{x}^{(t)}) - \widetilde{F}(\mathbf{x}^{(t)}, \mathbf{y}^{(t)}) \right) \right].$$

where the final inequality follows from smoothness and concavity of $F$ in $\mathbf{y}$. $\qquad\square$

## B.5   Convergence under Polyak Łojasiewicz (PL) Condition

In case the global function satisfies Assumption 6, the results in this section follow with minor modifications. The crucial difference is that Lemma A.6 no longer holds. Lemma B.2 and Lemma B.3 follow exactly. The statement of Lemma B.4 needs some modification, since we use Lemma A.6 in the proof.

**Lemma B.9.** *Suppose the local loss functions $\{f_i\}$ satisfy Assumptions 1, 4, 6, and the stochastic oracles for the local functions satisfy Assumption 3. Under the conditions of Lemma B.9, the iterates $\{\mathbf{x}_i^{(t)}, \mathbf{y}_i^{(t)}\}$ generated by Algorithm 1 satisfy*

$$L_f^2 \sum_{i=1}^{n} \frac{p_i}{\|\boldsymbol{a}_i\|_1} \sum_{k=0}^{\tau_i-1} a_i^{(k)}(\tau_i) \Delta_{\mathbf{x},\mathbf{y}}^{(t,k)}(i) \leq 2 \left([\eta_x^c]^2 + [\eta_y^c]^2\right) L_f^2 \sigma_L^2 \sum_{i=1}^{n} p_i \|\boldsymbol{a}_{i,-1}\|_2^2 + 4L_f^2 M_{\mathbf{a}_{-1}} \left([\eta_x^c]^2 + [\eta_y^c]^2\right) \sigma_G^2$$
$$+ 8L_f^2 M_{\mathbf{a}_{-1}} \beta_G^2 [\eta_x^c]^2 \mathbb{E} \left\| \nabla \widetilde{\Phi}(\mathbf{x}^{(t)}) \right\|^2 + 8\kappa L_f^3 M_{\mathbf{a}_{-1}} \beta_G^2 \left(2[\eta_x^c]^2 + [\eta_y^c]^2\right) \mathbb{E} \left[ \widetilde{\Phi}(\mathbf{x}^{(t)}) - \widetilde{F}(\mathbf{x}^{(t)}, \mathbf{y}^{(t)}) \right],$$

*where $M_{\mathbf{a}_{-1}} \triangleq \max_i \left( \|\boldsymbol{a}_{i,-1}\|_1^2 + \beta_L^2 \|\boldsymbol{a}_{i,-1}\|_2^2 \right)$.*

The bound in Lemma B.8 also changes to

$$\sum_{i=1}^{n} \frac{w_i^2}{\|\boldsymbol{a}_i\|_1^2} \sum_{k=0}^{\tau_i-1} [a_i^{(k)}(\tau_i)]^2 \mathbb{E} \left\| \nabla_y f_i(\mathbf{x}_i^{(t,k)}, \mathbf{y}_i^{(t,k)}) \right\|^2$$

$$\leq 2 \sum_{i=1}^{n} \frac{w_i^2}{\|\boldsymbol{a}_i\|_1^2} \sum_{k=0}^{\tau_i-1} [a_i^{(k)}(\tau_i)]^2 L_f^2 \Delta_{\mathbf{x},\mathbf{y}}^{(t,k)}(i) + 2 \left( \max_i \frac{p_i \|\boldsymbol{a}_i\|_2^2}{\|\boldsymbol{a}_i\|_1^2} \right) \left[ \sigma_G^2 + 2\beta_G^2 \kappa L_f \mathbb{E} \left( \widetilde{\Phi}(\mathbf{x}^{(t)}) - \widetilde{F}(\mathbf{x}^{(t)}, \mathbf{y}^{(t)}) \right) \right].$$

The same bound in Lemma B.5 holds, but with more stringent conditions on learning rates, namely $\eta_y^c L_f \beta_G \leq \frac{1}{16\kappa\sqrt{M_{\mathbf{a}_{-1}}}}$ and $\tau_{\text{eff}} \gamma_y^s \kappa L_f \beta_G^2 \frac{1}{P} \max \left\{ \kappa \beta_L^2 \max_i \frac{\|\boldsymbol{a}_i\|_2^2}{\|\boldsymbol{a}_i\|_1^2}, 1 \right\} \leq \frac{1}{64}$. Consequently, the bounds in Theorem 1 hold, under slightly more stringent conditions on the learning rates.

## C   Convergence of Fed-Norm-SGDA+ for Nonconvex Concave Functions (Theorem 2)

We organize this section as follows. First, in Appendix C.1 we present some intermediate results, which we use in the proof of Theorem 2. Next, in Appendix C.2, we present the proof of Theorem 2, which is followed by the proofs of the intermediate results in Appendix C.3. Finally, we discuss the extension of our results to nonconvex-one-point-concave functions in Appendix C.4.

The problem we solve is

$$\min_{\mathbf{x}} \max_{\mathbf{y}} \left\{ \widetilde{F}(\mathbf{x}, \mathbf{y}) \triangleq \sum_{i=1}^{n} w_i f_i(\mathbf{x}, \mathbf{y}) \right\}.$$

We define $\widetilde{\Phi}(\mathbf{x}) \triangleq \max_{\mathbf{y}} \widetilde{F}(\mathbf{x}, \mathbf{y})$ and $\widetilde{\mathbf{y}}^*(\mathbf{x}) \in \arg\max_{\mathbf{y}} \widetilde{F}(\mathbf{x}, \mathbf{y})$. Since $\widetilde{F}(\mathbf{x}, \cdot)$ is no longer strongly concave, $\mathbf{y}^*(\mathbf{x})$ need not be unique. In Algorithm 1-Fed-Norm-SGDA+ , the client updates are given by

$$
\begin{aligned}
\mathbf{x}_i^{(t,k)} &= \mathbf{x}^{(t)} - \eta_x^c \sum_{j=0}^{k-1} a_i^{(j)}(k) \nabla_x f_i(\mathbf{x}_i^{(t,j)}, \mathbf{y}_i^{(t,j)}; \xi_i^{(t,j)}), \\
\mathbf{y}_i^{(t,k)} &= \mathbf{y}^{(t)} + \eta_y^c \sum_{j=0}^{k-1} a_i^{(j)}(k) \nabla_y f_i(\widehat{\mathbf{x}}^{(s)}, \mathbf{y}_i^{(t,j)}; \xi_i^{(t,j)}),
\end{aligned}
\tag{42}
$$

where $1 \leq k \leq \tau_i$. The server updates are given by

$$\mathbf{x}^{(t+1)} = \mathbf{x}^{(t)} - \tau_{\text{eff}} \gamma_x^s \mathbf{g}_\mathbf{x}^{(t)}, \qquad \mathbf{y}^{(t+1)} = \mathbf{y}^{(t)} + \tau_{\text{eff}} \gamma_y^s \mathbf{g}_\mathbf{y}^{(t)}, \tag{43}$$

where $\mathbf{g}_\mathbf{x}^{(t)}, \mathbf{g}_\mathbf{y}^{(t)}$ are defined in (3). The normalized (stochastic) gradient vectors are defined as

$$
\begin{aligned}
\mathbf{g}_{\mathbf{x},i}^{(t)} &= \frac{1}{\|\boldsymbol{a}_i\|_1} \sum_{k=0}^{\tau_i-1} a_i^{(k)}(\tau_i) \nabla_x f_i\left(\mathbf{x}_i^{(t,k)}, \mathbf{y}_i^{(t,k)}; \xi_i^{(t,k)}\right); \quad \mathbf{h}_{\mathbf{x},i}^{(t)} = \frac{1}{\|\boldsymbol{a}_i\|_1} \sum_{k=0}^{\tau_i-1} a_i^{(k)}(\tau_i) \nabla_x f_i\left(\mathbf{x}_i^{(t,k)}, \mathbf{y}_i^{(t,k)}\right), \\
\mathbf{g}_{\mathbf{y},i}^{(t)} &= \frac{1}{\|\boldsymbol{a}_i\|_1} \sum_{k=0}^{\tau_i-1} a_i^{(k)}(\tau_i) \nabla_y f_i\left(\widehat{\mathbf{x}}^{(s)}, \mathbf{y}_i^{(t,k)}; \xi_i^{(t,k)}\right); \quad \mathbf{h}_{\mathbf{y},i}^{(t)} = \frac{1}{\|\boldsymbol{a}_i\|_1} \sum_{k=0}^{\tau_i-1} a_i^{(k)}(\tau_i) \nabla_y f_i\left(\widehat{\mathbf{x}}^{(s)}, \mathbf{y}_i^{(t,k)}\right).
\end{aligned}
\tag{44}
$$

## C.1 Intermediate Lemmas

As discussed in Section 5.2, we analyze the convergence of the smoothed envelope function $\widetilde{\Phi}_{1/2L_f}$. We begin with a bound on the one-step decay of this function.

**Lemma C.1** (One-step decay of Smoothed-Envelope). *Suppose the local loss functions $\{f_i\}$ satisfy Assumptions 1, 3, 7, and 8. Then, the iterates generated by Algorithm 1-Fed-Norm-SGDA+ satisfy*

$$
\begin{aligned}
\mathbb{E}\left[\widetilde{\Phi}_{1/2L_f}(\mathbf{x}^{(t+1)})\right] &\leq \mathbb{E}\left[\widetilde{\Phi}_{1/2L_f}(\mathbf{x}^{(t)})\right] + \tau_{\text{eff}}^2 [\gamma_x^s]^2 L_f \frac{n}{P} \left[\sum_{i=1}^{n} \frac{w_i^2 \|\boldsymbol{a}_i\|_2^2}{\|\boldsymbol{a}_i\|_1^2} \left(\sigma_L^2 + \beta_L^2 G_\mathbf{x}^2\right) + G_\mathbf{x}^2 \left(\frac{P-1}{n-1} + \frac{n-P}{n-1} \sum_{i=1}^{n} w_i^2\right)\right] \\
&+ 2\tau_{\text{eff}} \gamma_x^s \left\{ L_f^2 \sum_{i=1}^{n} \frac{w_i}{\|\boldsymbol{a}_i\|_1} \sum_{k=0}^{\tau_i-1} a_i^{(k)}(\tau_i) \Delta_{\mathbf{x},\mathbf{y}}^{(t,k)}(i) + L_f \mathbb{E}\left[\widetilde{\Phi}(\mathbf{x}^{(t)}) - \widetilde{F}(\mathbf{x}^{(t)}, \mathbf{y}^{(t)})\right]\right\} - \frac{\tau_{\text{eff}} \gamma_x^s}{8} \mathbb{E}\left\|\nabla\widetilde{\Phi}_{1/2L_f}(\mathbf{x}^{(t)})\right\|^2,
\end{aligned}
$$

*where $\Delta_{\mathbf{x},\mathbf{y}}^{(t,k)}(i) = \mathbb{E}\left[\|\mathbf{x}_i^{(t,k)} - \mathbf{x}^{(t)}\|^2 + \|\mathbf{y}_i^{(t,k)} - \mathbf{y}^{(t)}\|^2\right]$ is the drift of client $i \in [n]$, at the $k$-th local step of epoch $t$.*

Between two successive synchronization time instants (for example, $t, t+1$), the clients drift apart due to local descent/ascent steps, resulting in the $\{\Delta_{\mathbf{x},\mathbf{y}}^{(t,k)}(i)\}_{i,k}$ terms. Also, $\mathbb{E}\left[\widetilde{\Phi}(\mathbf{x}^{(t)}) - \widetilde{F}(\mathbf{x}^{(t)}, \mathbf{y}^{(t)})\right]$ quantifies the error of the inner maximization. In the subsequent lemmas, we bound both these error terms.

**Lemma C.2** (Consensus Error). *Suppose the local loss functions $\{f_i\}$ satisfy Assumptions 1, 4, 7, and 8. The stochastic oracles for the local functions satisfy Assumption 3. Further, in Algorithm 1-Fed-Norm-SGDA+, we choose the client learning rate $\eta_y^c$ such that $\eta_y^c \leq \frac{1}{2L_f(\max_i \|\boldsymbol{a}_i\|_1)\sqrt{2\max\{1,\beta_L^2\}}}$. Then, the iterates $\{\mathbf{x}_i^{(t)}, \mathbf{y}_i^{(t)}\}$ generated by Algorithm 1-Fed-Norm-SGDA+ satisfy*

$$L_f^2 \sum_{i=1}^{n} \frac{w_i}{\|\boldsymbol{a}_i\|_1} \sum_{k=0}^{\tau_i-1} a_i^{(k)}(\tau_i) \Delta_{\mathbf{x},\mathbf{y}}^{(t,k)}(i) \leq 2\left([\eta_x^c]^2 + [\eta_y^c]^2\right) L_f^2 \sigma_L^2 \sum_{i=1}^{n} w_i \|\mathbf{a}_{i,-1}\|_2^2 + 4L_f^2 M_{\mathbf{a}_{-1}}\left([\eta_x^c]^2 G_\mathbf{x}^2 + [\eta_y^c]^2 \sigma_G^2\right)$$

$$+ 8[\eta_y^c]^2 L_f^3 M_{\mathbf{a}_{-1}} \beta_G^2 \mathbb{E}\left[\widetilde{\Phi}(\widehat{\mathbf{x}}^{(s)}) - \widetilde{F}(\widehat{\mathbf{x}}^{(s)}, \mathbf{y}^{(t)})\right],$$

*where* $M_{\mathbf{a}_{-1}} \triangleq \max_i(\|\mathbf{a}_{i,-1}\|_1^2 + \beta_L^2 \|\mathbf{a}_{i,-1}\|_2^2)$.

Note that consensus error depends on the difference $\mathbb{E}[\widetilde{\Phi}(\widehat{\mathbf{x}}^{(s)}) - \widetilde{F}(\widehat{\mathbf{x}}^{(s)}, \mathbf{y}^{(t)})]$. This is different from the term $\mathbb{E}[\widetilde{\Phi}(\mathbf{x}^{(t)}) - \widetilde{F}(\mathbf{x}^{(t)}, \mathbf{y}^{(t)})]$ in Lemma C.1. Since in Algorithm 1-Fed-Norm-SGDA+ , the **x**-component stays fixed at $\widehat{\mathbf{x}}^{(s)}$ for $S$ communication rounds while updating $\mathbf{y}_i^{(t,k)}$, the difference

$$\sum_{t=kS}^{(k+1)S-1} \mathbb{E}\left[\widetilde{\Phi}(\widehat{\mathbf{x}}^{(s)}) - \widetilde{F}(\widehat{\mathbf{x}}^{(s)}, \mathbf{y}^{(t)})\right]$$

can be interpreted as the optimization error, when maximizing the concave function $F(\widehat{\mathbf{x}}^{(s)}, \cdot)$ over $S$ communication rounds. Next, we bound this error. The following result essentially extends the analysis of FedNova (Wang et al. (2020)) to concave maximization (analogously, convex minimization) problems. We also generalize the corresponding analyses in Khaled et al. (2020); Koloskova et al. (2020) to heterogeneous local updates.

**Lemma C.3** (Local SG updates for Concave Maximization). *Suppose the local functions satisfy Assumptions 1, 3, 4 and 7. Further, let $\left\|\mathbf{y}^{(t)}\right\|^2 \leq R$ for all $t$. We run Algorithm 1-Fed-Norm-SGDA+ with client step-size $\eta_y^c$ such that $64[\eta_y^c]^2 M_{\mathbf{a}_{-1}} L_f^2 \beta_G^2 \frac{n}{P} \leq 1$. Further, the server step-size $\gamma_y^s$ satisfies*

$$2\tau_{eff}\gamma_y^s L_f \frac{n}{P} \max\{\beta_G^2, 1\} \max\left\{\beta_L^2 \max_i \frac{w_i \|\boldsymbol{a}_i\|_2^2}{\|\boldsymbol{a}_i\|_1^2}, \frac{n-P}{n-1} \max_i w_i\right\} \leq \frac{1}{8},$$

$$2\tau_{eff}\gamma_y^s L_f \frac{n}{P} \max\left\{\frac{P-1}{n-1}, \beta_L^2 \max_{i,k} \frac{w_i a_i^{(k)}(\tau_i)}{\|\boldsymbol{a}_i\|_1}\right\} \leq \frac{1}{8}.$$

*Then the iterates generated by Algorithm 1-Fed-Norm-SGDA+ satisfy*

$$\frac{1}{S} \sum_{t=sS}^{(s+1)S-1} \mathbb{E}\left[\widetilde{\Phi}(\widehat{\mathbf{x}}^{(s)}) - \widetilde{F}(\widehat{\mathbf{x}}^{(s)}, \mathbf{y}^{(t)})\right]$$

$$\leq \frac{4R}{\tau_{eff}\gamma_y^s S} + \tau_{eff}\gamma_y^s \frac{n}{P} \left[\sigma_L^2 \sum_{i=1}^n \frac{w_i^2 \|\boldsymbol{a}_i\|_2^2}{\|\boldsymbol{a}_i\|_1^2} + 2\sigma_G^2 \left(\frac{n-P}{n-1} \max_i w_i + \beta_L^2 \max_i \frac{w_i \|\boldsymbol{a}_i\|_2^2}{\|\boldsymbol{a}_i\|_1^2}\right)\right]$$

$$+ 4L_f([\eta_x^c]^2 + [\eta_y^c]^2) \left[\sigma_L^2 \sum_{i=1}^n w_i \|\mathbf{a}_{i,-1}\|_2^2 + 2M_{\mathbf{a}_{-1}}(G_\mathbf{x}^2 + \sigma_G^2)\right],$$

*where* $M_{\mathbf{a}_{-1}} \triangleq \max_i \left(\|\mathbf{a}_{i,-1}\|_1^2 + \beta_L^2 \|\mathbf{a}_{i,-1}\|_2^2\right)$.

*Remark* 10. It is worth noting that the proof of Lemma C.3 does not require global concavity of local functions. Rather, given **x**, we only need concavity of local functions $\{f_i\}$ at some point $\mathbf{y}^*(\mathbf{x})$. This is precisely the one-point-concavity assumption (Assumption 9) discussed earlier in Deng & Mahdavi (2021); Sharma et al. (2022). Therefore, Lemma C.3 for a much larger class of functions. Further, the bound in Lemma C.3 improves the corresponding bounds derived in existing work. As we discuss in Appendix C.4, this helps us achieve improve complexity results for nonconvex-one-point-concave (NC-1PC) functions.

Next, we bound the difference $\mathbb{E}\left[\widetilde{\Phi}(\mathbf{x}^{(t)}) - \widetilde{F}(\mathbf{x}^{(t)}, \mathbf{y}^{(t)})\right]$.

**Lemma C.4.** *Suppose the local functions satisfy Assumptions 1, 3, 4, 8. Then the iterates generated by Algorithm 1-Fed-Norm-SGDA+ satisfy*

$$\frac{1}{T} \sum_{t=0}^{T-1} \mathbb{E}\left[\widetilde{\Phi}(\mathbf{x}^{(t)}) - \widetilde{F}(\mathbf{x}^{(t)}, \mathbf{y}^{(t)})\right] \leq \frac{1}{T} \sum_{s=0}^{T/S-1} \sum_{t=sS}^{(s+1)S-1} \mathbb{E}\left[\widetilde{\Phi}(\widehat{\mathbf{x}}^{(s)}) - \widetilde{F}(\widehat{\mathbf{x}}^{(s)}, \mathbf{y}^{(t)})\right]$$

$$+ 2\tau_{eff}\gamma_x^s G_\mathbf{x}(S-1)\sqrt{\frac{n}{P}} \sqrt{\sum_{i=1}^n \frac{w_i^2 \|\boldsymbol{a}_i\|_2^2}{\|\boldsymbol{a}_i\|_1^2} (\sigma_L^2 + \beta_L^2 G_\mathbf{x}^2) + G_\mathbf{x}^2 \left(\frac{P-1}{n-1} + \frac{n-P}{n-1} \sum_{i=1}^n w_i^2\right)}.$$

### C.2 Proof of Theorem 2

For the sake of completeness, we first state the full statement of Theorem 2 here.

**Theorem 3.** *Suppose the local loss functions $\{f_i\}$ satisfy Assumptions 1, 3, 4, 7, 8. Further, let $\left\|\mathbf{y}^{(t)}\right\|^2 \leq R$ for all $t$. We run Algorithm 1-Fed-Norm-SGDA+ with client step-size $\eta_y^c$ such that $64[\eta_y^c]^2 M_{\mathbf{a}_{-1}} L_f^2 \max\{\beta_G^2 \frac{n}{P}, 1\} \leq 1$. Further, the server step-size $\gamma_y^s$ satisfies*

$$2\tau_{\text{eff}}\gamma_y^s L_f \frac{n}{P} \max\{\beta_G^2, 1\} \max\left\{\beta_L^2 \max_i \frac{w_i\|\boldsymbol{a}_i\|_2^2}{\|\boldsymbol{a}_i\|_1^2}, \frac{n-P}{n-1}\max_i w_i\right\} \leq \frac{1}{8},$$

$$2\tau_{\text{eff}}\gamma_y^s L_f \frac{n}{P} \max\left\{\frac{P-1}{n-1}, \beta_L^2 \max_{i,k} \frac{w_i a_i^{(k)}(\tau_i)}{\|\boldsymbol{a}_i\|_1}\right\} \leq \frac{1}{8}.$$

*Then the iterates generated by Algorithm 1-Fed-Norm-SGDA+ satisfy*

$$\frac{1}{T}\sum_{t=0}^{T-1} \mathbb{E}\left\|\nabla\widetilde{\Phi}_{1/2L_f}(\mathbf{x}^{(t)})\right\|^2 \leq \mathcal{O}\left(\frac{\bar{\Delta}_{\widetilde{\Phi}}}{\tau_{\text{eff}}\gamma_x^s T} + \tau_{\text{eff}}\gamma_x^s L_f\left[\frac{A_w}{P\tau_{\text{eff}}}\left(\sigma_L^2 + \beta_L^2 G_\mathbf{x}^2\right) + G_\mathbf{x}^2\left(\frac{n(P-1)}{P(n-1)} + F_w\right)\right]\right)$$

$$+ \mathcal{O}\left(\tau_{\text{eff}}\gamma_x^s L_f G_\mathbf{x}(S-1)\sqrt{\frac{A_w}{P\tau_{\text{eff}}}\left(\sigma_L^2 + \beta_L^2 G_\mathbf{x}^2\right) + G_\mathbf{x}^2\left(\frac{n(P-1)}{P(n-1)} + F_w\right)}\right)$$

$$+ \mathcal{O}\left(\frac{L_f R}{\tau_{\text{eff}}\gamma_y^s S} + \frac{\gamma_y^s L_f}{P}\left(A_w\sigma_L^2 + \sigma_G^2\left(\tau_{\text{eff}}\frac{n-P}{n-1}E_w + B_w\beta_L^2\right)\right)\right)$$

$$+ \mathcal{O}\left(\left([\eta_x^c]^2 + [\eta_y^c]^2\right)L_f^2\left[C_w\sigma_L^2 + D\left(G_\mathbf{x}^2 + \sigma_G^2\right)\right]\right),$$

*where $\widetilde{\Phi}_{1/2L_f}(\mathbf{x}) \triangleq \min_{\mathbf{x}'}\widetilde{\Phi}(\mathbf{x}') + L_f\|\mathbf{x}' - \mathbf{x}\|^2$ is the envelope function, $\bar{\Delta}_{\widetilde{\Phi}} \triangleq \widetilde{\Phi}_{1/2L_f}(\mathbf{x}_0) - \min_{\mathbf{x}}\widetilde{\Phi}_{1/2L_f}(\mathbf{x})$, $A_w \triangleq n\tau_{\text{eff}}\sum_{i=1}^n \frac{w_i^2\|\boldsymbol{a}_i\|_2^2}{\|\boldsymbol{a}_i\|_1^2}$, $B_w \triangleq n\tau_{\text{eff}}\max_i \frac{w_i\|\boldsymbol{a}_i\|_2^2}{\|\boldsymbol{a}_i\|_1^2}$, $E_w \triangleq n\max_i w_i$, $C_w \triangleq \sum_{i=1}^n w_i(\|\boldsymbol{a}_i\|_2^2 - [\alpha_i^{(t,\tau_i-1)}]^2)$, and $D \triangleq \max_i(\|\mathbf{a}_{i,-1}\|_1^2 + \beta_L^2\|\mathbf{a}_{i,-1}\|_2^2)$, $F_w \triangleq \frac{n(n-P)}{P(n-1)}\sum_{i=1}^n w_i^2$. With the following parameter values:*

$$\eta_x^c = \eta_y^c = \Theta\left(\frac{1}{L_f\bar{\tau}T^{3/8}}\right), \gamma_x^s = \Theta\left(\frac{P^{1/4}}{(\tau_{\text{eff}}T)^{3/4}}\right), \quad \gamma_y^s = \Theta\left(\frac{P^{3/4}}{(\tau_{\text{eff}}T)^{1/4}}\right), \quad S = \Theta\left(\sqrt{\frac{T}{\tau_{\text{eff}}P}}\right),$$

*where $\bar{\tau} = \frac{1}{n}\sum_{i=1}^n \tau_i$, we can further simplify to*

$$\min_{t\in[T]} \mathbb{E}\|\nabla\widetilde{\Phi}_{1/2L_f}(\mathbf{x}^{(t)})\|^2 \leq \underbrace{\mathcal{O}\left(\left(\sigma_G^2\frac{n-P}{n-1}\bar{\Delta}_{\widetilde{\Phi}}\frac{E_w}{PT}\sqrt{1+F_w}\right)^{1/4}\right)}_{\text{Partial participation error}} + \underbrace{\mathcal{O}\left(\frac{C_w\sigma_L^2 + D(G_\mathbf{x}^2 + \sigma_G^2)}{\bar{\tau}^2 T^{3/4}}\right)}_{\text{Local updates error}}$$

$$+ \underbrace{\mathcal{O}\left(\left(\frac{\bar{\Delta}_{\widetilde{\Phi}}\sigma_L^2 A_w}{\tau_{\text{eff}}PT}\sqrt{1+F_w}\right)^{1/4} + \frac{\bar{\Delta}_{\widetilde{\Phi}}(1+F_w)}{T^{3/4}}\left(\frac{\tau_{\text{eff}}P}{A_w + \tau_{\text{eff}}\frac{n-P}{n-1}E_w}\right)^{1/4}\right)}_{\text{Error with full synchronization}},$$

*Proof.* We sum the bound in Lemma C.1 over $t = 0$ to $T - 1$ and rearrange the terms to get

$$\frac{1}{T}\sum_{t=0}^{T-1} \mathbb{E}\left\|\nabla\widetilde{\Phi}_{1/2L_f}(\mathbf{x}^{(t)})\right\|^2$$

$$\leq \frac{8}{\tau_{\text{eff}}\gamma_x^s T}\sum_{t=0}^{T-1} \mathbb{E}\left[\widetilde{\Phi}_{1/2L_f}(\mathbf{x}^{(t)}) - \widetilde{\Phi}_{1/2L_f}(\mathbf{x}^{(t+1)})\right]$$

$$+ 8\tau_{\text{eff}}\gamma_x^s L_f \frac{n}{P}\left[\sum_{i=1}^n \frac{w_i^2\|\boldsymbol{a}_i\|_2^2}{\|\boldsymbol{a}_i\|_1^2}\left(\sigma_L^2 + \beta_L^2 G_\mathbf{x}^2\right) + G_\mathbf{x}^2\left(\frac{P-1}{n-1} + \frac{n-P}{n-1}\sum_{i=1}^n w_i^2\right)\right]$$

$$+ 16L_f \frac{1}{T} \sum_{t=0}^{T-1} \mathbb{E}\left[\widetilde{\Phi}(\mathbf{x}^{(t)}) - \widetilde{F}(\mathbf{x}^{(t)}, \mathbf{y}^{(t)})\right] + \frac{16}{T} \sum_{s=0}^{T/S-1} \sum_{t=sS}^{(s+1)S-1} L_f^2 \sum_{i=1}^{n} \frac{w_i}{\|\boldsymbol{a}_i\|_1} \sum_{k=0}^{\tau_i-1} a_i^{(k)}(\tau_i) \Delta_{\mathbf{x},\mathbf{y}}^{(t,k)}(i)$$

$$\leq \frac{8\left[\widetilde{\Phi}_{1/2L_f}(\mathbf{x}^{(0)}) - \widetilde{\Phi}_{1/2L_f}(\mathbf{x}^{(T)})\right]}{\tau_{\text{eff}} \gamma_x^s T} + 8\tau_{\text{eff}} \gamma_x^s L_f \frac{n}{P} \left[\sum_{i=1}^{n} \frac{w_i^2 \|\boldsymbol{a}_i\|_2^2}{\|\boldsymbol{a}_i\|_1^2} \left(\sigma_L^2 + \beta_L^2 G_{\mathbf{x}}^2\right) + G_{\mathbf{x}}^2 \left(\frac{P-1}{n-1} + \frac{n-P}{n-1} \sum_{i=1}^{n} w_i^2\right)\right]$$

$$+ 32\left([\eta_x^c]^2 + [\eta_y^c]^2\right) L_f^2 \sigma_L^2 \sum_{i=1}^{n} w_i \|\mathbf{a}_{i,-1}\|_2^2 + 64L_f^2 M_{\mathbf{a}_{-1}} \left([\eta_x^c]^2 G_{\mathbf{x}}^2 + [\eta_y^c]^2 \sigma_G^2\right) \qquad \text{(From Lemma C.2)}$$

$$+ 128[\eta_y^c]^2 L_f^3 M_{\mathbf{a}_{-1}} \beta_G^2 \frac{1}{T} \sum_{s=0}^{T/S-1} \sum_{t=sS}^{(s+1)S-1} \mathbb{E}\left[\widetilde{\Phi}(\widehat{\mathbf{x}}^{(s)}) - \widetilde{F}(\widehat{\mathbf{x}}^{(s)}, \mathbf{y}^{(t)})\right] + 16L_f \frac{1}{T} \sum_{t=0}^{T-1} \mathbb{E}\left[\widetilde{\Phi}(\mathbf{x}^{(t)}) - \widetilde{F}(\mathbf{x}^{(t)}, \mathbf{y}^{(t)})\right]$$

$$\leq \frac{8\bar{\Delta}_{\widetilde{\Phi}}}{\tau_{\text{eff}} \gamma_x^s T} + 8\tau_{\text{eff}} \gamma_x^s L_f \frac{n}{P} \left[\sum_{i=1}^{n} \frac{w_i^2 \|\boldsymbol{a}_i\|_2^2}{\|\boldsymbol{a}_i\|_1^2} \left(\sigma_L^2 + \beta_L^2 G_{\mathbf{x}}^2\right) + G_{\mathbf{x}}^2 \left(\frac{P-1}{n-1} + \frac{n-P}{n-1} \sum_{i=1}^{n} w_i^2\right)\right]$$

$$\text{(where } \bar{\Delta}_{\widetilde{\Phi}} \triangleq \widetilde{\Phi}_{1/2L_f}(\mathbf{x}_0) - \min_{\mathbf{x}} \widetilde{\Phi}_{1/2L_f}(\mathbf{x})\text{)}$$

$$+ 32\left([\eta_x^c]^2 + [\eta_y^c]^2\right) L_f^2 \left[\sigma_L^2 \sum_{i=1}^{n} w_i \|\mathbf{a}_{i,-1}\|_2^2 + 2M_{\mathbf{a}_{-1}} \left(G_{\mathbf{x}}^2 + \sigma_G^2\right)\right]$$

$$+ 32\tau_{\text{eff}} \gamma_x^s L_f G_{\mathbf{x}}(S-1) \sqrt{\frac{n}{P}} \sqrt{\sum_{i=1}^{n} \frac{w_i^2 \|\boldsymbol{a}_i\|_2^2}{\|\boldsymbol{a}_i\|_1^2} \left(\sigma_L^2 + \beta_L^2 G_{\mathbf{x}}^2\right) + G_{\mathbf{x}}^2 \left(\frac{P-1}{n-1} + \frac{n-P}{n-1} \sum_{i=1}^{n} w_i^2\right)}$$

$$\text{(From Lemma C.4)}$$

$$+ 18L_f \left[\frac{4R}{\tau_{\text{eff}} \gamma_y^s S} + \tau_{\text{eff}} \gamma_y^s \frac{n}{P} \left(\sigma_L^2 \sum_{i=1}^{n} \frac{w_i^2 \|\boldsymbol{a}_i\|_2^2}{\|\boldsymbol{a}_i\|_1^2} + 2\sigma_G^2 \left(\frac{n-P}{n-1} \max_i w_i + \beta_L^2 \max_i \frac{w_i \|\boldsymbol{a}_i\|_2^2}{\|\boldsymbol{a}_i\|_1^2}\right)\right)\right]$$

$$\text{(From Lemma C.3; using } A_m \leq \min\{\tfrac{1}{2}, \tfrac{1}{16\beta_G^2}\}\text{)}$$

$$+ 72[\eta_y^c]^2 L_f^2 \left[\sigma_L^2 \sum_{i=1}^{n} w_i \|\mathbf{a}_{i,-1}\|_2^2 + 2\sigma_G^2 M_{\mathbf{a}_{-1}}\right]. \tag{45}$$

We can simplify the notation using the constants $A_w \triangleq n\tau_{\text{eff}} \sum_{i=1}^{n} \frac{w_i^2 \|\boldsymbol{a}_i\|_2^2}{\|\boldsymbol{a}_i\|_1^2}$, $B_w \triangleq \tau_{\text{eff}} n \left(\max_i \frac{w_i \|\boldsymbol{a}_i\|_2^2}{\|\boldsymbol{a}_i\|_1^2}\right)$, $E_w \triangleq n \max_i w_i$, $C_w \triangleq \sum_{i=1}^{n} w_i \|\mathbf{a}_{i,-1}\|_2^2$, $D \triangleq M_{\mathbf{a}_{-1}}$, $F_w \triangleq \frac{n}{P} \sum_{i=1}^{n} w_i^2$ and drop the numerical constants, for simplicity, to get

$$\frac{1}{T} \sum_{t=0}^{T-1} \mathbb{E}\left\|\nabla \widetilde{\Phi}_{1/2L_f}(\mathbf{x}^{(t)})\right\|^2$$

$$\lesssim \frac{\bar{\Delta}_{\widetilde{\Phi}}}{\tau_{\text{eff}} \gamma_x^s T} + \tau_{\text{eff}} \gamma_x^s L_f \left[\frac{A_w}{P\tau_{\text{eff}}} \left(\sigma_L^2 + \beta_L^2 G_{\mathbf{x}}^2\right) + G_{\mathbf{x}}^2 \left(\frac{n(P-1)}{P(n-1)} + \frac{(n-P)}{(n-1)} F_w\right)\right]$$

$$+ \tau_{\text{eff}} \gamma_x^s L_f G_{\mathbf{x}}(S-1) \sqrt{\frac{A_w}{P\tau_{\text{eff}}} \left(\sigma_L^2 + \beta_L^2 G_{\mathbf{x}}^2\right) + G_{\mathbf{x}}^2 \left(\frac{n(P-1)}{P(n-1)} + \frac{(n-P)}{(n-1)} F_w\right)}$$

$$+ \frac{L_f R}{\tau_{\text{eff}} \gamma_y^s S} + \frac{\gamma_y^s L_f}{P} \left(A_w \sigma_L^2 + \sigma_G^2 \left(\tau_{\text{eff}} \frac{n-P}{n-1} E_w + B_w \beta_L^2\right)\right) + \left([\eta_x^c]^2 + [\eta_y^c]^2\right) L_f^2 \left[C_w \sigma_L^2 + D\left(G_{\mathbf{x}}^2 + \sigma_G^2\right)\right]$$

$$= \frac{\bar{\Delta}_{\widetilde{\Phi}}}{\tau_{\text{eff}} \gamma_x^s T} + \tau_{\text{eff}} \gamma_x^s L_f \mathcal{I}_1^2 + \frac{\gamma_y^s L_f \mathcal{I}_2}{P} + L_f \left[\tau_{\text{eff}} \gamma_x^s G_{\mathbf{x}}(S-1)\mathcal{I}_1 + \frac{R}{\tau_{\text{eff}} \gamma_y^s S}\right]$$

$$+ \left([\eta_x^c]^2 + [\eta_y^c]^2\right) L_f^2 \left[C_w \sigma_L^2 + D(G_{\mathbf{x}}^2 + \sigma_G^2)\right], \tag{46}$$

where in (46), to simplify notation, we have defined $\mathcal{I}_1 \triangleq \sqrt{\frac{A_w}{P\tau_{\text{eff}}} \left(\sigma_L^2 + \beta_L^2 G_{\mathbf{x}}^2\right) + G_{\mathbf{x}}^2 \left(\frac{n(P-1)}{P(n-1)} + \frac{(n-P)}{(n-1)} F_w\right)}$, $\mathcal{I}_2 \triangleq A_w \sigma_L^2 + (B_w \beta_L^2 + \tau_{\text{eff}} \frac{n-P}{n-1} E_w) \sigma_G^2$.

Next, we optimize the algorithm parameters $S, \gamma_x^s, \gamma_y^s, \eta_y^c, \eta_y^c$ to achieve a tight bound on (46). If $R = 0$, we let $S = 1$. Else, let $S = \sqrt{\frac{R}{\tau_{\text{eff}}^2 \gamma_x^s \gamma_y^s G_{\mathbf{x}} \mathcal{I}_1}}$. Substituting this in (46), we get

$$\frac{1}{T} \sum_{t=0}^{T-1} \mathbb{E} \left\| \nabla \widetilde{\Phi}_{1/2L_f}(\mathbf{x}^{(t)}) \right\|^2 \lesssim \frac{\bar{\Delta}_{\widetilde{\Phi}}}{\tau_{\text{eff}} \gamma_x^s T} + \tau_{\text{eff}} \gamma_x^s L_f \mathcal{I}_1^2 + \frac{\gamma_y^s L_f \mathcal{I}_2}{P} + L_f \sqrt{\frac{R \gamma_x^s G_{\mathbf{x}} \mathcal{I}_1}{\gamma_y^s}}$$
$$+ \left( [\eta_x^c]^2 + [\eta_y^c]^2 \right) L_f^2 \left[ C_w \sigma_L^2 + D(G_{\mathbf{x}}^2 + \sigma_G^2) \right], \tag{47}$$

Next, we focus on the terms in (47) containing $\gamma_y^s$: $L_f \left[ \frac{\gamma_y^s \mathcal{I}_2}{P} + \sqrt{\frac{R \gamma_x^s G_{\mathbf{x}} \mathcal{I}_1}{\gamma_y^s}} \right]$. To optimize these, we choose $\gamma_y^s = \left( \frac{P}{2\mathcal{I}_2} \right)^{2/3} (R \gamma_x^s G_{\mathbf{x}} \mathcal{I}_1)^{1/3}$. Substituting in (47), we get

$$\frac{1}{T} \sum_{t=0}^{T-1} \mathbb{E} \left\| \nabla \widetilde{\Phi}_{1/2L_f}(\mathbf{x}^{(t)}) \right\|^2 \lesssim \frac{\bar{\Delta}_{\widetilde{\Phi}}}{\tau_{\text{eff}} \gamma_x^s T} + \tau_{\text{eff}} \gamma_x^s L_f \mathcal{I}_1^2 + L_f \left( \frac{\mathcal{I}_2}{P} R \gamma_x^s G_{\mathbf{x}} \mathcal{I}_1 \right)^{1/3}$$
$$+ \left( [\eta_x^c]^2 + [\eta_y^c]^2 \right) L_f^2 \left[ C_w \sigma_L^2 + D(G_{\mathbf{x}}^2 + \sigma_G^2) \right], \tag{48}$$

Finally, we focus on the terms in (48) containing $\gamma_x^s$: $\frac{\bar{\Delta}_{\widetilde{\Phi}}}{\tau_{\text{eff}} \gamma_x^s T} + L_f \left( \frac{\mathcal{I}_2}{P} R \gamma_x^s G_{\mathbf{x}} \mathcal{I}_1 \right)^{1/3}$. We ignore the higher order linear term. With $\gamma_x^s = \left( \frac{3 \bar{\Delta}_{\widetilde{\Phi}}}{\tau_{\text{eff}} L_f T} \right)^{3/4} \left( \frac{\mathcal{I}_2}{P} R G_{\mathbf{x}} \mathcal{I}_1 \right)^{-1/4}$, and absorbing numerical constants inside $\mathcal{O}(\cdot)$ we get,

$$\frac{1}{T} \sum_{t=0}^{T-1} \mathbb{E} \left\| \nabla \widetilde{\Phi}_{1/2L_f}(\mathbf{x}^{(t)}) \right\|^2$$
$$\leq \mathcal{O} \left( \frac{\left( \bar{\Delta}_{\widetilde{\Phi}} \mathcal{I}_1 \mathcal{I}_2 \right)^{1/4}}{(\tau_{\text{eff}} P T)^{1/4}} \right) + \mathcal{O} \left( \frac{(\bar{\Delta}_{\widetilde{\Phi}} \tau_{\text{eff}} P)^{1/4}}{T^{3/4}} \frac{\mathcal{I}_1^2}{(\mathcal{I}_1 \mathcal{I}_2)^{1/4}} \right) + \mathcal{O} \left( \left( [\eta_x^c]^2 + [\eta_y^c]^2 \right) L_f^2 \left[ C_w \sigma_L^2 + D(G_{\mathbf{x}}^2 + \sigma_G^2) \right] \right)$$
$$\leq \mathcal{O} \left( \frac{\left( \bar{\Delta}_{\widetilde{\Phi}} \sqrt{G_{\mathbf{x}}^2(1 + F_w)} A_w \sigma_L^2 \right)^{1/4}}{(\tau_{\text{eff}} P T)^{1/4}} \right) + \mathcal{O} \left( \frac{\left( \bar{\Delta}_{\widetilde{\Phi}} \sqrt{G_{\mathbf{x}}^2(1 + F_w)} \tau_{\text{eff}} \frac{n-P}{n-1} E_w \sigma_G^2 \right)^{1/4}}{(\tau_{\text{eff}} P T)^{1/4}} \right)$$
$$+ \mathcal{O} \left( \frac{(\bar{\Delta}_{\widetilde{\Phi}} \tau_{\text{eff}} P)^{1/4}}{T^{3/4}} \frac{G_{\mathbf{x}}^2 \left( 1 + \frac{n-P}{n-1} F_w \right)}{\left( A_w + \tau_{\text{eff}} \frac{n-P}{n-1} E_w \right)^{1/4}} \right) + \mathcal{O} \left( \left( [\eta_x^c]^2 + [\eta_y^c]^2 \right) L_f^2 \left[ C_w \sigma_L^2 + D(G_{\mathbf{x}}^2 + \sigma_G^2) \right] \right). \tag{49}$$

Lastly, we specify the algorithm parameters in terms of $n, T, \tau_{\text{eff}}, \bar{\tau}$.

$$\gamma_x^s = \Theta \left( \frac{P^{1/4}}{(\tau_{\text{eff}} T)^{3/4}} \right), \quad \gamma_y^s = \Theta \left( \frac{P^{3/4}}{(\tau_{\text{eff}} T)^{1/4}} \right), \quad S = \Theta \left( \sqrt{\frac{T}{\tau_{\text{eff}} P}} \right).$$

Finally, choosing the client learning rates $\eta_x^c = \eta_y^c = \frac{1}{L_f \bar{\tau} T^{3/8}}$, we get the result. $\qquad \square$

**Convergence in terms of $F$**

*Proof of Corollary 2.1.* Following Lin et al. (2020a), we define

$$\widetilde{\Phi}_{1/2L_f}(\mathbf{x}) \triangleq \min_{\mathbf{x}'} \left\{ \widetilde{\Phi}(\mathbf{x}') + L_f \|\mathbf{x}' - \mathbf{x}\|^2 \right\}; \qquad \widetilde{\mathbf{x}} \triangleq \arg\min_{\mathbf{x}'} \left\{ \widetilde{\Phi}(\mathbf{x}') + L_f \|\mathbf{x}' - \mathbf{x}\|^2 \right\},$$
$$\Phi_{1/2L_f}(\mathbf{x}) \triangleq \min_{\mathbf{x}'} \left\{ \Phi(\mathbf{x}') + L_f \|\mathbf{x}' - \mathbf{x}\|^2 \right\}; \qquad \bar{\mathbf{x}} \triangleq \arg\min_{\mathbf{x}'} \left\{ \Phi(\mathbf{x}') + L_f \|\mathbf{x}' - \mathbf{x}\|^2 \right\}. \tag{50}$$

Also, it follows from Lemma 2.2 in Davis & Drusvyatskiy (2019) that $\nabla\widetilde{\Phi}_{1/2L_f}(\mathbf{x}) = 2L_f(\mathbf{x} - \widetilde{\mathbf{x}})$ and $\nabla\Phi_{1/2L_f}(\mathbf{x}) = 2L_f(\mathbf{x} - \bar{\mathbf{x}})$. Therefore,

$$\left\|\nabla\Phi_{1/2L_f}(\mathbf{x})\right\|^2 \leq 2\left\|\nabla\Phi_{1/2L_f}(\mathbf{x}) - \nabla\widetilde{\Phi}_{1/2L_f}(\mathbf{x})\right\|^2 + 2\left\|\nabla\widetilde{\Phi}_{1/2L_f}(\mathbf{x})\right\|^2$$
$$= 8L_f^2\left\|\widetilde{\mathbf{x}} - \bar{\mathbf{x}}\right\|^2 + 2\left\|\nabla\widetilde{\Phi}_{1/2L_f}(\mathbf{x})\right\|^2$$

Consequently, we obtain

$$\min_{t\in[T]}\left\|\nabla\Phi_{1/2L_f}(\mathbf{x}^{(t)})\right\|^2 \leq \frac{1}{T}\sum_{t=0}^{T-1}\left\|\nabla\Phi_{1/2L_f}(\mathbf{x}^{(t)})\right\|^2$$
$$\leq \frac{2}{T}\sum_{t=0}^{T-1}\left[\left\|\nabla\widetilde{\Phi}_{1/2L_f}(\mathbf{x}^{(t)})\right\|^2 + 4L_f^2\left\|\widetilde{\mathbf{x}}^{(t)} - \bar{\mathbf{x}}^{(t)}\right\|^2\right]. \qquad (51)$$

where $\widetilde{\mathbf{x}}^{(t)}, \bar{\mathbf{x}}^{(t)}$ follow the same definition as in (50), with $\mathbf{x}$ replaced with $\mathbf{x}^{(t)}$. $\qquad\square$

*Proof of Corollary 2.2.* If clients are weighted equally ($w_i = p_i = 1/n$ for all $i$), with each carrying out $\tau$ steps of local SGDA+, then (6) reduces to

$$\min_{t\in[T]}\mathbb{E}\left\|\nabla\Phi_{1/2L_f}(\mathbf{x}^{(t)})\right\|^2 \leq \mathcal{O}\left(\frac{1}{(\tau PT)^{1/4}} + \frac{(\tau P)^{1/4}}{T^{3/4}}\right) + \mathcal{O}\left(\frac{\sigma_L^2 + \tau(G_{\mathbf{x}}^2 + \sigma_G^2)}{\tau T^{3/4}}\right) + \mathcal{O}\left(\left(\frac{n-P}{n-1}\cdot\frac{1}{PT}\right)^{1/4}\right).$$

- For full client participation, this reduces to

$$\min_{t\in[T]}\mathbb{E}\left\|\nabla\Phi_{1/2L_f}(\mathbf{x}^{(t)})\right\|^2 \leq \mathcal{O}\left(\frac{1}{(\tau nT)^{1/4}} + \frac{(\tau n)^{1/4}}{T^{3/4}}\right).$$

  To reach an $\epsilon$-stationary point, assuming $n\tau \leq T$, the per-client gradient complexity is $T\tau = \mathcal{O}\left(\frac{1}{n\epsilon^8}\right)$. Since $\tau \leq T/n$, the minimum number of communication rounds required is $T = \mathcal{O}\left(\frac{1}{\epsilon^4}\right)$.

- For partial participation, $\mathcal{O}\left(\left(\frac{n-P}{n-1}\cdot\frac{1}{PT}\right)^{1/4}\right)$ is the dominant term, and we do not get any convergence benefit of multiple local updates. Consequently, per-gradient client complexity and number of communication rounds are both $T\tau = \mathcal{O}\left(\frac{1}{P\epsilon^8}\right)$, for $\tau = \mathcal{O}(1)$. However, if the data across clients comes from identical distributions ($\sigma_G = 0$), then we recover per-client gradient complexity of $\mathcal{O}\left(\frac{1}{P\epsilon^8}\right)$, and number of communication rounds $= \mathcal{O}\left(\frac{1}{\epsilon^4}\right)$.

$\qquad\square$

**Special Cases**

- Centralized, deterministic case ($\sigma_L = \sigma_G = 0, \beta_G = 1, \tau_{\text{eff}} = n = 1$): in this case $A_w = B_w = 1, C_w = D = 0$. Also, $\mathcal{I}_1 = G_{\mathbf{x}}\sqrt{\beta_L^2 + 1}, \mathcal{I}_2 = 0$. The bound in (45) reduces to

$$\frac{1}{T}\sum_{t=0}^{T-1}\mathbb{E}\left\|\nabla\widetilde{\Phi}_{1/2L_f}(\mathbf{x}^{(t)})\right\|^2 \leq \mathcal{O}\left(\frac{\bar{\Delta}_{\widetilde{\Phi}}}{\gamma_x^s T} + \gamma_x^s L_f G_{\mathbf{x}}^2\left[(\beta_L^2 + 1) + (S-1)\sqrt{\beta_L^2 + 1}\right] + \frac{L_f R}{\gamma_y^s S}\right). \quad (52)$$

  For $\beta_L = 0$, (52) yields the convergence result in Lin et al. (2020a).

- Single node, stochastic case ($\sigma_G = 0, \beta_G = 1, \tau_{\text{eff}} = n = 1$): in this case $A_w = B_w = 1, C_w = D = 0$. Also, $\mathcal{I}_1 = \sqrt{\sigma_L^2 + (\beta_L^2 + 1)G_{\mathbf{x}}^2}, \mathcal{I}_2 = \sigma_L^2$. The bound in (45) reduces to

$$\frac{1}{T}\sum_{t=0}^{T-1}\mathbb{E}\left\|\nabla\widetilde{\Phi}_{1/2L_f}(\mathbf{x}^{(t)})\right\|^2 \leq \Theta\left(\frac{\bar{\Delta}_{\widetilde{\Phi}}}{\gamma_x^s T} + \gamma_x^s L_f(\sigma_L^2 + (\beta_L^2 + 1)G_{\mathbf{x}}^2) + \gamma_y^s L_f \sigma_L^2\right)$$

$$+ \Theta \left( L_f \left[ \gamma_x^s G_\mathbf{x}(S-1)\sqrt{\sigma_L^2 + (\beta_L^2+1)G_\mathbf{x}^2} + \frac{R}{\gamma_y^s S} \right] \right). \qquad (53)$$

Again, for $\beta_L = 0$, (53) yields the convergence result in Lin et al. (2020a).

- Multiple equally weighted ($w_i = 1/n, \forall\, i \in [n]$) clients, full client participation, stochastic case with synchronous client updates ($\tau_{\text{eff}} = 1$): in this case $A_w = 1, B_w = 1, C_w = D = 0, 0$. The bound in (45) reduces to

$$\frac{1}{T}\sum_{t=0}^{T-1} \mathbb{E}\left\| \nabla\widetilde{\Phi}_{1/2L_f}(\mathbf{x}^{(t)}) \right\|^2 \le \Theta\left( \frac{\bar{\Delta}_{\widetilde{\Phi}}}{\gamma_x^s T} + \gamma_x^s L_f \left( G_\mathbf{x}^2 + \frac{\sigma_L^2 + \beta_L^2 G_\mathbf{x}^2}{n} \right) \right)$$
$$+ \Theta\left( \frac{\gamma_y^s L_f}{n}(\sigma_L^2 + \sigma_G^2\beta_L^2) + \gamma_x^s L_f G_\mathbf{x}(S-1)\sqrt{G_\mathbf{x}^2 + \frac{\sigma_L^2 + \beta_L^2 G_\mathbf{x}^2}{n}} + \frac{R L_f}{\gamma_y^s S} \right), \qquad (54)$$

Note that unlike existing analyses of synchronous update algorithms Woodworth et al. (2020); Yun et al. (2022); Sharma et al. (2022), the bound in (54) depends on the inter-client heterogeneity $\sigma_G^2$. This is due to the more general noise assumption (Assumption 3). In the existing works, $\beta_L^2$ is assumed zero, in which case, the bound in (54) is also independent of $\sigma_G^2$. See Appendix A.2 for a more detailed explanation.

- Multiple, equally weighted ($w_i = 1/n, \forall\, i \in [n]$) clients, full client participation, multiple, but equal number of client updates ($\tau_i = \tau_{\text{eff}} = \tau, \forall\, i \in [n]$). In this case $A_w = B_w = 1, C_w = \tau - 1, D = (\tau-1)(\tau-1+\beta_L^2)$. The bound in (45) then reduces to

$$\frac{1}{T}\sum_{t=0}^{T-1} \mathbb{E}\left\| \nabla\widetilde{\Phi}_{1/2L_f}(\mathbf{x}^{(t)}) \right\|^2 \le \Theta\left( \frac{\bar{\Delta}_{\widetilde{\Phi}}}{\tau\gamma_x^s T} + \tau\gamma_x^s L_f \left( G_\mathbf{x}^2 + \frac{\sigma_L^2 + \beta_L^2 G_\mathbf{x}^2}{n\tau} \right) + \frac{\gamma_y^s L_f(\sigma_L^2 + \beta_L^2\sigma_G^2)}{n} \right) \qquad (55)$$
$$+ \Theta\left( L_f \left[ \tau\gamma_x^s G_\mathbf{x}(S-1)\sqrt{G_\mathbf{x}^2 + \frac{\sigma_L^2 + \beta_L^2 G_\mathbf{x}^2}{n\tau}} + \frac{R}{\tau\gamma_y^s S} \right] + (\tau-1)\left([\eta_x^c]^2 + [\eta_y^c]^2\right)L_f^2\left[\sigma_L^2 + (\tau-1+\beta_L^2)(G_\mathbf{x}^2 + \sigma_G^2)\right] \right).$$

For $\beta_L = \beta_G = 0$, this setting reduces to the one considered in Sharma et al. (2022). However, as stated earlier, our bound on the local update error is tighter.

### C.3 Proofs of the Intermediate Lemmas

*Proof of Lemma C.1.* Using the definition in (50) $\bar{\mathbf{x}}^{(t)} = \arg\min_\mathbf{x} \widetilde{\Phi}(\mathbf{x}) + L_f \left\| \mathbf{x} - \mathbf{x}^{(t)} \right\|^2$. Also, note that

$$\widetilde{\Phi}_{1/2L_f}(\mathbf{x}^{(t+1)}) \le \widetilde{\Phi}(\bar{\mathbf{x}}^{(t)}) + L_f \left\| \bar{\mathbf{x}}^{(t)} - \mathbf{x}^{(t+1)} \right\|^2. \qquad (56)$$

Using the $\mathbf{x}$ updates in (43),

$$\mathbb{E}\left\| \bar{\mathbf{x}}^{(t)} - \mathbf{x}^{(t+1)} \right\|^2 = \mathbb{E}\left\| \bar{\mathbf{x}}^{(t)} - \mathbf{x}^{(t)} + \tau_{\text{eff}}\gamma_x^s \sum_{i\in\mathcal{C}^{(t)}} \tilde{w}_i \mathbf{g}_{\mathbf{x},i}^{(t)} \right\|^2$$

$$= \mathbb{E}\left\| \bar{\mathbf{x}}^{(t)} - \mathbf{x}^{(t)} \right\|^2 + \tau_{\text{eff}}^2[\gamma_x^s]^2\mathbb{E}\left\| \sum_{i\in\mathcal{C}^{(t)}} \tilde{w}_i \mathbf{g}_{\mathbf{x},i}^{(t)} \right\|^2 + 2\tau_{\text{eff}}\gamma_x^s\mathbb{E}\left\langle \bar{\mathbf{x}}^{(t)} - \mathbf{x}^{(t)}, \sum_{i=1}^{n} w_i \mathbf{h}_{\mathbf{x},i}^{(t)} \right\rangle$$

$$\le \mathbb{E}\left\| \bar{\mathbf{x}}^{(t)} - \mathbf{x}^{(t)} \right\|^2 + 2\tau_{\text{eff}}\gamma_x^s\mathbb{E}\left\langle \bar{\mathbf{x}}^{(t)} - \mathbf{x}^{(t)}, \nabla_x\widetilde{F}(\mathbf{x}^{(t)}, \mathbf{y}^{(t)}) \right\rangle + \tau_{\text{eff}}^2[\gamma_x^s]^2\mathbb{E}\left\| \sum_{i\in\mathcal{C}^{(t)}} \tilde{w}_i \mathbf{g}_{\mathbf{x},i}^{(t)} \right\|^2$$

$$+ \tau_{\text{eff}}\gamma_x^s\mathbb{E}\left[ \frac{L_f}{2}\left\| \bar{\mathbf{x}}^{(t)} - \mathbf{x}^{(t)} \right\|^2 + \frac{2}{L_f}\left\| \sum_{i=1}^{n}\frac{w_i}{\|\boldsymbol{a}_i\|_1}\sum_{k=0}^{\tau_i-1} a_i^{(k)}(\tau_i)\left( \nabla_x f_i(\mathbf{x}_i^{(t,k)}, \mathbf{y}_i^{(t,k)}) - \nabla_x f_i(\mathbf{x}^{(t)}, \mathbf{y}^{(t)}) \right) \right\|^2 \right]$$

$$\leq \mathbb{E}\left\|\bar{\mathbf{x}}^{(t)} - \mathbf{x}^{(t)}\right\|^2 + \tau_{\text{eff}}^2[\gamma_x^s]^2\mathbb{E}\left\|\sum_{i\in\mathcal{C}^{(t)}}\tilde{w}_i\mathbf{g}_{\mathbf{x},i}^{(t)}\right\|^2 \tag{57}$$

$$+ \tau_{\text{eff}}\gamma_x^s\mathbb{E}\left[\frac{L_f}{2}\left\|\bar{\mathbf{x}}^{(t)} - \mathbf{x}^{(t)}\right\|^2 + 2L_f\sum_{i=1}^{n}\frac{w_i}{\|\boldsymbol{a}_i\|_1}\sum_{k=0}^{\tau_i-1}a_i^{(k)}(\tau_i)\Delta_{\mathbf{x},\mathbf{y}}^{(t,k)}(i) + 2\left\langle\bar{\mathbf{x}}^{(t)} - \mathbf{x}^{(t)}, \nabla_x\widetilde{F}(\mathbf{x}^{(t)}, \mathbf{y}^{(t)})\right\rangle\right], \tag{58}$$

where (58) follows from $L_f$-smoothness (Assumption 1) and Jensen's inequality. From (18), (19), we can bound $\mathbb{E}\big\|\sum_{i\in\mathcal{C}^{(t')}}\tilde{w}_i\mathbf{g}_{\mathbf{x},i}^{(t')}\big\|^2$ as follows.

$$\mathbb{E}\left\|\sum_{i\in\mathcal{C}^{(t')}}\tilde{w}_i\mathbf{g}_{\mathbf{x},i}^{(t')}\right\|^2 \leq \frac{n}{P}\sum_{i=1}^{n}\frac{w_i^2}{\|\boldsymbol{a}_i\|_1^2}\sum_{k=0}^{\tau_i-1}[a_i^{(k)}(\tau_i)]^2\left[\sigma_L^2 + \beta_L^2\mathbb{E}\left\|\nabla_x f_i(\mathbf{x}_i^{(t,k)}, \mathbf{y}_i^{(t,k)})\right\|^2\right]$$

$$+ \frac{n}{P}\left(\frac{P-1}{n-1}\right)\mathbb{E}\left\|\sum_{i=1}^{n}w_i\mathbf{h}_{\mathbf{x},i}^{(t)}\right\|^2 + \frac{n}{P}\frac{n-P}{n-1}\sum_{i=1}^{n}w_i^2\mathbb{E}\left\|\mathbf{h}_{\mathbf{x},i}^{(t)}\right\|^2$$

$$\leq \frac{n}{P}\sum_{i=1}^{n}\frac{w_i^2\|\boldsymbol{a}_i\|_2^2}{\|\boldsymbol{a}_i\|_1^2}\left(\sigma_L^2 + \beta_L^2 G_{\mathbf{x}}^2\right) + \frac{n(P-1)}{P(n-1)}G_{\mathbf{x}}^2 + \frac{n(n-P)}{P(n-1)}G_{\mathbf{x}}^2\sum_{i=1}^{n}w_i^2, \tag{59}$$

where the final inequality by using Assumption 8. Next, we bound the inner product term in (58). Using $L_f$-smoothness of $F$ (Assumption 1):

$$\mathbb{E}\left\langle\bar{\mathbf{x}}^{(t)} - \mathbf{x}^{(t)}, \nabla_x\widetilde{F}(\mathbf{x}^{(t)}, \mathbf{y}^{(t)})\right\rangle \leq \mathbb{E}\left[\widetilde{F}(\bar{\mathbf{x}}^{(t)}, \mathbf{y}^{(t)}) - \widetilde{F}(\mathbf{x}^{(t)}, \mathbf{y}^{(t)}) + \frac{L_f}{2}\left\|\bar{\mathbf{x}}^{(t)} - \mathbf{x}^{(t)}\right\|^2\right]$$

$$\leq \mathbb{E}\left[\widetilde{\Phi}(\bar{\mathbf{x}}^{(t)}) + L_f\left\|\bar{\mathbf{x}}^{(t)} - \mathbf{x}^{(t)}\right\|^2\right] - \mathbb{E}\widetilde{F}(\mathbf{x}^{(t)}, \mathbf{y}^{(t)}) - \frac{L_f}{2}\mathbb{E}\left\|\bar{\mathbf{x}}^{(t)} - \mathbf{x}^{(t)}\right\|^2$$

$$\leq \mathbb{E}\left[\widetilde{\Phi}(\mathbf{x}^{(t)}) + L_f\left\|\mathbf{x}^{(t)} - \mathbf{x}^{(t)}\right\|^2\right] - \mathbb{E}\widetilde{F}(\mathbf{x}^{(t)}, \mathbf{y}^{(t)}) - \frac{L_f}{2}\mathbb{E}\left\|\bar{\mathbf{x}}^{(t)} - \mathbf{x}^{(t)}\right\|^2 \quad\text{(by definition of } \bar{\mathbf{x}}^{(t)}\text{)}$$

$$\leq \mathbb{E}\left[\widetilde{\Phi}(\mathbf{x}^{(t)}) - \widetilde{F}(\mathbf{x}^{(t)}, \mathbf{y}^{(t)}) - \frac{L_f}{2}\left\|\bar{\mathbf{x}}^{(t)} - \mathbf{x}^{(t)}\right\|^2\right]. \tag{60}$$

Substituting the bounds from (58) and (60) into (56), we get

$$\mathbb{E}\left[\widetilde{\Phi}_{1/2L_f}(\mathbf{x}^{(t+1)})\right] \leq \mathbb{E}\left[\widetilde{\Phi}(\bar{\mathbf{x}}^{(t)}) + L_f\left\|\bar{\mathbf{x}}^{(t)} - \mathbf{x}^{(t)}\right\|^2\right]$$

$$+ \tau_{\text{eff}}^2[\gamma_x^s]^2 L_f\frac{n}{P}\left[\sum_{i=1}^{n}\frac{w_i^2\|\boldsymbol{a}_i\|_2^2}{\|\boldsymbol{a}_i\|_1^2}\left(\sigma_L^2 + \beta_L^2 G_{\mathbf{x}}^2\right) + G_{\mathbf{x}}^2\left(\frac{P-1}{n-1} + \frac{n-P}{n-1}\sum_{i=1}^{n}w_i^2\right)\right]$$

$$+ 2\tau_{\text{eff}}\gamma_x^s L_f^2\sum_{i=1}^{n}\frac{w_i}{\|\boldsymbol{a}_i\|_1}\sum_{k=0}^{\tau_i-1}a_i^{(k)}(\tau_i)\Delta_{\mathbf{x},\mathbf{y}}^{(t,k)}(i) + 2\tau_{\text{eff}}\gamma_x^s L_f\mathbb{E}\left[\widetilde{\Phi}(\mathbf{x}^{(t)}) - \widetilde{F}(\mathbf{x}^{(t)}, \mathbf{y}^{(t)})\right] - \frac{\tau_{\text{eff}}\gamma_x^s L_f^2}{2}\mathbb{E}\left\|\bar{\mathbf{x}}^{(t)} - \mathbf{x}^{(t)}\right\|^2$$

$$\leq \mathbb{E}\left[\widetilde{\Phi}_{1/2L_f}(\mathbf{x}^{(t)})\right] + \tau_{\text{eff}}^2[\gamma_x^s]^2 L_f\frac{n}{P}\left[\sum_{i=1}^{n}\frac{w_i^2\|\boldsymbol{a}_i\|_2^2}{\|\boldsymbol{a}_i\|_1^2}\left(\sigma_L^2 + \beta_L^2 G_{\mathbf{x}}^2\right) + G_{\mathbf{x}}^2\left(\frac{P-1}{n-1} + \frac{n-P}{n-1}\sum_{i=1}^{n}w_i^2\right)\right]$$

$$+ 2\tau_{\text{eff}}\gamma_x^s\left\{L_f^2\sum_{i=1}^{n}\frac{w_i}{\|\boldsymbol{a}_i\|_1}\sum_{k=0}^{\tau_i-1}a_i^{(k)}(\tau_i)\Delta_{\mathbf{x},\mathbf{y}}^{(t,k)}(i) + L_f\mathbb{E}\left[\widetilde{\Phi}(\mathbf{x}^{(t)}) - \widetilde{F}(\mathbf{x}^{(t)}, \mathbf{y}^{(t)})\right]\right\} - \frac{\tau_{\text{eff}}\gamma_x^s}{8}\mathbb{E}\left\|\nabla\widetilde{\Phi}_{1/2L_f}(\mathbf{x}^{(t)})\right\|^2,$$

where we use $\nabla\widetilde{\Phi}_{1/2L_f}(\mathbf{x}) = 2L_f(\mathbf{x} - \bar{\mathbf{x}})$ from (50). $\qquad\square$

*Proof of Lemma C.2.* We use the client update equations for individual iterates in (42). To bound $\Delta_{\mathbf{x},\mathbf{y}}^{(t,k)}(i)$, first we bound the **x**-error $\mathbb{E}\left\|\mathbf{x}_i^{(t,k)} - \mathbf{x}^{(t)}\right\|^2$. Starting from (24), using Assumption 8, for $1 \le k \le \tau_i$,

$$
\frac{1}{\|\boldsymbol{a}_i\|_1} \sum_{k=0}^{\tau_i-1} a_i^{(k)}(\tau_i) \mathbb{E}\left\|\mathbf{x}_i^{(t,k)} - \mathbf{x}^{(t)}\right\|^2 \le \frac{[\eta_x^c]^2}{\|\boldsymbol{a}_i\|_1} \sum_{k=0}^{\tau_i-1} a_i^{(k)}(\tau_i) \left[ \sum_{j=0}^{k-1} [a_i^{(j)}(k)]^2 \left(\sigma_L^2 + \beta_L^2 G_{\mathbf{x}}^2\right) + \left(\sum_{j=0}^{k-1} a_i^{(j)}(k)\right) \sum_{j=0}^{k-1} a_i^{(j)}(k) G_{\mathbf{x}}^2 \right]
$$

$$
\le [\eta_x^c]^2 \left[\sigma_L^2 \|\mathbf{a}_{i,-1}\|_2^2 + G_{\mathbf{x}}^2 \left(\|\mathbf{a}_{i,-1}\|_1^2 + \beta_L^2 \|\mathbf{a}_{i,-1}\|_2^2\right)\right], \tag{61}
$$

where we use (25). Next, we bound $\mathbb{E}\left\|\mathbf{y}_i^{(t,k)} - \mathbf{y}^{(t)}\right\|^2$, using the bound from (29), to get

$$
\frac{1}{\|\boldsymbol{a}_i\|_1} \sum_{k=0}^{\tau_i-1} a_i^{(k)}(\tau_i) \mathbb{E}\left\|\mathbf{y}_i^{(t,k)} - \mathbf{y}^{(t)}\right\|^2 \le [\eta_y^c]^2 \sigma_L^2 \|\mathbf{a}_{i,-1}\|_2^2 + 2[\eta_y^c]^2 L_f^2 \left(\|\mathbf{a}_{i,-1}\|_1 + \beta_L^2 \alpha\right) \sum_{k=0}^{\tau_i-1} a_i^{(k)}(\tau_i) \Delta_{\mathbf{y}}^{(t,k)}(i)
$$

$$
+ 2[\eta_y^c]^2 \left(\|\mathbf{a}_{i,-1}\|_1^2 + \beta_L^2 \|\mathbf{a}_{i,-1}\|_2^2\right) \mathbb{E}\left\|\nabla_y f_i(\widehat{\mathbf{x}}^{(s)}, \mathbf{y}^{(t)})\right\|^2. \tag{62}
$$

Compared to (29), the difference is the presence of $\Delta_{\mathbf{y}}^{(t,k)}(i)$ in (62), rather than $\Delta_{\mathbf{x},\mathbf{y}}^{(t,k)}(i)$. Taking a weighted sum over agents in (62), we get

$$
L_f^2 \sum_{i=1}^n \frac{w_i}{\|\boldsymbol{a}_i\|_1} \sum_{k=0}^{\tau_i-1} a_i^{(k)}(\tau_i) \Delta_{\mathbf{y}}^{(t,k)}(i) \le 2[\eta_y^c]^2 L_f^2 \left[\sigma_L^2 \sum_{i=1}^n w_i \|\mathbf{a}_{i,-1}\|_2^2 + 2M_{\mathbf{a}_{-1}} \left(\beta_G^2 \mathbb{E}\left\|\nabla_y \widetilde{F}(\widehat{\mathbf{x}}^{(s)}, \mathbf{y}^{(t)})\right\|^2 + \sigma_G^2\right)\right]. \tag{63}
$$

where, we choose $\eta_y^c$ such that $A_m \triangleq 2L_f^2 [\eta_y^c]^2 \max_i \|\boldsymbol{a}_i\|_1 \left(\|\mathbf{a}_{i,-1}\|_1 + \beta_L^2 \alpha\right) \le \frac{1}{2}$, and define $M_{\mathbf{a}_{-1}} \triangleq \max_i \left(\|\mathbf{a}_{i,-1}\|_1^2 + \beta_L^2 \|\mathbf{a}_{i,-1}\|_2^2\right)$. Next, it follows from $L_f$-smoothness (Assumption 1) and Lemma A.7 that

$$
\mathbb{E}\left\|\nabla_y \widetilde{F}\left(\widehat{\mathbf{x}}^{(s)}, \mathbf{y}^{(t)}\right)\right\|^2 \le 2L_f \mathbb{E}\left[\widetilde{\Phi}(\widehat{\mathbf{x}}^{(s)}) - \widetilde{F}(\widehat{\mathbf{x}}^{(s)}, \mathbf{y}^{(t)})\right].
$$

Subsequently, combining (61) and (63), we get

$$
L_f^2 \sum_{i=1}^n \frac{w_i}{\|\boldsymbol{a}_i\|_1} \sum_{k=0}^{\tau_i-1} a_i^{(k)}(\tau_i) \Delta_{\mathbf{x},\mathbf{y}}^{(t,k)}(i) \le 2\left([\eta_x^c]^2 + [\eta_y^c]^2\right) L_f^2 \sigma_L^2 \sum_{i=1}^n w_i \|\mathbf{a}_{i,-1}\|_2^2 + 4L_f^2 M_{\mathbf{a}_{-1}} \left([\eta_x^c]^2 G_{\mathbf{x}}^2 + [\eta_y^c]^2 \sigma_G^2\right)
$$

$$
+ 8[\eta_y^c]^2 L_f^3 M_{\mathbf{a}_{-1}} \beta_G^2 \mathbb{E}\left[\widetilde{\Phi}(\widehat{\mathbf{x}}^{(s)}) - \widetilde{F}(\widehat{\mathbf{x}}^{(s)}, \mathbf{y}^{(t)})\right].
$$

which finishes the proof. $\qquad\square$

*Proof of Lemma C.3.* We define $\mathbf{y}^*(\widehat{\mathbf{x}}^{(s)}) \in \arg\max_{\mathbf{y}} \widetilde{F}(\widehat{\mathbf{x}}^{(s)}, \mathbf{y})$. Then,

$$
\mathbb{E}\left\|\mathbf{y}^{(t+1)} - \mathbf{y}^*(\widehat{\mathbf{x}}^{(s)})\right\|^2 \stackrel{(43)}{=} \mathbb{E}\left\|\mathbf{y}^{(t)} + \tau_{\text{eff}} \gamma_y^s \mathbf{g}_{\mathbf{y}}^{(t)} - \mathbf{y}^*(\widehat{\mathbf{x}}^{(s)})\right\|^2
$$

$$
= \mathbb{E}\left\|\mathbf{y}^{(t)} - \mathbf{y}^*(\widehat{\mathbf{x}}^{(s)})\right\|^2 + \tau_{\text{eff}}^2 [\gamma_y^s]^2 \mathbb{E}\left\|\mathbf{g}_{\mathbf{y}}^{(t)}\right\|^2 + 2\tau_{\text{eff}} \gamma_y^s \mathbb{E}\left\langle \mathbf{y}^{(t)} - \mathbf{y}^*(\widehat{\mathbf{x}}^{(s)}), \sum_{i=1}^n w_i \mathbf{h}_{\mathbf{y},i}^{(t)}\right\rangle. \tag{64}
$$

$\mathbb{E}\left\|\mathbf{g}_{\mathbf{y}}^{(t)}\right\|^2$ is bounded in (38). We only need to further bound $\mathbb{E}\left\|\sum_{i=1}^n w_i \mathbf{h}_{\mathbf{y},i}^{(t)}\right\|^2$.

$$
\mathbb{E}\left\|\sum_{i=1}^n w_i \mathbf{h}_{\mathbf{y},i}^{(t)}\right\|^2 \le \mathbb{E}\left\|\sum_{i=1}^n \frac{w_i}{\|\boldsymbol{a}_i\|_1} \sum_{k=0}^{\tau_i-1} [a_i^{(k)}(\tau_i)] \left(\nabla_y f_i(\widehat{\mathbf{x}}^{(s)}, \mathbf{y}_i^{(t,k)}) - \nabla_y f_i(\widehat{\mathbf{x}}^{(s)}, \mathbf{y}^{(t)}) + \nabla_y f_i(\widehat{\mathbf{x}}^{(s)}, \mathbf{y}^{(t)})\right)\right\|^2
$$

$$
\le 2L_f^2 \sum_{i=1}^n \frac{w_i}{\|\boldsymbol{a}_i\|_1} \sum_{k=0}^{\tau_i-1} [a_i^{(k)}(\tau_i)] \Delta_{\mathbf{y}}^{(t,k)}(i) + 2\left\|\nabla_y \widetilde{F}(\widehat{\mathbf{x}}^{(s)}, \mathbf{y}^{(t)})\right\|^2 \qquad \text{(Jensen's inequality)}
$$

$$\leq 2L_f^2 \sum_{i=1}^{n} \frac{w_i}{\|\boldsymbol{a}_i\|_1} \sum_{k=0}^{\tau_i-1} [a_i^{(k)}(\tau_i)]\Delta_{\mathbf{y}}^{(t,k)}(i) + 4L_f \mathbb{E}\left[\widetilde{\Phi}(\widehat{\mathbf{x}}^{(s)}) - \widetilde{F}(\widehat{\mathbf{x}}^{(s)}, \mathbf{y}^{(t)})\right]. \tag{65}$$

Next, we bound the third term in (64).

$$\mathbb{E}\left\langle \mathbf{y}^{(t)} - \mathbf{y}^*(\widehat{\mathbf{x}}^{(s)}), \sum_{i=1}^{n} w_i \mathbf{h}_{\mathbf{y},i}^{(t)}\right\rangle = \mathbb{E}\left\langle \mathbf{y}^{(t)} - \mathbf{y}^*(\widehat{\mathbf{x}}^{(s)}), \sum_{i=1}^{n} \frac{w_i}{\|\boldsymbol{a}_i\|_1} \sum_{k=0}^{\tau_i-1} [a_i^{(k)}(\tau_i)]\nabla_y f_i(\widehat{\mathbf{x}}^{(s)}, \mathbf{y}_i^{(t,k)})\right\rangle$$

$$= \sum_{i=1}^{n} \frac{w_i}{\|\boldsymbol{a}_i\|_1} \sum_{k=0}^{\tau_i-1} [a_i^{(k)}(\tau_i)]\mathbb{E}\left[\left\langle \mathbf{y}^{(t)} - \mathbf{y}_i^{(t,k)}, \nabla_y f_i(\widehat{\mathbf{x}}^{(s)}, \mathbf{y}_i^{(t,k)})\right\rangle + \left\langle \mathbf{y}_i^{(t,k)} - \mathbf{y}^*(\widehat{\mathbf{x}}^{(s)}), \nabla_y f_i(\widehat{\mathbf{x}}^{(s)}, \mathbf{y}_i^{(t,k)})\right\rangle\right]$$

$$\leq \sum_{i=1}^{n} \frac{w_i}{\|\boldsymbol{a}_i\|_1} \sum_{k=0}^{\tau_i-1} [a_i^{(k)}(\tau_i)]\mathbb{E}\left[f_i(\widehat{\mathbf{x}}^{(s)}, \mathbf{y}^{(t)}) - f_i(\widehat{\mathbf{x}}^{(s)}, \mathbf{y}_i^{(t,k)}) + \frac{L_f}{2}\left\|\mathbf{y}^{(t)} - \mathbf{y}_i^{(t,k)}\right\|^2 \right. \qquad (L_f\text{-smoothness})$$

$$\left. + f_i(\widehat{\mathbf{x}}^{(s)}, \mathbf{y}_i^{(t,k)}) - f_i(\widehat{\mathbf{x}}^{(s)}, \mathbf{y}^*(\widehat{\mathbf{x}}^{(s)}))\right] \qquad (\text{Concavity in } \mathbf{y})$$

$$= \frac{L_f}{2}\sum_{i=1}^{n} \frac{w_i}{\|\boldsymbol{a}_i\|_1} \sum_{k=0}^{\tau_i-1} [a_i^{(k)}(\tau_i)]\Delta_{\mathbf{y}}^{(t,k)}(i) - \mathbb{E}\left[\widetilde{\Phi}(\widehat{\mathbf{x}}^{(s)}) - \widetilde{F}(\widehat{\mathbf{x}}^{(s)}, \mathbf{y}^{(t)})\right]. \tag{66}$$

Substituting (38), (65), (66) in (64), we get

$$\mathbb{E}\left\|\mathbf{y}^{(t+1)} - \mathbf{y}^*(\widehat{\mathbf{x}}^{(s)})\right\|^2$$

$$\leq \mathbb{E}\left\|\mathbf{y}^{(t)} - \mathbf{y}^*(\widehat{\mathbf{x}}^{(s)})\right\|^2 + \tau_{\text{eff}}^2[\gamma_y^s]^2\left[\frac{\sigma_L^2 n}{P}\sum_{i=1}^{n}\frac{w_i^2\|\boldsymbol{a}_i\|_2^2}{\|\boldsymbol{a}_i\|_1^2} + \frac{2\sigma_G^2 n}{P}\left(\frac{n-P}{n-1}\max_i w_i + \beta_L^2 \max_i \frac{w_i\|\boldsymbol{a}_i\|_2^2}{\|\boldsymbol{a}_i\|_1^2}\right)\right]$$

$$- 2\tau_{\text{eff}}\gamma_y^s\left(1 - 2\tau_{\text{eff}}\gamma_y^s L_f \frac{n}{P}\left[\frac{P-1}{n-1} + \beta_G^2\left(\frac{n-P}{n-1}\max_i w_i + \beta_L^2 \max_i \frac{w_i\|\boldsymbol{a}_i\|_2^2}{\|\boldsymbol{a}_i\|_1^2}\right)\right]\right)\mathbb{E}\left[\widetilde{\Phi}(\widehat{\mathbf{x}}^{(s)}) - \widetilde{F}(\widehat{\mathbf{x}}^{(s)}, \mathbf{y}^{(t)})\right]$$

$$+ \tau_{\text{eff}}\gamma_y^s L_f\left[1 + 2\tau_{\text{eff}}\gamma_y^s L_f \frac{n}{P}\left(\frac{P-1}{n-1} + \frac{n-P}{n-1}\max_i w_i + \beta_L^2 \max_{i,k}\frac{w_i a_i^{(k)}(\tau_i)}{\|\boldsymbol{a}_i\|_1}\right)\right]\sum_{i=1}^{n}\frac{w_i}{\|\boldsymbol{a}_i\|_1}\sum_{k=0}^{\tau_i-1}[a_i^{(k)}(\tau_i)]\Delta_{\mathbf{x},\mathbf{y}}^{(t,k)}(i), \tag{67}$$

since $\Delta_{\mathbf{y}}^{(t,k)} \leq \Delta_{\mathbf{x},\mathbf{y}}^{(t,k)}$. Using the bound on $\Delta_{\mathbf{x},\mathbf{y}}^{(t,k)}$ from Lemma C.2,

$$\sum_{i=1}^{n}\frac{w_i}{\|\boldsymbol{a}_i\|_1}\sum_{k=0}^{\tau_i-1}a_i^{(k)}(\tau_i)\Delta_{\mathbf{x},\mathbf{y}}^{(t,k)}(i) \leq 2\left([\eta_x^c]^2 + [\eta_y^c]^2\right)\sigma_L^2\sum_{i=1}^{n}w_i\|\mathbf{a}_{i,-1}\|_2^2 + 4M_{\mathbf{a}_{-1}}\left([\eta_x^c]^2 G_{\mathbf{x}}^2 + [\eta_y^c]^2\sigma_G^2\right)$$

$$+ 8[\eta_y^c]^2 L_f M_{\mathbf{a}_{-1}}\beta_G^2\mathbb{E}\left[\widetilde{\Phi}(\widehat{\mathbf{x}}^{(s)}) - \widetilde{F}(\widehat{\mathbf{x}}^{(s)}, \mathbf{y}^{(t)})\right]. \tag{68}$$

We substitute (68) in (67), and simplify the terms using the choice of $\gamma_y^s, \eta_y^c$ to get

$$\mathbb{E}\left\|\mathbf{y}^{(t+1)} - \mathbf{y}^*(\widehat{\mathbf{x}}^{(s)})\right\|^2$$

$$\leq \mathbb{E}\left\|\mathbf{y}^{(t)} - \mathbf{y}^*(\widehat{\mathbf{x}}^{(s)})\right\|^2 + \tau_{\text{eff}}^2[\gamma_y^s]^2\left[\frac{\sigma_L^2 n}{P}\sum_{i=1}^{n}\frac{w_i^2\|\boldsymbol{a}_i\|_2^2}{\|\boldsymbol{a}_i\|_1^2} + \frac{2\sigma_G^2 n}{P}\left(\frac{n-P}{n-1}\max_i w_i + \beta_L^2 \max_i \frac{w_i\|\boldsymbol{a}_i\|_2^2}{\|\boldsymbol{a}_i\|_1^2}\right)\right]$$

$$- \tau_{\text{eff}}\gamma_y^s\mathbb{E}\left[\widetilde{\Phi}(\widehat{\mathbf{x}}^{(s)}) - \widetilde{F}(\widehat{\mathbf{x}}^{(s)}, \mathbf{y}^{(t)})\right] + 4\tau_{\text{eff}}\gamma_y^s L_f([\eta_x^c]^2 + [\eta_y^c]^2)\left[\sigma_L^2\sum_{i=1}^{n}w_i\|\mathbf{a}_{i,-1}\|_2^2 + 2M_{\mathbf{a}_{-1}}(G_{\mathbf{x}}^2 + \sigma_G^2)\right].$$

using $\gamma_y^s, \eta_y^c$ that satisfy

$$2\tau_{\text{eff}}\gamma_y^s L_f \frac{n}{P}\left(\frac{P-1}{n-1} + \frac{n-P}{n-1}\max_i w_i + \beta_L^2 \max_{i,k}\frac{w_i a_i^{(k)}(\tau_i)}{\|\boldsymbol{a}_i\|_1}\right) \leq 1,$$

$$2\tau_{\text{eff}}\gamma_y^s L_f \frac{n}{P}\left[\frac{P-1}{n-1} + \beta_G^2\left(\frac{n-P}{n-1}\max_i w_i + \beta_L^2\max_i \frac{w_i\|\boldsymbol{a}_i\|_2^2}{\|\boldsymbol{a}_i\|_1^2}\right)\right] \le \frac{1}{4},$$

$$2L_f\frac{n}{P}\left[8[\eta_y^c]^2 M_{\mathbf{a}_{-1}}L_f\beta_G^2\right] \le \frac{1}{4}$$

Then the coefficient of $\mathbb{E}\left[\widetilde{\Phi}(\widehat{\mathbf{x}}^{(s)}) - \widetilde{F}(\widehat{\mathbf{x}}^{(s)}, \mathbf{y}^{(t)})\right]$ can we bounded by $-\tau_{\text{eff}}\gamma_y^s$. Consequently, by rearranging the terms and summing over $t$, we get the result.

$$\frac{1}{S}\sum_{t=sS}^{(s+1)S-1}\mathbb{E}\left[\widetilde{\Phi}(\widehat{\mathbf{x}}^{(s)}) - \widetilde{F}(\widehat{\mathbf{x}}^{(s)}, \mathbf{y}^{(t)})\right]$$

$$\le \frac{\mathbb{E}\left\|\mathbf{y}^{sS} - \mathbf{y}^*(\widehat{\mathbf{x}}^{(s)})\right\|^2}{\tau_{\text{eff}}\gamma_y^s S} + \tau_{\text{eff}}\gamma_y^s\frac{n}{P}\left[\sigma_L^2\sum_{i=1}^n\frac{w_i^2\|\boldsymbol{a}_i\|_2^2}{\|\boldsymbol{a}_i\|_1^2} + 2\sigma_G^2\left(\frac{n-P}{n-1}\max_i w_i + \beta_L^2\max_i\frac{w_i\|\boldsymbol{a}_i\|_2^2}{\|\boldsymbol{a}_i\|_1^2}\right)\right]$$

$$+ 4L_f([\eta_x^c]^2 + [\eta_y^c]^2)\left[\sigma_L^2\sum_{i=1}^n w_i\|\mathbf{a}_{i,-1}\|_2^2 + 2M_{\mathbf{a}_{-1}}(G_{\mathbf{x}}^2 + \sigma_G^2)\right].$$

$\square$

*Proof of Lemma C.4.* Let $t = sS, sS+1, \ldots, (s+1)S-1$, where $k$ is a positive integer. Let $\widehat{\mathbf{x}}^{(s)}$ is the latest snapshot iterate for the $\mathbf{y}$-update in Algorithm 1-Fed-Norm-SGDA+ . Then

$$\mathbb{E}\left[\widetilde{\Phi}(\mathbf{x}^{(t)}) - \widetilde{F}(\mathbf{x}^{(t)}, \mathbf{y}^{(t)})\right]$$

$$= \mathbb{E}\left[\widetilde{F}(\mathbf{x}^{(t)}, \mathbf{y}^*(\mathbf{x}^{(t)})) - \widetilde{F}(\widehat{\mathbf{x}}^{(s)}, \mathbf{y}^*(\widehat{\mathbf{x}}^{(s)})) + \widetilde{F}(\widehat{\mathbf{x}}^{(s)}, \mathbf{y}^*(\widehat{\mathbf{x}}^{(s)})) - \widetilde{F}(\widehat{\mathbf{x}}^{(s)}, \mathbf{y}^{(t)}) + \widetilde{F}(\widehat{\mathbf{x}}^{(s)}, \mathbf{y}^{(t)}) - \widetilde{F}(\mathbf{x}^{(t)}, \mathbf{y}^{(t)})\right]$$

$$\le \mathbb{E}\left[\widetilde{F}(\mathbf{x}^{(t)}, \mathbf{y}^*(\mathbf{x}^{(t)})) - \widetilde{F}(\widehat{\mathbf{x}}^{(s)}, \mathbf{y}^*(\mathbf{x}^{(t)}))\right] + \mathbb{E}\left[\widetilde{F}(\widehat{\mathbf{x}}^{(s)}, \mathbf{y}^*(\widehat{\mathbf{x}}^{(s)})) - \widetilde{F}(\widehat{\mathbf{x}}^{(s)}, \mathbf{y}^{(t)})\right] + G_{\mathbf{x}}\mathbb{E}\left\|\mathbf{x}^{(t)} - \widehat{\mathbf{x}}^{(s)}\right\|$$

$$\le 2G_{\mathbf{x}}\mathbb{E}\left\|\mathbf{x}^{(t)} - \widehat{\mathbf{x}}^{(s)}\right\| + \mathbb{E}\left[\widetilde{\Phi}(\widehat{\mathbf{x}}^{(s)}) - \widetilde{F}(\widehat{\mathbf{x}}^{(s)}, \mathbf{y}^{(t)})\right]. \tag{69}$$

where, $\mathbf{y}^*(\cdot) \in \arg\max_{\mathbf{y}}\widetilde{F}(\cdot, \mathbf{y})$ and (69) follows from $G_{\mathbf{x}}$-Lipschitz continuity of $F(\cdot, \mathbf{y})$ (Assumption 8). Next, we see that

$$\mathbb{E}\left\|\widehat{\mathbf{x}}^{(s)} - \mathbf{x}^{(t)}\right\| \le \sqrt{\mathbb{E}\left\|\widehat{\mathbf{x}}^{(s)} - \mathbf{x}^{(t)}\right\|^2} \qquad\qquad\text{(Jensen's inequality)}$$

$$\overset{(43)}{=} \sqrt{\mathbb{E}\left\|\tau_{\text{eff}}\gamma_x^s\sum_{t'=sS}^{t-1}\sum_{i\in\mathcal{C}^{(t')}}\tilde{w}_i\mathbf{g}_{\mathbf{x},i}^{(t')}\right\|^2}$$

$$\le \tau_{\text{eff}}\gamma_x^s\sqrt{(S-1)\sum_{t'=sS}^{t-1}\mathbb{E}\left\|\sum_{i\in\mathcal{C}^{(t')}}\tilde{w}_i\mathbf{g}_{\mathbf{x},i}^{(t')}\right\|^2}$$

$$\le \tau_{\text{eff}}\gamma_x^s(S-1)\sqrt{\frac{n}{P}}\sqrt{\sum_{i=1}^n\frac{w_i^2\|\boldsymbol{a}_i\|_2^2}{\|\boldsymbol{a}_i\|_1^2}(\sigma_L^2 + \beta_L^2 G_{\mathbf{x}}^2) + G_{\mathbf{x}}^2\left(\frac{P-1}{n-1} + \frac{n-P}{n-1}\sum_{i=1}^n w_i^2\right)}.$$

(from (59))

Using this bound in (69), and summing over $t$, we get

$$\frac{1}{S}\sum_{t=sS}^{(s+1)S-1}\mathbb{E}\left[\widetilde{\Phi}(\mathbf{x}^{(t)}) - \widetilde{F}(\mathbf{x}^{(t)}, \mathbf{y}^{(t)})\right] \le \frac{1}{S}\sum_{t=sS}^{(s+1)S-1}\mathbb{E}\left[\widetilde{\Phi}(\widehat{\mathbf{x}}^{(s)}) - \widetilde{F}(\widehat{\mathbf{x}}^{(s)}, \mathbf{y}^{(t)})\right]$$

$$+ 2\tau_{\text{eff}}\gamma_x^s G_{\mathbf{x}}(S-1)\sqrt{\frac{n}{P}}\sqrt{\sum_{i=1}^n\frac{w_i^2\|\boldsymbol{a}_i\|_2^2}{\|\boldsymbol{a}_i\|_1^2}(\sigma_L^2 + \beta_L^2 G_{\mathbf{x}}^2) + G_{\mathbf{x}}^2\left(\frac{P-1}{n-1} + \frac{n-P}{n-1}\sum_{i=1}^n w_i^2\right)}.$$

Finally, summing over $s = 0$ to $T/S - 1$ we get the result. $\square$

### C.4 Extending the result for Nonconvex One-Point-Concave (NC-1PC) Functions

Carefully revisiting the proof of Theorem 2, we notice that Lemma C.1 and Lemma C.2 do not rely on the concavity assumption. Lemma C.3 does use concavity of local functions $\{f_i\}$. However, it is only needed to derive (66). Further, this only requires concavity of local functions at a global point $\mathbf{y}^*(\widehat{\mathbf{x}}^{(s)})$. Therefore, as mentioned earlier in Remark 10, it holds even for NC-1PC functions. This is an independent result in itself, since we have extended the existing convergence result of local stochastic gradient method for convex minimization (concave maximization) problems, to a much more general one-point-convex minimization (or one-point-convex maximization) problem. Therefore, we restate it here for the more general case.

**Lemma C.5** (Local SG updates for One-Point-Concave Maximization). *Suppose the local loss functions $\{f_i\}$ satisfy Assumptions 1, 3, 4, 8. Suppose for all $\mathbf{x}$, all the $f_i$'s satisfy Assumption 9 at a common global minimizer $\mathbf{y}^*(\mathbf{x})$, and that $\left\|\mathbf{y}^{(t)}\right\|^2 \leq R$ for all $t$. If we run Fed-Norm-SGDA+ with the same conditions on the client and server step-sizes $\eta_y^c, \gamma_y^s$ respectively, as in Lemma C.3, then the iterates generated by Fed-Norm-SGDA+ also satisfy the bound in Lemma C.3.*

Next, Lemma C.4 also holds irrespective of concavity. Therefore, the resulting convergence result in Theorem 2 for nonconvex-concave minimax problems holds for a much larger class of functions. We restate the modified theorem statement briefly.

**Theorem.** *Suppose the local loss functions $\{f_i\}$ satisfy Assumptions 1, 3, 4, 8. Suppose for all $\mathbf{x}$, all the $f_i$'s satisfy Assumption 9 at a common global minimizer $\mathbf{y}^*(\mathbf{x})$, and that $\left\|\mathbf{y}^{(t)}\right\|^2 \leq R$ for all $t$. If we run Algorithm 1-Fed-Norm-SGDA+ with the same conditions on the client and server step-sizes $\eta_y^c, \gamma_y^s$ respectively, as in Theorem 3, then the iterates generated by Algorithm 1-Fed-Norm-SGDA+ also satisfy the bound in Theorem 3.*

*Remark* 11. Again, choosing client weights $\{w_i\}$ the same as in the original global objective $\{p_i\}$, we get convergence in terms of the original objective $F$.

## D  Additional Experiments

For communicating parameters and related information amongst the clients, ethernet connections were used. Our algorithm was implemented using parallel training tools in PyTorch 1.0.0 and Python 3.6.3.

For both robust NN Training and fair classification experiments, we use batch-size of 32 in all the algorithms. Momentum parameter 0.9 is used only in Momentum Local SGDA(+).

**Robust NN Training.**   Here we further explore performance of Fed-Norm-SGDA+ on the robust NN training problem. We use VGG-11 model to classify CIFAR10 dataset. In Figure 6, we show the training loss curves corresponding to the results in Figure 3 on varying number of local steps. Similarly, in Figure 6, we show the training loss curves corresponding to Figure 4 on the effect of partial participation. In Figure 7, we demonstrate the effect of increasing data heterogeneity across clients, whle in Figure 9 we show the advantage of using multiple clients for the federated minimax problem. With $k$-fold increase in $n$, we observe an almost $k$-fold drop in the number of communication rounds needed to reach a target test accuracy (70% here.).

We use batch-size of 32. Momentum parameter 0.9 is used only in Local SGDA+(M).

Table 3: Parameter values for experiments in robust NN training experiments.

| Communication rounds | 1-100 | 101-200 | >200 |
|---|---|---|---|
| Client Learning Rate ($\eta_y^c$) | 0.02 | $2 \times 10^{-3}$ | $2 \times 10^{-4}$ |
| Client Learning Rate ($\eta_x^c$) | 0.016 | $1.6 \times 10^{-3}$ | $1.6 \times 10^{-4}$ |
| Server Learning Rate ($\gamma_x^s = \gamma_y^s$) | 1 | 1 | 1 |

**Fair Classification**   We also demonstrate the impact of partial client participation in the fair classification problem. Figure 10 complements Figure 10 in the main paper, evaluating fairness of a VGG11 model on

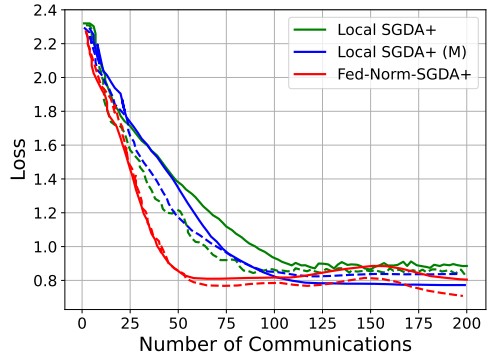

Figure 6: Comparison of the effect of heterogeneous number of local updates $\{\tau_i\}$ on the performance of Fed-Norm-SGDA+ (Algorithm 1), Local SGDA+, and Local SGDA+ with momentum, while solving (7) on CIFAR10 dataset, with VGG11 model. The solid (dashed) curves are for $E = 5$ ($E = 7$), and $\alpha = 0.1$.

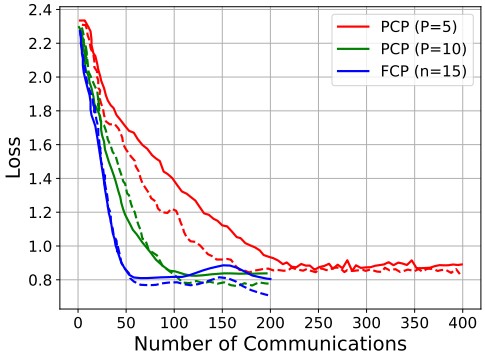

Figure 7: Comparison of the effects of partial client participation (PCP) on the performance of Fed-Norm-SGDA+, for the robust NN training problem on the CIFAR10 dataset, with the VGG11 model. The figure shows the robust test accuracy. The solid (dashed) curves are for $\alpha = 0.1$ ($\alpha = 1.0$).

CIFAR10 dataset. We have plotted the test accuracy of the model over the worst distribution. With an increasing number of participating clients, the performance consistently improves.

Batch-size of 32 is used. Momentum parameter 0.9 is used only in Local SGDA (M).

Table 4: Parameter values for experiments in fair classification experiments.

| Client Learning Rate ($\eta_y^c$) | 0.02 |
|---|---|
| Client Learning Rate ($\eta_x^c$) | 0.016 |
| Server Learning Rate ($\gamma_x^s = \gamma_y^s$) | 1 |

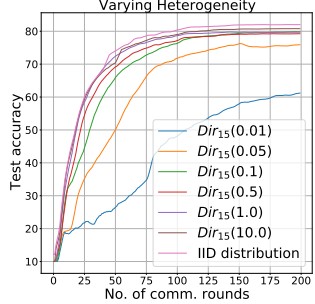

Figure 8: Effect of inter-client data heterogeneity (quantified by $\alpha$) on the performance of Fed-Norm-SGDA+ in a robust NN training task.

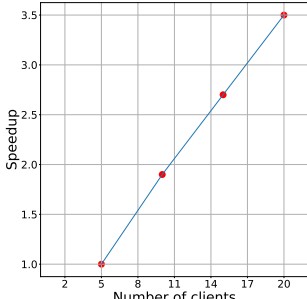

Figure 9: Effect of increasing client-set on the performance of Fed-Norm-SGDA+ in a robust NN training task.

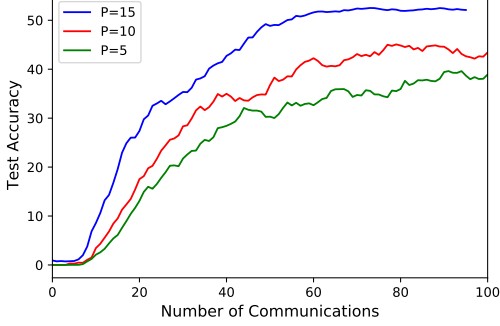

Figure 10: Effect of partial client participation on the performance of Fed-Norm-SGDA in a fair image classification task.

