# OpenReview forum: "Federated Minimax Optimization with Client Heterogeneity"
_TMLR — Accepted by TMLR_

### Review · Reviewer_YTqc · 2023-06-09

**Summary Of Contributions:**

This paper considers federated minimax problems, motivated by applications such as robust/adversarial training and fair classification. The paper studies variants of the stochastic gradient descent-ascent algorithm with local updates, while also accounting for challenges such as partial client participation and "systems heterogeneity," realized as different clients running different numbers of local updates per round. In this sense, the paper can be seen as an extension of the FedNova method to minimax problems. Theoretical analysis in several relevant scenarios shows that the proposed methods achieve superior convergence rates than prior work. Experiments validate the theory.

**Audience:**

Yes

**Broader Impact Concerns:**

None noted

**Claims And Evidence:**

Yes

**Requested Changes:**

1. The notion of stochastic gradient complexity in Definition 2 is only well-defined and meaningful if all clients perform the same number of local steps and participate in exactly the same number of rounds during training. I suggest updating it to reflect the more general scenario considered in the paper.

2. Davis & Drusvyatskiy (2019) study the Moreau envelope in the context of non-smooth weakly convex minimization. It may be worth mentioning if any of the assumptions (Assumption 1 or 7) imply that $\Phi$ is weakly convex. Is an additional assumption needed for $\Phi$ to be bounded below?

3. In the experiments, in addition to showing test accuracy please also include training accuracy, since this is the quantity the algorithm is actually optimizing.

4. Nit: In Theorems 1 and 2, please clarify what are "appropriate" learning rates. In particular, which parameters of the problem do they depend on?

**Strengths And Weaknesses:**

Overall this is a very good paper and I'm supportive of it appearing in TMLR. The topics addressed in the paper are certainly of interest to the federated learning community. The paper is generally well-written and the results appear to be both sound and a substantial improvement in rates over the previous literature. By properly addressing issues like partial client participation and systems heterogeneity, this paper reduces the gap between federated minimax optimization algorithms/analyses and federated learning systems in practice.

No significant weaknesses noted. Rather than be repetitive, I mention the few suggested changes I have in the next section.

---

### Review · Reviewer_2XcN · 2023-07-07

**Summary Of Contributions:**

This paper focuses on federated learning in the context of minimax optimization, with the primary objective of reducing communication costs while preserving the optimal convergence rate. The key contribution lies in considering two settings: nonconvex-PL (including nonconvex-strongly-concave as a specific case) and nonconvex-one-point-concave (including nonconvex-concave as a special case).

In comparison to previous works, this research offers two generalizations within the settings:

a. System Heterogeneity: The approach accounts for scenarios where each client performs a distinct number of local steps.

b. Weighted Component Functions


**Audience:**

Yes

**Claims And Evidence:**

Yes

**Requested Changes:**

Some other minor comments:

1. The definition and usage of $w_i$ and $p_i$ are confusing. In (2), $w_i$ is initially defined, but later, in the line above (3) and Corollary 1.1, it seems that we have the ability to set $w_i$. I recommend improving the clarity surrounding this aspect, although it is currently understandable.

2. Typo in (2): $G_i^t$ should be $G_{x, i}^t$.

3. Typo in the equation of (8): $y^{\prime}$ should be corrected.



**Strengths And Weaknesses:**

## Strength

1. It achieves state-of-the-art gradient complexities in both NC-SC and NC-C settings.

2. The improvement in the communication cost is evident in theory. In NC-SC setting, it improves from $O(\epsilon^{-3})$ to $O(\epsilon^{-2})$. In NC-C setting, it improves from  $O(\epsilon^{-7})$ to $O(\epsilon^{-4})$.

3. In the full participation setting, a linear speedup is achieved, which is reasonable.

## Weakness and questions

1. I think it does not really consider different optimizers for each client. In Algorithm 1, each client only utilizes a stochastic gradient descent/ascent type update, differing solely in the stepsize employed. In this case, the introduction of the more general SGDA update rule on Page 5 appears unnecessary.

2. The derivation of Corollary 1.2 is unclear. In equation (4), the first term incorporates $\sqrt{\frac{n-P}{n-1}}$, but in the proof of Corollary 1.2 on Page 26, the square root for this term is absent in the first equation. Additionally, in the second equation of this proof, when full client participation is assumed, it seems that the second term should be $1/\tau T$ rather than $1/T$.

---

### Review · Reviewer_E6s9 · 2023-08-05

**Summary Of Contributions:**

This paper considers optimizing a nonconvex minimax objective in a federated setting where the data distributions and system capabilities vary among participating data holders.  The authors propose a general communication efficient (via periodic averaging) method that facilitates an unequal number of local steps among clients and establish convergence under different assumptions on the loss functions. The proposed method, a simple normalization tweak incorporated into standard Local SGDA,  subsumes existing literature on Local SGDA as a special case and obtained rates improve upon known results in terms of communication complexity.

The key observation is that in the presence of heterogeneous local updates, the Local SGDA converges to a different objective where the components (local empirical losses) are weighted differently, as a result by proposer normalization it is guaranteed to converge to the original empirical loss. The theoretical results are complemented by empirical results on robust neural training and fair classification problems.


**Audience:**

Yes

**Claims And Evidence:**

Yes

**Requested Changes:**

- In the experiments, it is not clear what learning rates used both in client and server sides. In Lemma B.5, the server side learning rate for x is $O(\kappa^2)$ times smaller than y, and I was wondering if this is the case in real experiments as well.

- The reduction in Eq 2, reminded of analysis of networked distributed optimization over a star network and I was left wondering how the proof techniques here (specifically handling different number of local updates) compares to the analysis methods in that context, e.g., the paper below:

Koloskova, Anastasia, Nicolas Loizou, Sadra Boreiri, Martin Jaggi, an Sebastian Stich. "A unified theory of decentralized sgd with changing topology and local updates." In International Conference on Machine Learning, pp. 5381-5393. PMLR, 2020.


-- Minor --

Some examples below (there are many more, and the authors are encouraged to proofread carefully)

- Page 5: the equation right after paragraph starting ‘Generalized Local SGDA Update Rule.’, both updates are ascent, for x it should be decent
- Page 2: Section 5.1, 5.2 → Sections
- Eq 2 needs alignment
- Lemma A.4 & A.5, these should be vectors and bold faced
- …


**Strengths And Weaknesses:**

The paper proposes a simple yet effective algorithm that simultaneously tackles inter-client data heterogeneity, partial client participation, and system heterogeneity issues in federated nonconvex minimax optimization-- making it very practical. The proposed algorithm also enjoys provable convergence rates for NC-SC/PL and NC-C/1PC objectives, and improves/mathces upon known results in terms of communication complexity. The paper nicely discusses the results in the context of existing literature and the analysis looks sound as far as I checked. Overall, the paper is well-written in most aspects (except minor edits that can be fixed with a careful proofreading– see below for some examples).

---

### Decision · Action_Editor_Fvk1 · 2023-10-21

**Recommendation:** Accept with minor revision

**Comment:**

All the reviewers recommended acceptance. The authors should implement the changes and the corrections they have promised in their discussion with the reviewers. Regarding the discussion about the learning rates for Theorems 1 and 2, please at least include explicit pointers in the appendix for their definitions if you do not include them explicitly in the main paper.

**Audience:**

Yes.

**Claims And Evidence:**

Yes.